# Cenozoic deformation in the Tauern Window (Eastern Alps) constrained by in-situ Th-Pb dating of fissure monazite

Emmanuelle Ricchi[1], Christian A. Bergemann[2], Edwin Gnos[3], Alfons Berger[4], Daniela Rubatto[4,5],

Martin J. Whitehouse[6], Franz Walter[7]

[1]*Department of Earth Sciences, University of Geneva, Rue des Maraîchers 13, 1205 Geneva, Switzerland*

[2]*Institute of Geosciences, Heidelberg University, Im Neuenheimer Feld 236, 69120 Heidelberg, Germany*

[3]*Natural History Museum of Geneva, Route de Malagnou 1, 1208 Geneva, Switzerland*

[4]*Institute of Geological Sciences, University of Bern, Baltzerstrasse 1+3, 3012 Bern, Switzerland*

[5]*Institute of Earth Sciences, University of Lausanne, Geopolis, Lausanne, 1015 Switzerland*

[6]*Swedish Museum of Natural History, Box 50007, SE104-05 Stockholm, Sweden*

[7]*Institut für Erdwissenschaften, Karl-Franzens-Universität, Universitätsplatz 2, 8010 Graz, Austria*

Highlights:

- In-situ dating of hydrothermal monazite-(Ce)

- New constraints on the exhumation of the Tauern metamorphic dome

- Distinct tectonic pulses recorded from East to West

Keywords: Hydrothermal monazite; Alpine fissures; Th-Pb dating; Tauern Window; Protracted deformation

**Abstract.** Thorium-Pb crystallization ages of hydrothermal monazites from the western, central and eastern Tauern Window provide new insights into Cenozoic tectonic evolution of the Tauern metamorphic dome. Growth domain crystallization ages range from $21.7 \pm 0.4$ Ma to $10.0 \pm 0.2$ Ma. Three major periods of monazite growth are recorded between ~22 – 20 (peak at 21 Ma), 19 – 15 (major peak at 17 Ma) and 14 – 10 Ma (major peak around 12 Ma), respectively interpreted to be related to prevailing N-S shortening, in association with E-W extension, beginning strike-slip movements, and reactivation of strike-slip faulting. Fissure monazite ages largely overlap with zircon and apatite fission tracks data. Besides tracking the thermal evolution of the Tauern dome, monazite dates reflect episodic tectonic movement along major shear zones that took place during the formation of the dome. Geochronological and structural data from the Pfitschtal area in the western Tauern Window show the existence of two cleft generations separated in time by 4 Ma and related to strike-slip to oblique-slip faulting. Moreover, these two phases overprint earlier phases of fissure formation.

## 1      Introduction

In-situ Th-Pb dating of hydrothermal fissure monazite-(Ce) (in the following simply monazite) has recently been demonstrated to be a reliable method for dating tectonic activity under retrograde metamorphic conditions (Bergemann et al., 2017, 2018, 2019, 2020; Berger et al., 2013; Fitz-Diaz et al., 2019; Gasquet et al., 2010; Gnos et al., 2015; Grand'Homme et al., 2016a; Janots et al., 2012; Ricchi et al., 2019). These studies conducted through the entire Alpine orogenic belt allowed constraining tectonic activity in relation with exhumation and fault activity under retrograde lower greenschist to sub-greenschist facies metamorphic conditions.

Hydrothermal fissure monazite, concentrating LREE, Th and U, generally crystallizes in Ca-poor

lithologies, outside the stability field of titanite or epidote/allanite. However, once formed,

hydrothermal processes may cause dissolution-reprecipitation events leading to resetting of the

monazite Th-Pb decay system in parts of the crystal. Chemically and isotopically homogeneous

crystals indicate a single, rapid growth episode (e.g., Grand'Homme et al., 2016a). However,

crystals showing different growth domains indicative of successive growth episodes are more

common. In other cases, parts of, or entire grains display a patchy zoning due to dissolution-

reprecipitation processes (e.g., Ayers et al., 1999; Grand'Homme et al., 2016b). These processes

involve element fractionation resulting in crystal zones with often distinct Th/U values (Seydoux-

Guillaume et al., 2012).

The advantage of using hydrothermal monazite for dating tectonic activity is related to the high

closure temperature of monazite of >800°C, implying that diffusion in monazite is negligible

(Cherniak et al., 2004; Gardés et al., 2006, 2007) under P-T conditions at or below $450 - 500$ °C

and $0.3 - 0.4$ GPa (e.g., Mullis et al., 1994; Mullis, 1996; Sharp et al., 2005) where hydrothermal

fissures form. Fissure monazites date crystallization following chemical disequilibrium within a

fissure. This causes a dissolution-precipitation cycle that may include dissolution or partial

dissolution of existing fissure monazite. This has the consequence that late

dissolution/precipitation steps may be well recorded, whereas earlier growth domains may be

completely destroyed. Thus, monazite crystallization due to chemical disequilibrium is

interpreted as being related to tectonic activity (e.g. volume change, fissure propagation,

exposure of fresh host-rock, delamination of fissure wall, seismic activity, fluid loss or gain).

Recent studies have shown that fissure monazite typically forms between generally lower to

higher $200 - 400$°C (Gnos et al., 2015; Janots et al., 2019). For this reason, fissure monazite ages

are generally interpreted as dating crystallization or re-crystallization. Monazite geochronology

can thus be utilized to constrain shear and damage zone activity under greenschist and very-low

grade metamorphic conditions at least down to 200°C (e.g. Bergemann et al., 2017, 2018; Gnos

et al., 2015).

Fissures and clefts develop close to the brittle-ductile transition (< 450°C, Mullis, 1996) and are

usually oriented perpendicular to the foliation and lineation of the host-rock (Gnos et al., 2015).

Fissures are generally straight when they form and either became enlarged by subsequent tectonic

activity to form fluid-filled dm- to m- sized clefts, displaying a more open shape with rounded

surfaces (e.g. Ricchi et al., 2019) when the stress field retains the same orientation, or became

completely filled to form mineral veins. However, they may show complex shape when the stress

field direction changes during deformation. Fluid inclusion studies (e.g. , Mullis, 1996) show that

clefts generally suffered several deformation episodes.

Interaction of the fluid that fills the fissures with the surrounding rock leads to dissolution of

minerals in the wall rock and mineral precipitation in the fissure. As long as deformation

continues, fluid-filled clefts will react to deformation via dissolution-precipitation cycles due to

disequilibrium between fluid, rock wall and mineral assemblage within the cleft (e.g. Putnis,

2009). Thus, hydrothermal minerals like monazite do not only grow following the initial fissure

formation but form, continue to grow, or dissolve during subsequent deformation stages. The

timing of these growth or alteration stages may not always be resolvable with the precision of

currently available geochronological methods, but different growth stages may be distinguishable

through differences in the chemical composition (Grand'Homme et al., 2018). In contrast to the

surrounding country rock, the fissures and clefts remain highly reactive at low temperature due to

the presence of fluids. For this reasons, deformation steps during brittle deformation may be

registered through mineral growth or recrystallization in clefts (e.g., Berger et al., 2013) down to

conditions where clay minerals form in fault gauges.

The Tauern window (TW) is a thermal and structural dome of the eastern Alps (Fig. 1) exhumed

over a period of about 30 Ma starting from the Early Oligocene (e.g. Rosenberg et al., 2018;

Schmid et al., 2013). Previous monazite crystallization ages obtained in the eastern sub-dome of

the TW record tectonic activity between $19.0 \pm 0.5$ and $15.0 \pm 0.5$ Ma (Gnos et al., 2015). In the

current study, monazite geochronology is extended to the entire TW, in order to investigate its

Cenozoic deformation history. We particularly aim to establish a chronological record for the

younger exhumation history recorded by fissure monazite crystallization, to be compared with

known deformation phases.

A total of 23 monazite grains together with provenance data, and in some cases host-rock

information were dated (Table 1). Seven grains come from the western limb of the TW (INNB1

ZEI1, SCHR1, MAYR4, PFIT1, BURG2 and PLAN1; samples 1 to 7; Fig. 1), another seven

grains come from the eastern border of the western sub-dome (central TW; SCHEI1, HOPF2,

GART1, NOWA3, GART3, STEI2 and KNOR1; samples 8 to 14; Fig. 1) and nine grains were

collected in the eastern sub-dome (KAIS6, SALZ18, LOHN4, ORT1, EUKL2, HOAR1,

MOKR1, SAND1 and REIS1; samples 15 to 25; Fig. 1, Table 1). In order to capture at best the

tectonic activity during the exhumation of the TW, the investigated samples were selected in a

way that gives priority to sample localities in regions affected by major fault zones and at

lithological boundaries. In the following we will discuss the ages in term of sample ID numbers

(1-25) provided by Table 1.

**2      Geologic settings**

In the largest tectonic window of the Austroalpine nappe stack, the TW, Penninic (Glockner

Nappe System and Matrei Zone; Fig. 1) and Subpenninic nappes (mainly the Venediger Duplex)

are exposed (e.g. Schmid et al., 2013; Fig. 1). The TW metamorphic and structural dome consists of two sub-domes, with E-W striking upright folds in the internal parts and bordered by two major normal faults, the Katschberg Normal Fault (KNF) in the East and the Brenner Normal Fault (BNF) in the West (Fig. 1). The western sub-dome is dissected by numerous sinistral shear zones (Ahorn Shear Zone (ASZ), Olperer Shear Zone (OSZ), Tuxer Shear Zones (TSZ), Greiner Shear Zone (GSZ) and Ahrntal Shear Zone (AhSZ)) and is bordered by the Salzach-Ennstal-Mariazell-Puchberg fault (SEMP) in the north (Fig.1). The eastern sub-dome is bordered to the east by the Katschberg Normal Fault (KNF), continuing to the north into the dextral Katschberg Shear Zone System (KSZS), and to the south into an unnamed sinistral shear zone and oriented parallel to the Mölltal Fault (MöF). The deformation history of these fault complexes will be discussed later.

The Alpine evolution of the TW started in the Early Paleocene with the accretion and subduction of the Piemont-Liguria Ocean (Matrei Zone; Fig. 1) under the Apulian margin (Austroalpine nappe stack; e.g. Schmid et al., 2004, 2013; D1 deformation of Schmid et al., 2013; Fig.1, Table 2). In the Middle Eocene, the Valais Ocean and parts of the distal European margin (Glockner Nappe System, Eclogite Zone and parts of the Modereck Nappe System; Fig. 1) were equally subducted below the Austroalpine nappe stack and the Matrei Zone accreted during D1 deformation (D2 deformation of Schmid et al. 2013; Table 2). In the Late Eocene, exhumation was achieved by extrusion of the high-pressure units that went together with major folding of the D2 thrust formed between the subducted Glockner Nappe System and Modereck Nappe System (D3 deformation of Schmid et al. 2013; Table 2). In the Early Oligocene, nearly contemporaneous break off of the subducting European slab and formation of the Venediger Duplex (crustal-scale duplex structure) occurred, followed by the "Tauernkristallisation" (reheating of the whole nappe stack to amphibolite-facies conditions) (D4 deformation of Schmid

et al. 2013; Table 2). This was followed by an inversion of subduction polarity at ~23 to 21 Ma

(e.g., Rosenberg et al., 2018; Scharf et al., 2013; Schmid et al., 2013; Table 2). The following

exhumation of the TW started in the Early to Middle Miocene by Alpine N-S collisional

shortening and E-W orogen-parallel extension leading to folding, erosion and lateral extrusion

through shear zone development (e.g. Luth and Willingshofer, 2008; Rosenberg and Berger,

2009; Rosenberg and Garcia, 2011; Schmid et al., 2004, 2013; Selverstone, 1988; D5

deformation of Schmid et al., 2013). Previous shear zone age dating in the TW was achieved

using different geochronometers: Rb–Sr whole-rock–phengite dating (20 Ma; Blanckenburg et

al., 1989), Rb – Sr whole-rock–white mica dating (39 – 16 Ma; Glodny et al., 2008), Sm – Nd

dating on garnet (27.5 – 20 Ma; Pollington and Baxter, 2010, 2011) and $^{40}Ar/^{39}Ar$ dating on mica

(35 – 28 Ma; Urbanek et al., 2002). A recent detailed study by Schneider et al. (2013) using

texturally-controlled in-situ $^{40}Ar/^{39}Ar$ dating of syn-kinematic phengite and K-feldspar returned

ages between 33 – 15, 24 – 12 and 20 – 7 Ma. They were interpreted as recording deformation

along three major shear zones (ASZ, TSZ and GSZ respectively) of the western sub-dome.

**Sample location**

Fissure monazite is rare and difficult to find, meaning that this study could not have been

conducted without the help of crystal searchers who provided samples. Fissure monazites were

selected to cover all parts of the TW areas with known shear zones within it. It was, however,

unfortunately not possible to obtain exact coordinates for all of the samples (Table 1). This is due

to the rarity of fissure monazite, so that some samples were obtained from old finds or

collections. In other cases, the crystal searcher could not anymore precisely identify the fissure in

which the monazite was found. These samples are marked with "approx." in Table 1. We could

therefore only revisit some of the sample locations in order to add structural information.

Experience from other parts of the Alps (e.g. Bergemann et al., 2017, 2019; Ricchi et al., 2019)

shows that fissure monazite sampled within the damage or central zones of a shear zone generally

records shear zone activity well. Information on the source localities, host-rocks, degree of alpine

metamorphism and mineral associations is in Table 1.

**3      Methods**

The crystals where polished individually on a lapidary disk and embedded in epoxy together with

the monazite standard "44069" (425 Ma, Aleinikoff et al., 2006), following the same procedure

as Bergemann et al. (2017). Backscatter electron (BSE) images were acquired in order to

investigate the internal textural features of each grain (e.g. zoning, evidences of alteration, etc.)

using an EDS-equipped JEOL JSM7001F and a Zeiss DSM940A electron microscope at the

University of Geneva with a beam current of 3.5 nA and acceleration voltage of 15kV. BSE

images helped in the selection of Secondary-Ion Mass Spectrometry (SIMS) spot analysis points,

carefully placed in chemically distinct domains.

Ion probe U-Th-Pb analyses of 15 monazite crystals were conducted at the SwissSIMS Ion

microprobe facility, University of Lausanne, Switzerland and analyses of another 8 crystals were

performed at the Nordsim facility, Swedish Museum of Natural History, Stockholm (Tables 1 and

3). Both laboratories are equipped with a Cameca IMS 1280-HR instrument. The instruments

were run following the procedure of Janots et al. (2012), applying a -13 kV $O^{2-}$ primary beam, an

intensity of ~3 and 6 nA focused on the sample (SwissSIMS and Nordsim respectively) to

produce a spot of 15 – 20 micron in diameter. A mass resolution of 4300 – 5000 (M/ΔM,

$^{208}Pb/^{232}Th$ at 10% peak height) and an energy window 40 eV were applied, with data collection

in peak hopping mode using an ion-counting electron multiplier. All the unknowns were

standardised to 44069 (425 Ma; Aleinikoff et al., 2006) monazite and the uncertainty on the

standard $^{208}Pb/^{232}Th$ – ThO/Th calibration in each session was 1.7% on average.

A $^{207}Pb$ and $^{204}Pb$ common lead (Pbc) correction calculated at time zero was applied to the data

acquired at the SwissSIMS and Nordsim (Table 3), using the terrestrial Pb evolution model of

Stacey and Kramers (1975). The Cameca Customisable Ion Probe Software (CIPS) was used for

data reduction. $^{204}Pb$- and $^{207}Pb$-corrected ages agree within uncertainty (Table 3), but we

preferred to discuss $^{207}Pb$-corrected ages because they are more robust and consistent (better

statistics and less scatter in the data).  Calculation of weighted mean ages, based on $^{207}Pb$-

correction, and plotting was carried out using the program IsoPlot Ex 4.1 (Ludwig, 2003). Single

and weighted mean ages (or average ages) are quoted at the 1 sigma and at 95% confidence level

in the text, respectively.

Weighted mean $^{208}Pb/^{232}Th$ ages were calculated for each growth domain following the approach

of Bergemann et al. (2017, 2018, 2019, 2020) and Ricchi et al. (2019). Distinct chemical and

textural domains were carefully defined in each grain based on Th concentrations as function of

U concentrations and BSE image information. Since fissure monazite is dissolved and re-

precipitated under changing chemical conditions (e.g. Grand'Homme et al., 2018), spot analyses

affected by Pbc (resulting in older dates directly related to higher Pbc, i.e. positive Age-f208

correlation), inclusions or with high uncertainty ($1\sigma > 1$ abs.) were removed from the dataset and

marked in italic in Table 3. However, spot dates located on dissolution trails, generally providing

younger dates, were considered in the age ranges because they likely record a later phase of

monazite crystallization.

## 4      Results

### 4.1      Field observations

An example of deformed fissures and different stages of fissure formation is well exposed in outcrops along the road leading to Pfitscherjoch (in vicinity to the PFIT1 sample locality 5, Western TW; Table 1), where two fissure generations are present (Fig. 2a and b). In this outcrop an earlier fissure generation ($C_2$, green ellipses) is partly deformed during subsequent deformation, and a younger generation of fissures ($C_3$, blue ellipses) is also present. Sub-horizontal fissures ($C_3$) seem linked to a strongly inclined lineation ($L_3$, blue arrows), whereas older sub-vertical fissures ($C_2$) seem related to a weakly-inclined strike-slip lineation ($L_2$, green arrows). The older fissures are wider and sigmoidal in shape and contain muscovite which is not found in the younger fissures. In some cases younger fissures crosscut older ones (Fig. 2b). Moreover, the orientation of the foliation ($S_{2, 3}$, Fig. 2c) of these two fissure generations ($C_2$ and $C_3$) is different from the foliation ($S_1$, Fig. 2c) of early fissure formation mainly observed in the eastern part of the TW ($C_1$, Fig. 2c, discussed below). This suggests that in the Pfitscherjoch area, early fissures $C_1$ were overprinted by younger tectonic movements.

The large majority of the fissures present in all the investigated localities are oriented sub-vertically ($C_1$ and $C_2$ type in Fig. 2c), roughly striking NE-SW. For $C_2$, this would indicate a similar direction of extension for the development of this fissure type, which is in line with paleostress orientations provided by Bertrand et al. (2015). However, even if all sub-vertical fissures are sub-parallel, at least two generations exist. (i) Early sub-vertical fissures ($C_1$, Fig. 2c) are related to flat foliation ($S_1$) and E-W stretching lineation ($L_1$), these are oriented perpendicular to the main fold axes (and lineation) of the TW and are associated with E-W extension (e.g. Gnos et al., 2015; Rosenberg et al., 2018; Schneider et al., 2013). (ii) Younger sub-vertical fissures ($C_2$,

Fig. 2c) are associated with sub-vertical E-W oriented foliation ($S_2$) and flat to inclined lineation

($L_2$), and are oriented perpendicular to strike-slip faults (mainly in the western part of the TW;

Fig. 2c). At Pfitscherjoch, the shape of $C_2$ fissures, indicating overprinting by sinistral sense of

shear, is in agreement with the larger scale sinistral shearing of the GSZ shear zone. (iii) A third

generation of fissures ($C_3$, Fig. 2c) is locally observed, for example, in the Pfitscherjoch locality

(Fig. 2a and b) and is at high angle with $C_1$ and $C_2$ fissures. This third fissure generation observed

in the Pfitscherjoch locality is associated with a subvertical E-W striking foliation ($S_3$). Stretching

lineation related to the BNF activity is sub-parallel to $C_3$ lineation, however its foliation is

striking N-S (Fig. 2c). We suggest that $C_3$ fissures are related to oblique-slip movements in line

with the observed E-W striking foliation and not to the BNF activity.

**4.2    Monazite dating and composition**

The monazite grains selected for in-situ Th-Pb dating are mm-sized and, when BSE zoning is

visible, it shows two distinct textures: regular and patchy (Figs. 3, 4 and 5; Table 4). The term

regular refers to crystals showing growth-zonation, whereas a patchy texture is interpreted as

replacement by secondary dissolution/reprecipitation processes (e.g. Ayers et al., 1999;

Bergemann et al., 2018, 2019, 2020; Gnos et al., 2015). Thorium and U contents of the dated

fissure monazites display a large variability, ranging between ~200 to 63,000 ppm Th and ~2 to

2000 ppm U, with variations in Th/U ratio from 1 up to ~7000 (Figs. 3, 4 and 5; Table 3). $^{232}$Th-

$^{208}$Pb ages presented on the right-hand panel of figures 3, 4 and 5 are arranged according to the

order established in Table 3. A detailed description of each monazite grain is provided in the

Supplementary Information. Average ages are reported for group of dates on texturally and/or

chemically similar domains. In order to ensure that a group of dates from a domain is internally

consistent, rare outliers have been excluded to bring the MSWD within acceptable values

(MSWD < 3; Spencer et al., 2016). In a few cases the dates for specific monazite domains have a

scatter above analytical uncertainty (e.g. grain 6, 9, 24), which probably reflects the complex

formation process of fissure monazite.

The investigated grains from the western part of the western TW sub-dome come from the

Venediger Duplex, with the exception of sample 6, which comes from the Glockner Nappe

System (Fig. 1; Table 1). The samples 2, 4 and 6 were collected near to the major Brenner normal

fault, which delimits the TW to the west, and samples 1, 3, 5 and 7 were collected in the vicinity

of sinistral strike-slip faults (Fig. 1). Average growth domain ages range from $20.9 \pm 0.6$ to $10.0 \pm$

$0.2$ Ma (samples 3 and 2) with the youngest ages recorded in the western TW (Figs. 1, 3 and 6a;

Tables 3 and 4).

The central part of the TW displays growth domain ages between $18.3 \pm 1.1$ and $10.4 \pm 0.2$

(samples 8 and 14; Figs. 1 and 4; Tables 3 and 4), but the majority of the dated crystals in this

area record ages around 17 Ma (Fig. 6b). Samples 8, 9 and 10 were collected between the eastern

and western termination of the ASZ and the SEMP fault (Fig. 1). Another three samples (11, 12

and 13; Table 1) were collected in the northern prolongation of the AhSZ and a seventh monazite

(grain 14) was sampled near the southern border of the eastern part of the western sub-dome (Fig.

1).

The oldest ages are principally recorded in the eastern part of the TW at around 21 Ma (Fig. 6c).

Average ages of growth domains range from $21.7 \pm 0.4$ to $13.6 \pm 0.6$ Ma (samples 19a and 25;

Figs. 1, 4 and 6c; Tables 3 and 4). The samples were mainly collected at the western border of the

eastern sub-dome, in the Venediger Duplex or near the boundary with the Glockner/ Modereck

Nappe Systems (Fig. 1). Sample 25 was taken at some distance from the other samples, near the

south-eastern border of the eastern sub-dome (Fig. 1).

**5      Discussion**

**5.1     Fissure monazite ages**

The oldest monazite ages of $21.7 \pm 0.4$ to $19.9 \pm 0.3$ Ma (found in samples 19a and 20; Figs. 1, 6c

and 7a and d) are common in the eastern TW, but can also be found in the western area (Fig. 7a, c

and d, red symbols). This in line with regional fault activity recorded at ~21 Ma based on Pleuger

et al. (2012) (Fig. 8a) which corresponds to the main indentation phase (Favaro et al., 2017). We

interpret these as a first monazite crystallization event during E-W extension in association with

the dome formation (N-S shortening) when the existing clefts reached P-T conditions at which

fissure monazite starts to grow (phase 1, red symbols in Fig. 7). When comparing an assumed

fissure formation temperature of 450°C (typically obtained in quartz fluid inclusion studies on

early alpine fissures, (e.g., Mullis, 1996) with thermochronological data of the eastern TW

(compiled in Wölfler et al. 2012), the onset of fissure formation, predating primary monazite

crystallization, is estimated at around 25 Ma. Based on a comparison with thermochronological

data, monazite crystallization recorded between 19 – 15 Ma was estimated to have occurred at

~200 – 300°C in the eastern TW (Gnos et al., 2015). New monazite ages from this study in the

eastern TW are up to ~22 Ma (sample 19 in Fig. 1), suggesting that early monazite crystallization

in the area may have occurred at higher temperatures of 350 – 400°C.

While early fissure formation is related to E-W extension (leading to flat foliation and E-W

mineral lineations; Fig. 2c), we suggest that monazite formation also occurred along the sinistral

strike-slip to oblique-slip movements (vertical foliation and flat to inclined lineation; Fig. 2), particularly developed in the central and western part of the TW (e.g. Rosenberg et al., 2018; Schneider et al., 2013). These shear zones developed as a result of bending of the E-W oriented upright folds around a vertical axis (leading edge of the Dolomite indenter) (Fig. 1). This occurred when N-S shortening could no longer be accommodated by folding and doming within the TW. Associated with these movements is the formation of a younger generation of fissures (see Pfitscherjoch example above), the peak activity of which is recorded at ~17 Ma (phase (2), green symbols on figure 7). This fissure generation is associated with a steep foliation and a flat lineation (Fig. 2), but sub-parallel in orientation to the earlier fissure formation. The monazite ages at ~17 Ma found in the western and central TW (Figs. 1, 6 and 7; samples 5, 8, 13; Table 4) are in association with sinistral fault zones, as in the Pfitscherjoch region or near the eastern termination of the ASZ and AhSZ faults (see above). Unfortunately, we do not have structural information on the westernmost and easternmost analysed samples (6 and 25; Figs 1, 6 and 7), but they can be speculated to also have formed in association with a strike slip shear zone or the BNF and KNF in the case of samples 6 and 25 respectively. At larger scale, these movements seem to have been associated with the development of the sinistral Giudicarie Fault (GF, located at the southwestern corner of the TW), offsetting the Periadriatic Fault (PF; Fig. 8b, e.g., Pleuger et al., 2012). Ages of ~17 Ma are also recorded in the eastern part of the TW, likely linked to the KNF and Mölltal Fault (MöF) activity (samples 16, 21, 24 and 25; Fig. 7a and d; e.g. Favaro et al., 2017). In grains located in the western part of the eastern sub-dome, in the prolongation of the MöF (e.g. Kurz and Neubauer, 1996) (Fig. 1), numerous monazite growth domains yield ages between 15.6 ± 0.7 and 15.0 ± 0.5 Ma (bracketed by samples 22 and 21 from Gnos et al., 2015; Figs 1, 6c, 7a and d; green circles on figure 7d; Table 4). These ages date the latest known activity of this shear zone to ~15 Ma. Whereas younger ages, associated with reactivation of fault

zones are widespread in the central and western TW, tectonic movements seem to cease in the

eastern TW after this time (Fig. 8c).

The youngest monazite growth domain ages, principally recorded in the western sub-dome, range

from $13.2 \pm 0.3$ to $10.0 \pm 0.2$ Ma (samples 5 and 2; Table 4), indicate steps of reactivation of the

different sinistral strike-slip to oblique-slip movements along different faults (phase (3) and blue

symbols on figure 7). Based on our monazite crystallization data, the oldest activities of this

younger phase are recorded near the GSZ (sample 5) and in the prolongation of the AhSZ

(samples 7 and 9). The youngest activities are recorded in association with the ASZ, OSZ and

TSZ on figure 7a-c; samples 1, 2 and 4), and in the central TW in an area located south of the

main fault systems (sample 14, Figure 7a). In addition to faults activity at ~12 Ma (Fig. 8), coeval

strike-slip activity has also been documented in many areas of the central and western Alps (e.g.,

Bergemann et al., 2017, 2019; Berger et al., 2013; Gasquet et al., 2010; Grand'Homme et al.,

2016a; Pleuger et al., 2012; Ricchi et al., 2019).

In summary, in the western TW, monazite ages (Fig. 1) constrain the activities of the ASZ (18 –

12 Ma, samples 8 and 9), AhSZ (17 – 12 Ma, samples 13 and 7), TSZ/OSZ (11.5 – 10 Ma,

samples 1 and 2; older ages of sample 3 are probably related to extensional unroofing) and GSZ

(17 – 13 Ma). In the eastern part, the MöF is active between 19 and 15 Ma.

**5.2    Comparison with shear zone dating**

A number of attempts to date shear zone activity in the TW using Ar-Ar, Rb-Sr and Sm-Nd

techniques have been made in the past, which were, however, based on mineral separation

techniques without a clear structural control on the dated grains (e.g., Blanckenburg et al., 1989;

Glodny et al., 2008; Pollington and Baxter, 2010, 2011; Urbanek et al., 2002). An exception to

this is the $^{40}$Ar/$^{39}$Ar study of Schneider et al. (2013) on syn-kinematic phengite and K-feldspar

which will be used in the following as a comparison (Table 2). Fissure monazite ages largely

corroborate this work, similarly showing the longevity of different shear zones in the TW. The

ages confirm that even though most of the dated monazite samples are only located in the damage

zone in the vicinity of the core of the shear zones, fluid-filled fissures provide a sensitive system

where tectonic activity triggers fluid-enhanced dissolution/precipitation reactions at lower

greenschist to sub-greenschist facies conditions.

While Schneider et al. (2013) obtained crystallization age ranges of 33 – 15 Ma for the ASZ, 24 –

12 Ma for the TSZ and 20 – 7 Ma for the GSZ (Table 2), our data confirms fluid activity, and thus

possible tectonic activity, at 18 – 12, 11.5 – 10 and 17 – 13 Ma respectively (Fig. 7). However,

the oldest dates from Schneider et al. (2013) might also be interpreted as older grains that have

been aligned in the new foliation (Fig. 8). The data presented here indicate that all of the shear

zones where potentially active at least until ~13 – 12 Ma, and the Tuxer and/or Olperer shear

zones even until ~7 Ma as suggested by younger dates observed in grain 2 (Figs. 3b, 6a and 7,

Tables 3 and 4). However, the fissure monazite data does not date the initiation of the GSZ

(Selverstone et al., 1991), nor the earliest activity of the TSZ (greenschist to amphibolite facies;

Selverstone et al., 1984, 1991) or the ASZ (greenschist facies; Cole et al., 2007), since their

formation already started at amphibolite facies conditions. As Alpine fissures only form under

greenschist facies conditions, the oldest monazite crystallization ages are younger than the data

obtained by Schneider et al. (2013). This indicates that shear zone activity started earlier than the

fissure monazite record. As the monazite age range of the younger fault activity is comparable to

the data of Schneider et al. (2013), but is not the same for individual shear zones, it seems likely

that all shear zones of the western TW were active as recently as 8 - 7 Ma.

## 5.3 Comparison with fission track data

There is a wealth of zircon fission track (ZFT) data that can assist in describing the exhumation

and low grade tectonic activity in the TW (Bertrand, 2014; Bertrand et al., 2017; Dunkl et al.,

2003; Fügenschuh et al., 1997; Mancktelow et al., 2001; Most, 2003; Pomella et al., 2011;

Steenken et al., 2002; Stöckhert et al., 1999; Viola et al., 2001; Wölfler et al., 2008) and apatite

fission track (AFT) data (Bertrand, 2014; Bertrand et al., 2017; Coyle, 1994; Di Fiore, 2013;

Foeken et al., 2007; Fügenschuh et al., 1997; Grundmann and Morteani, 1985; Hejl, 1997;

Mancktelow et al., 2001; Most, 2003; Pomella et al., 2011; Staufenberg, 1987; Steenken et al.,

2002; Viola et al., 2001; Wölfler et al., 2008, 2012).

Three cross sections, DD' (perpendicular to the BNF), AA' (perpendicular to the western limb of

the western sub-dome) and EE' (parallel to the main axial plane of the TW) are presented in

figure 7, redrawn after Bertrand et al. (2017) and Schmid et al. (2013). Zircon and apatite fission

track data compiled by Bertrand et al. (2017) are displayed in the lower part of figure 7b-d and

compared to fissure monazite ages. As described in Bertrand et al. (2017) (first model), fission

tracks data along AA' cross section (Fig. 7c) nicely display a dome-like shape with younger ages

recorded near the sub-dome axial plane, were cooling was slower. By contrast, along the EE'

longitudinal cross section (Fig. 7d), ZFT and AFT are younger in the western and eastern border

of the TW where the two major extensional faults, the BNF and KNF, are respectively located.

Perpendicular to the BNF (DD' cross section, Fig. 7b), the fission tracks record cooling ages

younging from the footwall toward the plane of the normal fault (from 10 to 4 Ma for AFT;

second model of Bertrand et al. 2017). Along EE' cross section, the youngest monazite ages (15 –

10 Ma) lie between zircon and apatite fission track data (grey and blue symbols), whereas the

older ages (>17 Ma) do not follow the cooling trend and are equal to or older than the ZFT data.

This means that at least the fissure monazites recording older ages crystallized somewhere above

ZFT closure temperatures of ~240 – 280 °C (Bernet, 2009; Bernet and Garver, 2005; Reiners,

2005; Yamada et al., 1995) (Fig. 7d).

**5.4 Monazite Th/U as monitor of oxidizing and reducing conditions**

Extreme low and high Th/U ratios described in fissure monazite by Gnos et al. 2015 (T1, T2 and

T3 sample in figure 9) are also observed in some grains from this study (red and blue labels on

figure 9). Hydrothermal monazite from the TW associated with hematite in fissure typically

displays very high Th/U ratios of around 1200 (Fig. 9, red labels; Table 1), whereas grains

obtained from graphite-bearing host rocks show very low Th/U ratios around 8 (Fig. 9, blue

labels; Table 1). This attests for oxidizing and reducing fluid conditions in the fissure

environment, respectively.

The Th/U in monazite grains PFIT1 and MOKR1 would instead record a dynamic oxidation

environment due to variable fluid conditions. In PFIT1 monazite the Th/U decreases from core to

rim, whereas within MOKR1 the opposite evolution is observed (Fig. 9). Thus in the first case the

fissure environment evolves toward reducing conditions whereas in the second case there is an

evolution towards more oxidizing conditions. Many of the other grains indicate intermediate

oxidizing conditions and they could not be assigned to one of the two categories defined above,

as the presence of either hematite or graphite is uncertain (Fig. 9; grey labels).

**6**      **Conclusions**

Th-Pb ages of fissure monazite provides an extended record of exhumation of the TW during the

Miocene. The investigated monazites crystallized at temperatures <400°C in the presence of

hydrothermal fluids that circulated in open fissures formed through tectonic movements. The Th-

Pb ages recorded by fissure monazites are in general agreement with previously published

geochronological data and range between 21.7 ± 0.4 Ma to 10.0 ± 0.2 Ma. Spot dates suggests

that monazite crystallization in the metamorphic and structural TW dome occurred over a period

of ~16 Ma. The combination of structural and geochronological information allows relating

monazite growth with tectonic movements that affected the TW. The three major growth episodes

identified in this study, by dating monazite growth domains, are interpreted to be associated with

N-S shortening associated to E-W extension (22 – 20 Ma), contemporaneous N-S shortening and

sinistral strike-slip movements (19 – 15 Ma) and reactivation of strike-slips/normal faulting (14 –

10 Ma). Overall, fissure monazite age recording indicates that in the TW Cenozoic faults show

increased activity at ~21, ~17 and ~12 Ma, probably due to reorganization of plate movements

occurring at those times. Comparison of Th-Pb fissure monazite crystallization ages with existing

crystallization and cooling ages (e.g. AFT, ZFT, white mica from fault zones) show that the latest

stages of monazite crystallisation occurred at temperatures between apatite and zircon fission

track "closure" temperatures. This enlarged dataset also supports previous observations on fissure

monazites chemistry displaying extremely high Th/U ratios (~1200) under oxidizing conditions

in association with hematite.

*Data availability.* The data used in this study are available in Tables 3 and 4.

*Author contributions.* Fissure monazite samples were organized by EG and FW. Monazite for

dating were selected by ER, CAB, EG and AB according to tectonic settings and fault activity of

the study area. ER prepared the manuscript during her PhD project under the supervision of EG,

with contributions from all co-authors. Sample preparation and BSE imaging was performed by

ER and CAB. Data acquisition and reduction at the SwissSIMS and NordSim facility was

respectively carried out by ER and CAB under the supervision of DR and MJW.

*Competing interests*. The authors declare that they have no conflict of interest.

*Acknowledgments*. We thank Sepp Brugger, Kurt Novak, Franz Gartner, Peter Hellweger, Adolf

Meyer, Sebastian Plankensteiner, Johann Rappold, Josef Rathgeb, Alexandre Salzmann, Maria

Schaffhauser, Andreas Steiner and Ermin Welzl for having provided samples for this study. This

study was financed by the SNF grant number 200020-165513. Urs Klötzli and Jan Pleuger are

thanked for their helpful comments.

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

***Figures:***

***Fig. 1****: Tectonic map of the TW dome modified after Bertrand et al. (2017), Scharf et al. (2013), Schmid et*

*al. (2013) and Schneider et al. (2013). Yellow stars on the map represent samples locations and numbers*

*inside the stars refer to samples listed in Table 1. Range of weighted mean growth domain ages are*

*indicated for each grain from this study and Gnos et al. (2015), labelled in black and green respectively*

*on the map (see Table 4 for an exhaustive summary of all the ages). Only the spot date range is indicated*

*for grains 1, 4 and 6. Locations of AA', BB' and CC' cross sections are indicated by black lines and*

*individual cross sections are presented in figure 6 together with monazite crystallization ages. Two normal*

*faults delimit the western and eastern border of the TW, the Brenner Normal Fault (BNF) and the*

*Katschberg Normal Fault (KNF) respectively. Note that the KNF prolongation results in dextral and*

*sinistral strike-slips in the North and South respectively (KSZS: Katschberg Shear Zone System). Several*

*sinistral strike-slip faults (AhSZ: Ahrntal Shear Zone; ASZ: Ahorn Shear Zone; DAV: Defereggen-Antholz-*

*Vals Fault; GSZ: Greiner Shear Zone; InF: Inntal Fault; MüF: Mur-Mürz Fault; NF: Niedere Tauern*

*Southern Fault; OSZ: Olperer Shear Zone; SEMP: Salzach-Ennstal-Mariazell-Puchberg Fault; SpSZ:*

*Speikboden Shear Zone; TSZ: Tuxer Shear Zones; ZWD: Zwischenbergen-Wöllatratten and Drautal*

*Faults), dextral shear zones (HoF: Hochstuhl Fault; IsF: Iseltal Fault; KLT: Königsee-Lammertal-*

*Traunsee Fault; Mölltal Fault (MöF); PF: Pustertal Fault) and a reverse fault (MM: Meran-Maules*

*Fault) are also visible in red on the map.*

***Fig. 2****: a) Two generations of late fissures visible in a road outcrop located between monazite locality*

*(46°59.436'N/011°39.240'E) and Pfitscherjoch. Steeply oriented fissures (C₂: ~090/65) are older and*

*deformed (green ellipses), and seem related to a flatter lineation (L₂: ~250/30, green arrows) visible on*

*some of the foliation planes. Younger and flatter oriented fissures (C₃: ~085/30) are strait (blue ellipses),*

*and seem related to a steeper lineation (L₃: 270/70, blue arrows). These observations indicate that a*

*fissure can be deformed during its existence.  Lengths of hammer handle is 60 cm. b) Enlargement of a). c)*

*Schematic illustration of the 3 fissure generations observed in this study ($C_1$, $C_2$ and $C_3$), together with*

*respective orientation, foliation ($S_1$, $S_2$ and $S_3$) and lineation ($L_1$, $L_2$ and $L_3$). The first fissure generation*

*($C_1$) is related to E-W extension, the second fissure generation ($C_2$) is linked to strike-slip movements and*

*the third fissure generation ($C_3$) is related to the oblique-slip movements.*

***Fig. 3****: Chemical, textural and geochronological information of monazite grains from the western TW. On*

*BSE images, colour-filled circles correspond to ion probe spot locations. Note that the square-shape*

*shading in grain 4 is due to an artefact of composing BSE images with diverse contrast.*

***Fig. 4****: Chemical, textural and geochronological information of monazite grains from the central TW. On*

*BSE images, colour-filled circles correspond to ion probe spot locations. Note that the square-shape*

*shading in grains 10 and 11 is due to an artefact of composing BSE images with diverse contrast.*

***Fig. 5****: Chemical, textural and geochronological information of monazite grains from the eastern TW. On*

*BSE images, colour-filled circles correspond to ion probe spot locations. Note that the square-shape*

*shading in grains 15, 17, 18 and 20 is due to an artefact of composing BSE images with diverse contrast.*

***Fig. 6****: Cross sections of (a) the western, (b) the central part of the western sub-dome and (c) western end*

*of the eastern sub-dome, modified after (Schmid et al., 2013). See figure 1 for locations and legend.*

*Sample locations are indicated by yellow stars and identified by sample numbers listed in Table 1.*

*Monazite crystallization ages are present in the lower part of the figure and are linked to each sample by*

*light-grey dashed lines. Weighted mean ages from this study and from Gnos et al. (2015) are presented by*

*yellow diamonds and yellow circles, respectively, and blue bars correspond to the range of single spot*

*dates.*

***Fig. 7****: a) Map of the TW from figure 1 with sample locations colored as function of deformation episodes*

*(colored stars). See figure 1 for legend. b) DD' NE-SW cross section across the BNF, c) AA' NW-SE cross*

*section perpendicular to the axial plane of the western sub-dome and d) EE' longitudinal cross section*

*parallel to the main axial plane of the TW metamorphic dome, modified after Bertrand et al. (2017). In the*

*upper part, colored and numbered stars correspond to samples locations and are linked to corresponding Th-Pb monazite ages by dashed vertical lines. Sample numbers refer to Table 1. In the lower part, monazite weighted mean ages from this study and from Gnos et al. (2015) are labelled by colored diamond and circles, respectively, and the range of single spot dates is depicted by blue bars. The color code used for diamonds and circles follow deformation episodes explained in the discussion. Note that most error bars are smaller than the size of the diamonds and circles. Zircon and apatite fission track ages are from the Bertrand et al. (2017) compilation, light- and dark- grey dots with error bars, are displayed for comparison. Square brackets shown to the right delimit the main periods of monazite growth discussed in the text: (1) Early record of N-S shortening and associated E-W extension, (2) Contemporaneous N-S shortening and strike-slip, (3) Reactivation of strike-slip to oblique-slip.*

***Fig. 8****: Tectonic map of the Alps based on Pleuger et al. (2012) showing active Cenozoic faults at ~21 (in red), 17 (in green) and 12 Ma (in blue) respectively. Note that after 17 Ma the Giudicarie Fault (GF) becomes active and hence the Periadriatic Fault (PF) and the Mölltal Fault (MöF, dextral fault at the southeastern corner of the TW) become inactive. Sinistral strike-slip faulting starts at ~19 Ma and is affecting the western and central parts of the TW until at least 8 Ma. Future active faults are depicted in grey and inactive faults in black.*

***Fig. 9****: Th as function of U content obtained for all the monazite grains analysed in this study. Samples indicated by an asterisk are from Gnos et al. (2015). Fissure monazite grains associated to hematite (oxidizing conditions) are labelled in red whereas grains hosted in graphite bearing rocks (reducing conditions) are labelled in blue. Samples with intermediate composition and/or for which we have no information on the presence of hematite or graphite in the fissure environment are labelled in grey.*

***Tables:***

***Table 1****: Summary of monazite samples investigated in this study and from Gnos et al. (2015). Samples*

*name, number, location, host-rock lithology, metamorphic degree and fissure mineral association are*

*provided. Samples with approximate finding location are marked with "approx.".*

***Table 2****: Summary of deformation phases in the Tauern metamorphic dome.*

***Table 3****: Th-U-Pb analyses of monazite by ion microprobe (SwissSIMS and Nordsim). Analyses resulting*

*in unreliable dates (e.g. presence of cracks, affected by Pbc causing high uncertainty) were not considered*

*and are written in italic.*

***Table 4****: Summary of weighted mean ages of monazite growth domains and spot age ranges of each grain*

*from the TW.*

*Figure1.*

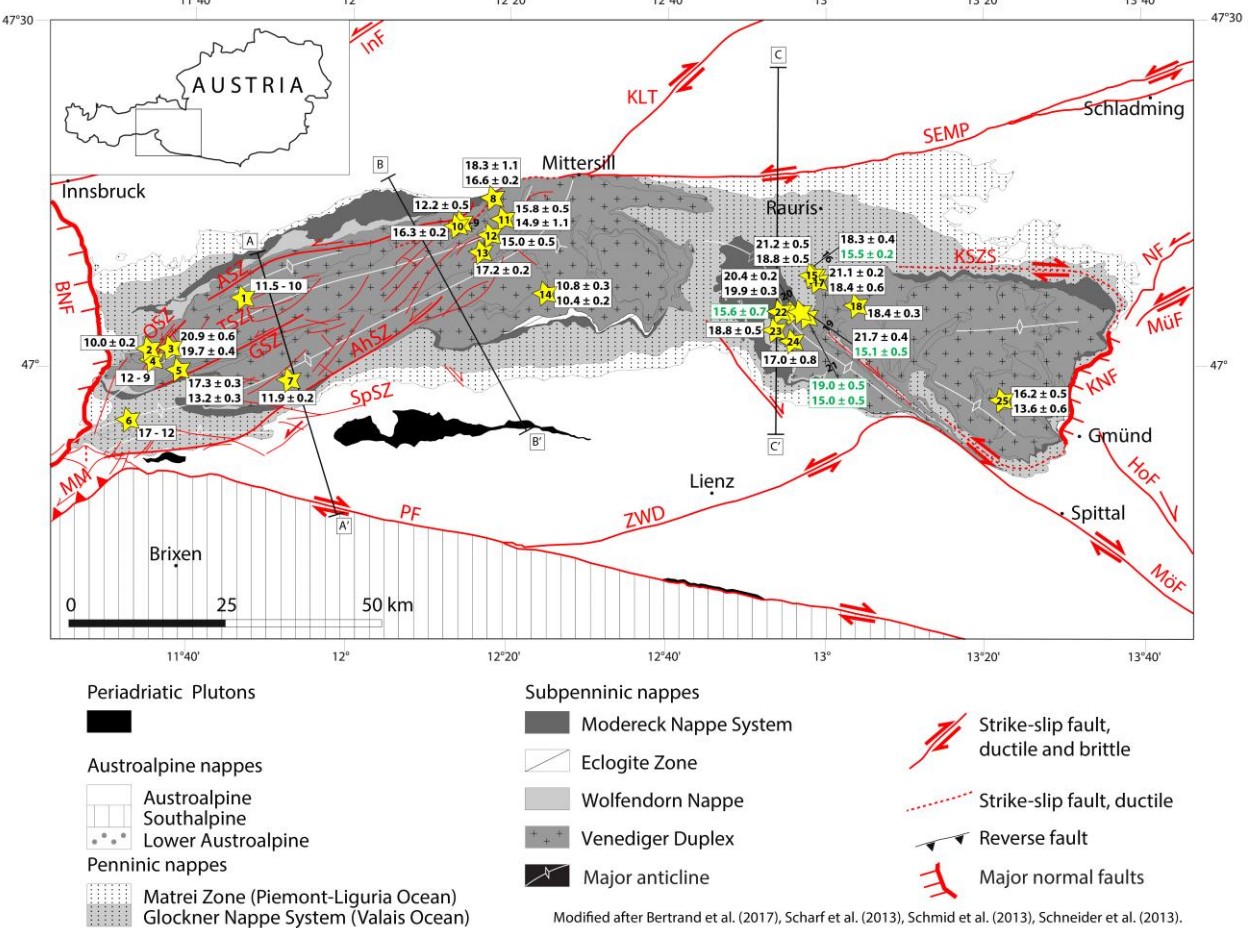

*Figure 2.*

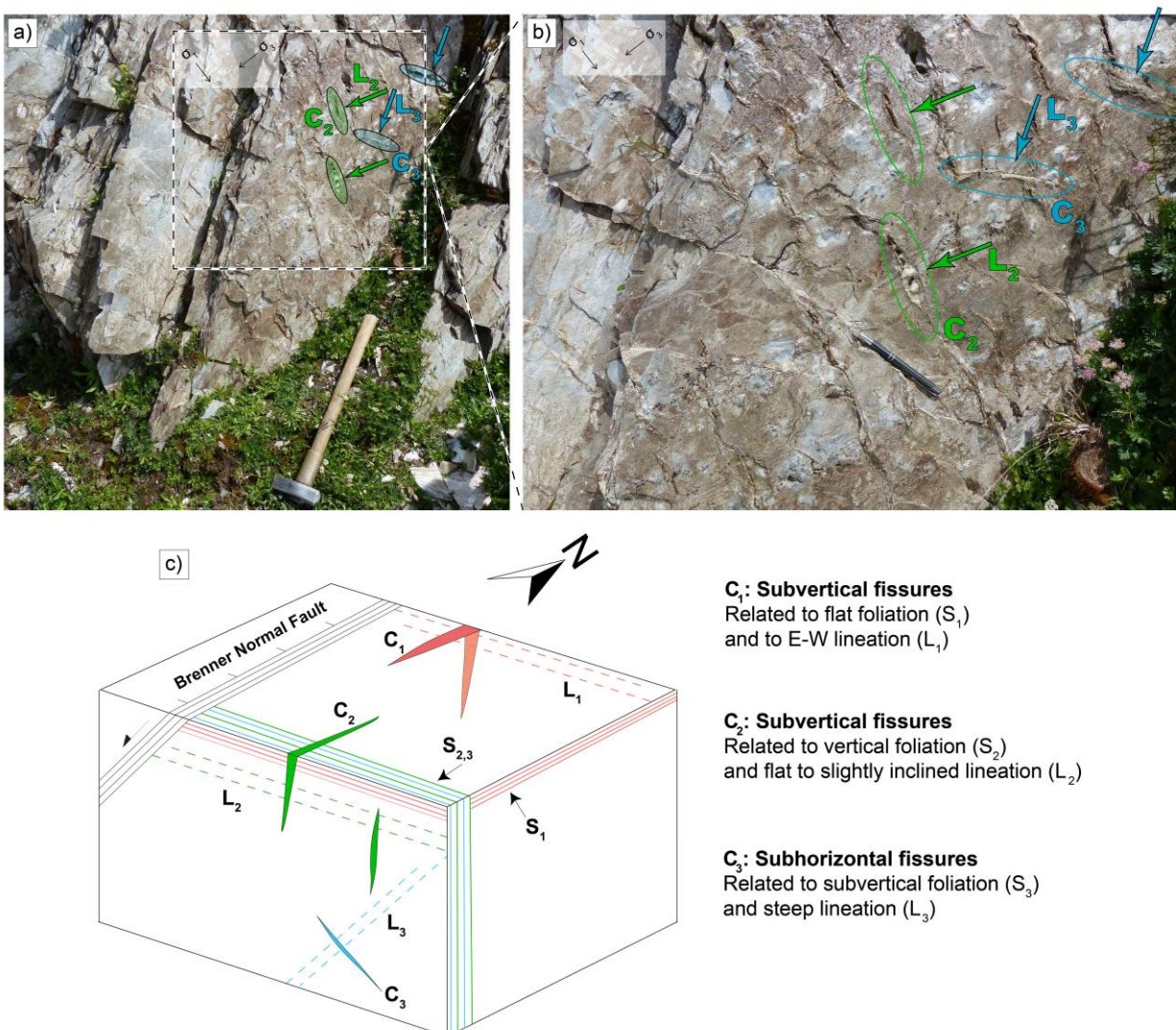

**C₁: Subvertical fissures**
Related to flat foliation (S₁)
and to E-W lineation (L₁)

**C₂: Subvertical fissures**
Related to vertical foliation (S₂)
and flat to slightly inclined lineation (L₂)

**C₃: Subhorizontal fissures**
Related to subvertical foliation (S₃)
and steep lineation (L₃)

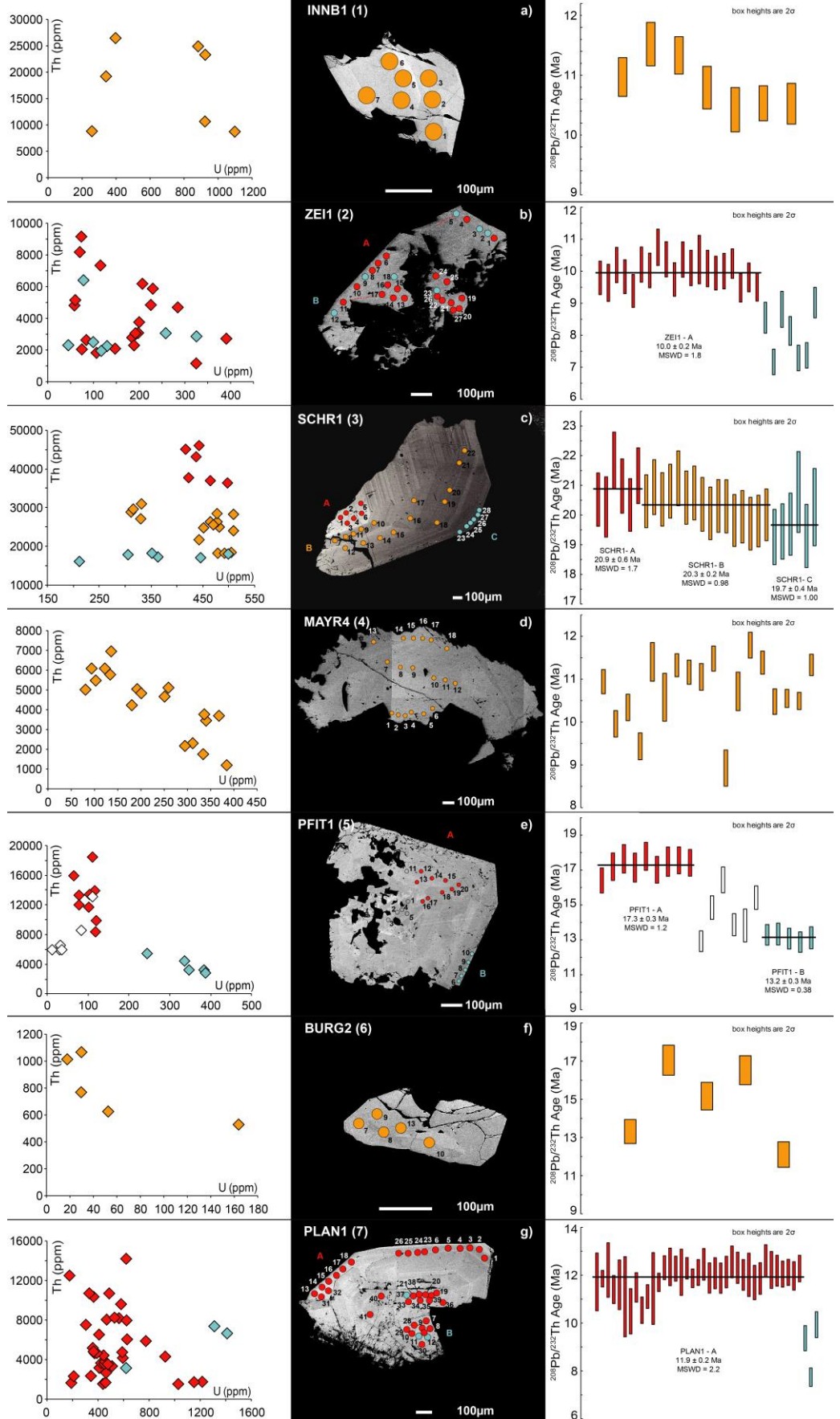

*Figure 3.*

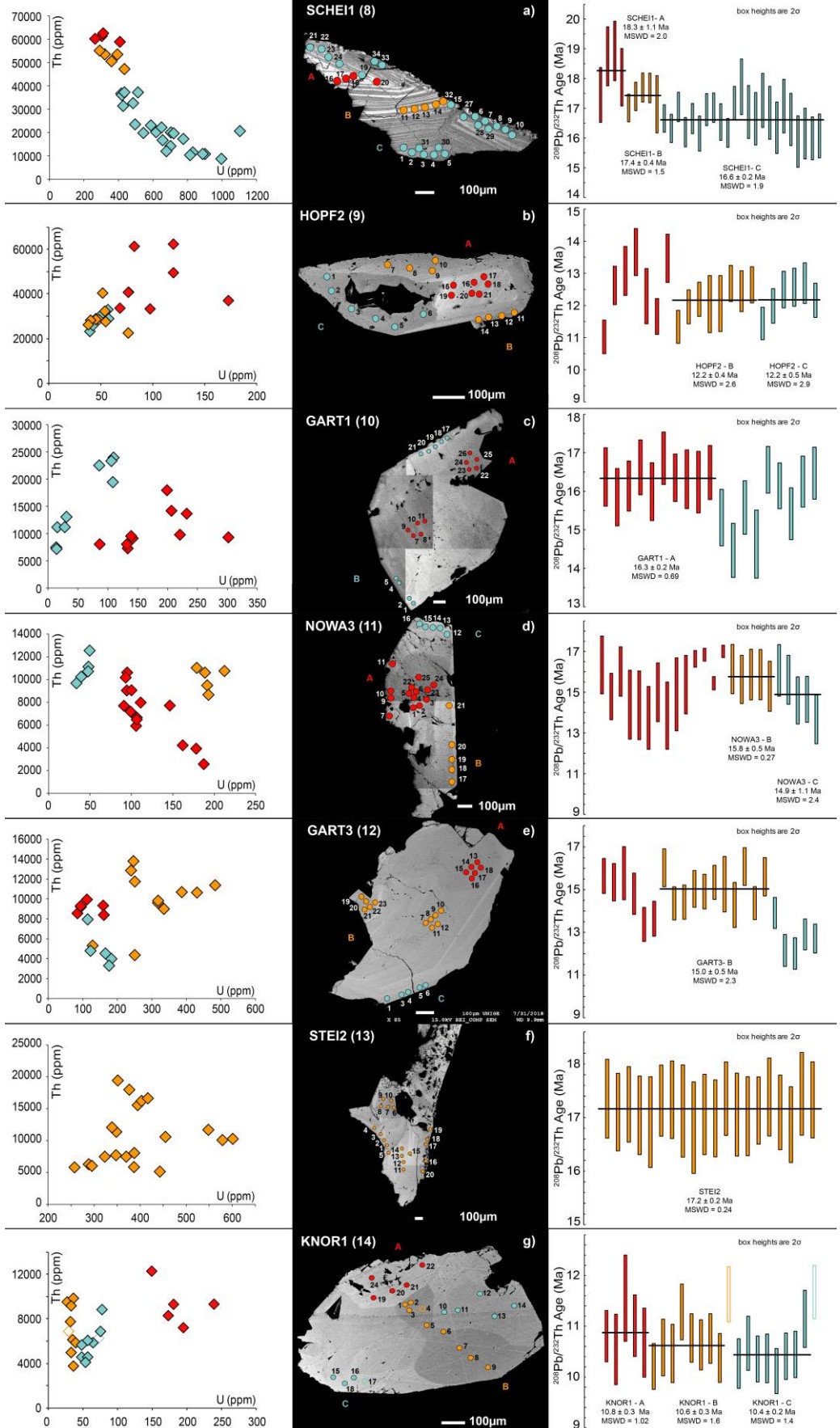

*Figure 4.*

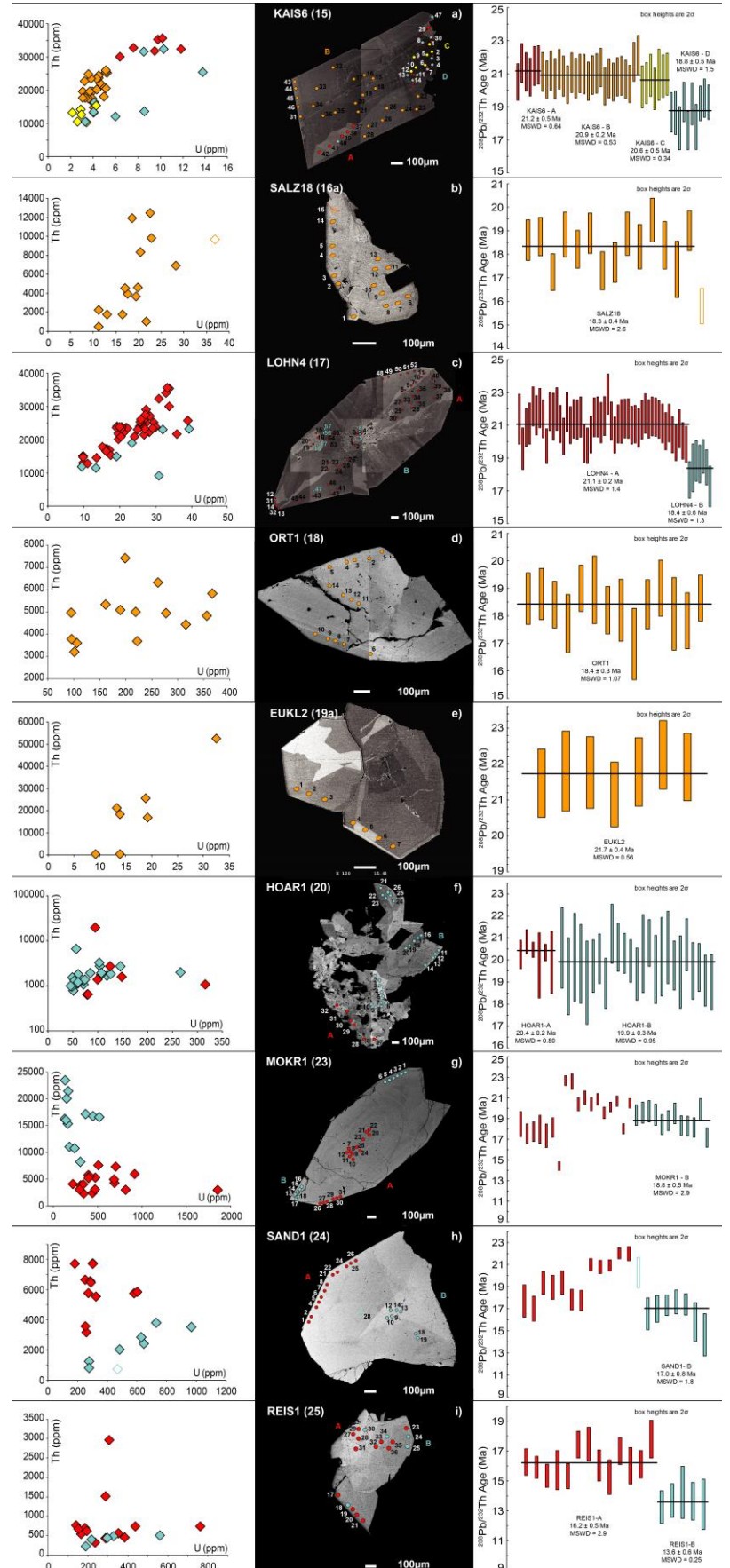

*Figure 5.*

*Figure 6.*

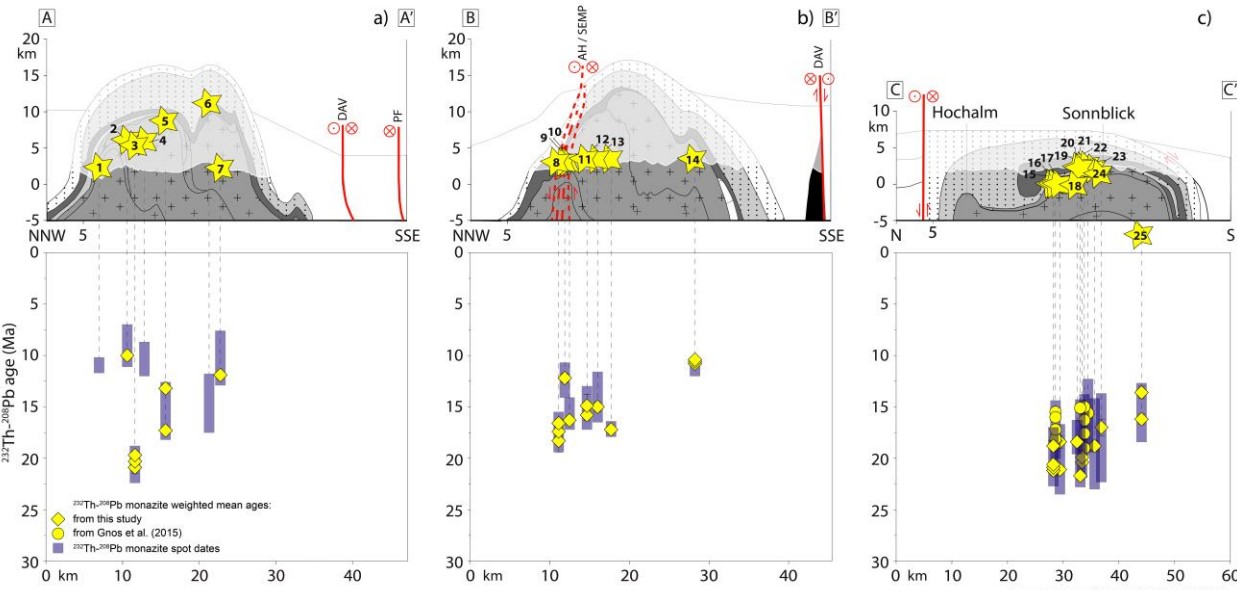

Cross sections modified after Schmid et al. (2013).

*Figure 7.*

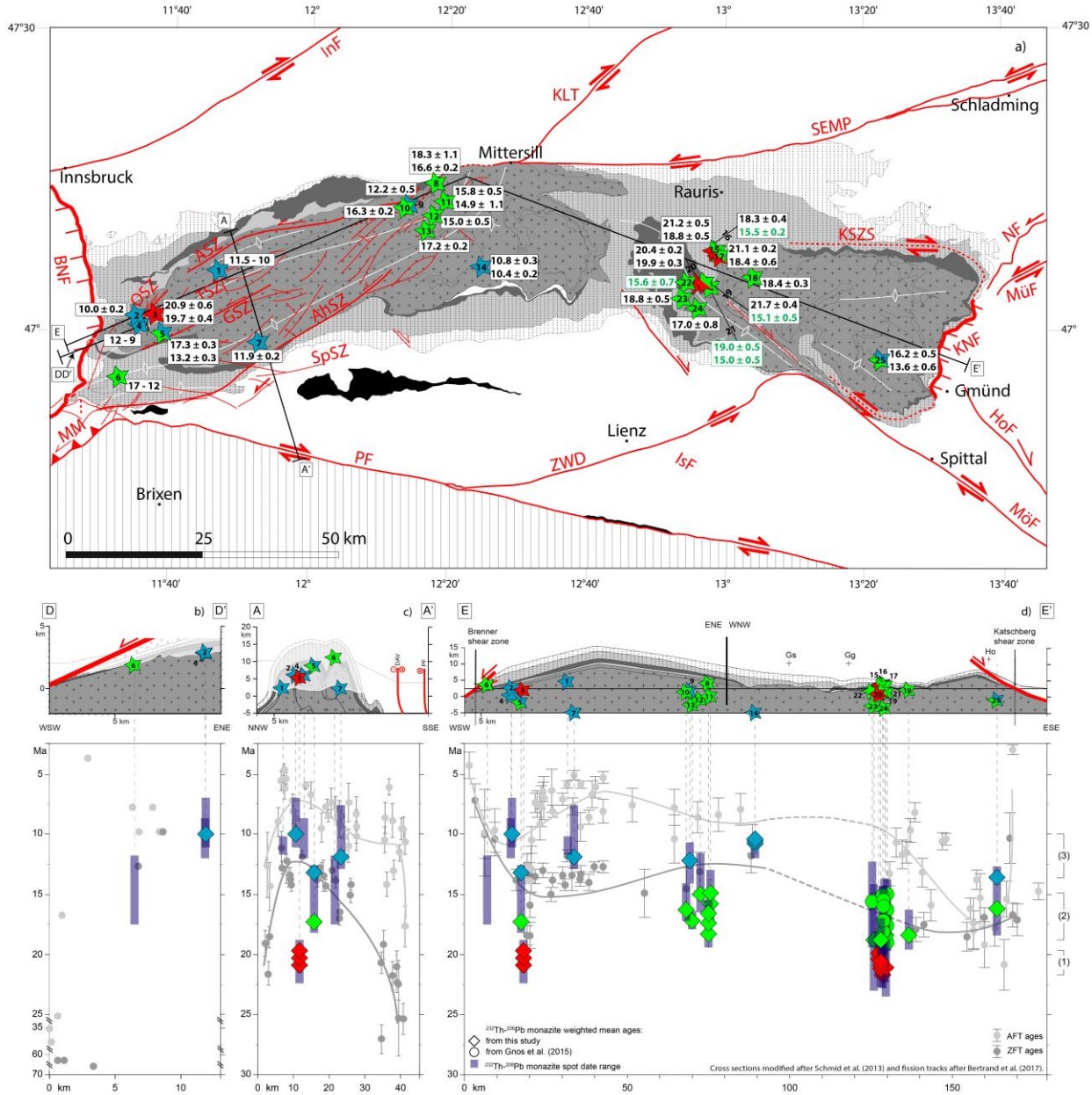

(1) Early record of N-S shortening and associated E-W extension; (2) Contemporaneous N-S shortening and strike-slip; (3) Reactivation of strike-slip to oblique-slip

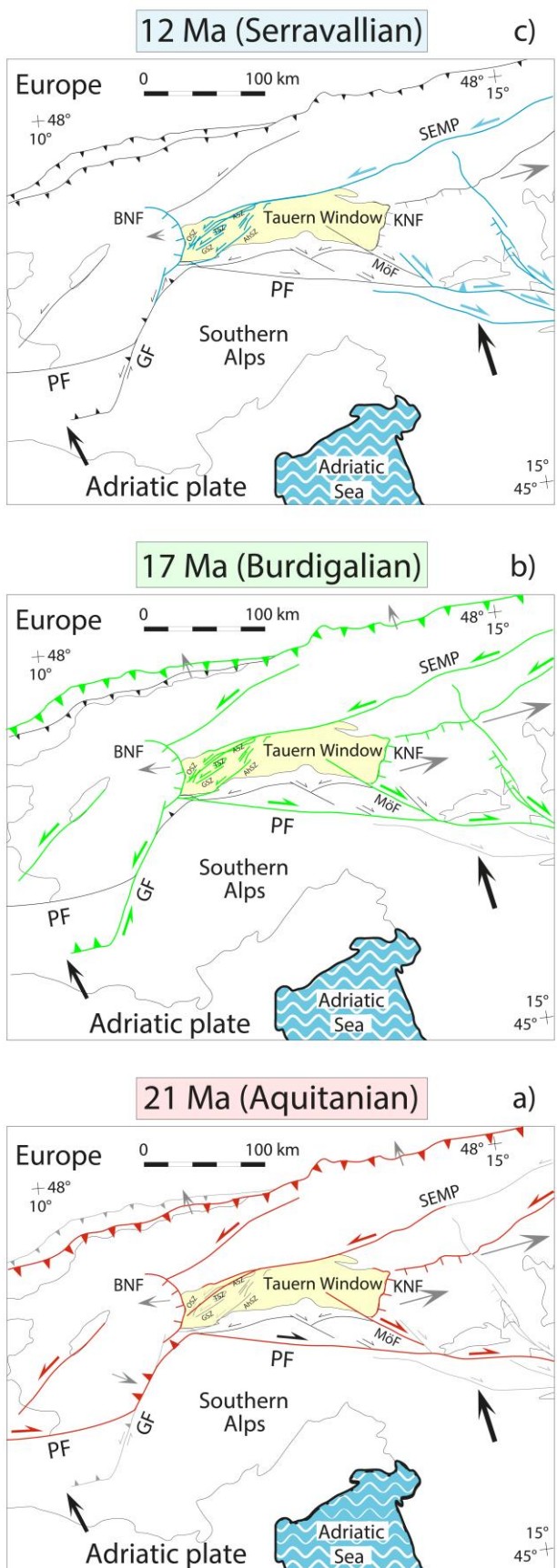

*Figure 8.*  Tectonic map of the Alps modified after Pleuger et al. (2012)

*Figure 9.*

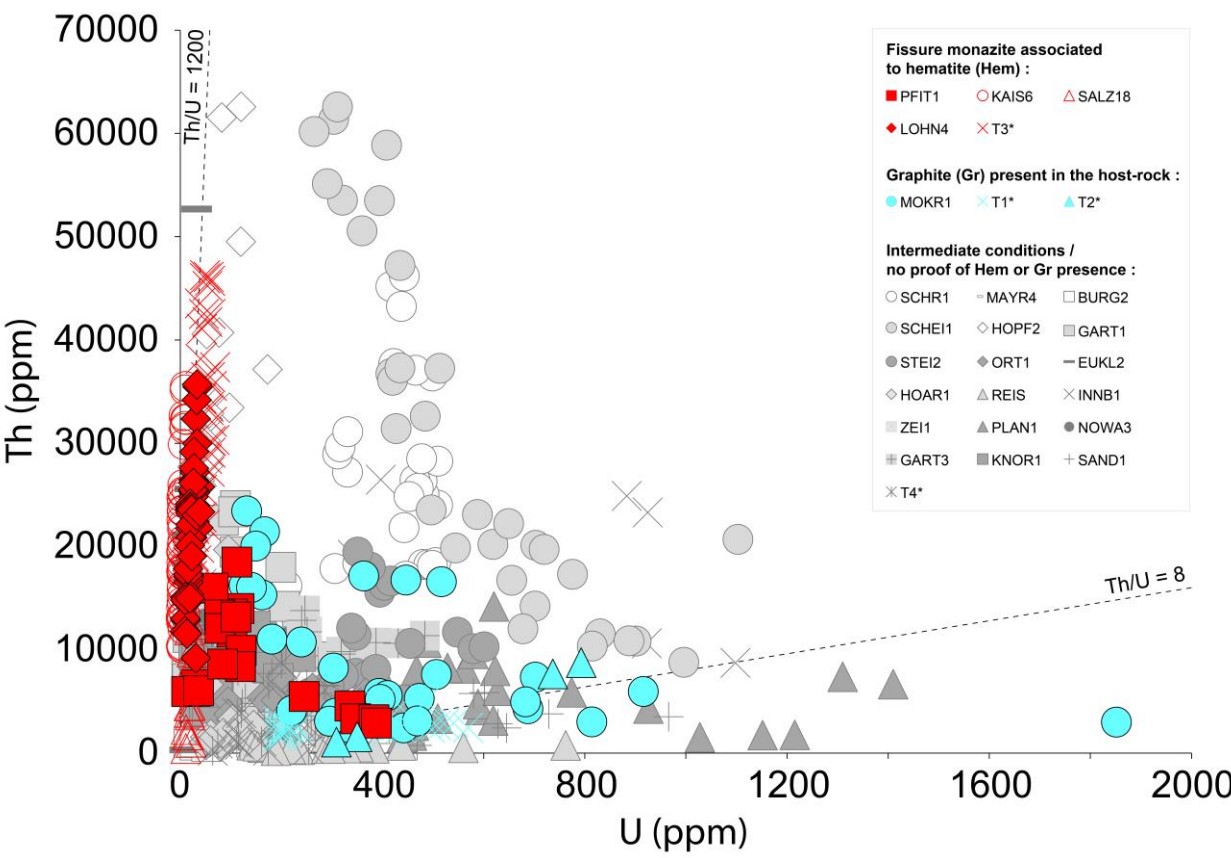

Table 1. Summary of monazite samples investigated in this study and by Gnos et al. (2015). Samples name, number, location, host-rock lithology, metamorphic degree and fissure mineral association are provided. Samples with approximate finding location are marked with "approx.".

| Locality | Sample | Nº | Latitude (°N) | Longitude (°E) | Remark | Host-rock | Host rock Alpine met. | Fissure mineral association | Reference | Ion probe |
|---|---|---|---|---|---|---|---|---|---|---|
| *Western Tauern Window* | | | | | | | | | | |
| Innerböden, Zillertal, Tirol, Austria | INNB1 | 1 | 47°05.850' | 011°47.667' | approx. | gneiss | AM | Qtz, Adl, Chl | this study | SwissSIMS |
| Zeischalm, Valsertal, Tirol, Austria | ZEI1 | 2 | 47°01.400' | 011°35.767' | approx. | gneiss | AM | Qtz, Adl, Chl | this study | SwissSIMS |
| Schrammacher, Zillertal, Tirol, Italy | SCHR1 | 3 | 47°01.47' | 011°38.60' | approx. | gneiss | AM | Qtz, Chl, Adl, Rt, Snt | this study | NordSim |
| Kluppen, Pfitschtal, Südtirol, Italy | MAYR4 | 4 | 47°00.400' | 011°36.217' | approx. | gneiss | AM | Qtz, Adl, Ant, Cc, Rt, Xnt | this study | SwissSIMS |
| Pfitscherjoch, Tirol, Austria | PFIT1 | 5 | 46°59.65' | 011°39.60' | | mica schist | AM | Qtz, Hm, Adl, Trm, Rt, Brk, Ant, Syn, Asc | this study | NordSim |
| § Burgumalpe, Pfitschtal, Südtirol, Italy | BURG2 | 6 | 46°55.217' | 011°33.350' | approx. | serpentinite | GAT | Ilm, Rt, Cc | this study | SwissSIMS |
| Schwarzenbach, Ahrntal, Südtirol, Italy | PLAN1 | 7 | 46°58.641' | 011°53.302' | approx. | gneiss | AM | Qtz, Adl | this study | SwissSIMS |
| *Central Tauern Window* | | | | | | | | | | |
| Scheissgraben (Kotriesen), Habachtal, Salzburg, Austria | SCHEI1 | 8 | 47°14.483' | 012°18.667' | | mica schist | UGS | Qtz, Ab, Ant, Ank, Rt | this study | SwissSIMS |
| Hopffeldboden, Oberzulzbachtal, Salzburg, Austria | HOPF2 | 9 | 47°12.278' | 012°14.844' | | gneiss | GAT | Qtz, Ilm, Rt, Ant, Brk, Asc, Syn, Ap, Chl, Lm | this study | NordSim |
| Hopffeldgraben, Oberzulzbachtal, Salzburg, Austria | GART1 | 10 | 47°12.100' | 012°14.167' | | gneiss | GAT | Qtz, Adl, Ant, Rtl, Asc + Bt and Chl from EDS analyses | this study | SwissSIMS |
| Wildenkarer-Wald, Habachtal, Salzburg, Austria | NOWA3 | 11 | 47°12.633' | 012°20.000' | approx. | gneiss | GAT | Qtz, Ab, Adl | this study | SwissSIMS |
| Beryller, Untersulzbachtal, Salzburg, Austria | GART3 | 12 | 47°11.250' | 012°18.517' | approx. | gneiss | AM | Qtz, Adl | this study | SwissSIMS |
| Sattelkar, Obersulzbachtal, Salzburg, Austria | STEI2 | 13 | 47°09.783' | 012°17.250' | approx. | gneiss | AM | Qtz, Adl, Rt, Chl | this study | SwissSIMS |
| Innerer Knorrkogel, Osttirol, Austria | KNOR1 | 14 | 47°06.117' | 012°25.183' | | gneiss | AM | Adl, Qtz, Chl | this study | NordSim |
| *Eastern Tauern Window* | | | | | | | | | | |
| * Kaiserer Steinbruch,Hüttwinkeltal, Rauris, Salzburg, Austria | KAIS6 | 15 | 47°07.787' | 012°58.708' | | meta-arenite | GAT | Qtz, Adl, Trm, Cc, Hm, Rt, Chl | this study | NordSim |
| * Lohninger Quarry, Hüttwinkeltal, Rauris, Salzburg, Austria | SALZ18 | 16a | 47°07.20' | 012°59.33' | | meta-arenite | GAT | Qtz, Adl, Trm, Cc, Hm, Rt, Chl | this study | NordSim |
| *\* Lohninger Quarry, Hüttwinkeltal, Rauris, Salzburg, Austria* | *T3* | *16b* | *47°07.20'* | *012°59.33'* | | *meta-arenite* | *GAT* | *Alb, Qtz, Trm, Hm, Rt* | *Gnos et al., 2015* | *NordSim* |
| * Lohninger Quarry, Hüttwinkeltal, Rauris, Salzburg, Austria | LOHN4 | 17 | 47°07.20' | 012°59.33' | | meta-arenite | GAT | Qtz, Adl, Trm, Cc, Hm, Rt, Chl | this study | NordSim |
| Ortberg bei Böckstein, Salzburg, Austria | ORT1 | 18 | 47°05.150' | 013°04.217' | | granitic gneiss | GAT | Qtz, Adl, Rt, Chl, Ap | this study | SwissSIMS |
| Euklaskluft, Griesswies, Salzburg, Austria | EUKL2 | 19a | 47°04.683' | 012°57.250' | | Bt-Mu schist | GAT | Alb, Pyr, Qtz, Rt, Chl, euclase, Xnt, goethite-todorokite-nordstrandite | this study | NordSim |
| *Euklaskluft, Griesswies, Salzburg, Austria* | *T2* | *19b* | *47°04.683'* | *012°57.250'* | | *Bt-Mu schist* | *GAT* | *Alb, Pyr, Qtz, Rt, Chl, euclase, Xnt, goethite-todorokite-nordstrandite* | *Gnos et al., 2015* | *NordSim* |
| Hocharn, Kärnten, Austria | HOAR1 | 20 | 47°04.500' | 012°56.083' | | granitic gneiss | GAT | Qtz, Ab, Rt | this study | SwissSIMS |
| *Erfurter Steig, Rauris, Salzburg* | *T1* | *21* | *47°04.133'* | *012°57.917'* | | *Bt-Mu schist* | *GAT* | *Qtz, Ab, Adl, phenakite, Rt, Cc* | *Gnos et al., 2015* | *NordSim* |
| *Gjaidtroghöhe, Grosses Fleisstal, Kärnten* | *T4* | *22* | *47°03.783'* | *012°54.650'* | | *gneiss* | *GAT* | *Qtz, Ab, Ant/Rt,Chl, Cc* | *Gnos et al., 2015* | *NordSim* |
| Mokritzen, Kleines Fleisstal, Kärnten, Austria | MOKR1 | 23 | 47°03.033' | 012°53.983' | | graphite bearing schist | GAT | Qtz, Sid | this study | SwissSIMS |
| Sandkopf, Grosses Zirknitztal, Kärnten, Austria | SAND1 | 24 | 47°01.983' | 012°56.05 | | banded gneiss | GAT | Qtz, Adl, Chl, Sid, Ant | this study | SwissSIMS |
| Kleiner Reisseck, Reisseckgruppe, Kärnten, Austria | REIS1 | 25 | 46°56.950' | 013°22.433' | | banded gneiss | AM | Qtz, Ant, Ab, Chl | this study | SwissSIMS |

Ab = albite; Adl =adularia; Ank = ankerite; Ant = anatase; Ap = fluorapatite; Asc = Aeschynite; Brk = brookite; Cc =calcite; Chl =chlorite; Clc =clinochlore; Hm = hematite; Ilm = ilmenite; Lm =limonite; Pyr = pyrite; Qtz = quartz; Rt = rutile; Sd = Siderite; Snt = senaite; Str = strontianite; Syn = synchisite; Trm = tourmaline; Xnt = xenotime; Alpine metamorphism: AM (Amphibolite facies), GAT (Greenschist-amphibolite transition), UGS (Upper greenschist facies). Bloc in glacial moraine : §. Bloc in rock slide : *.

Table 2. Summary of deformation phases in the Tauern metamorphic dome.

| Age [Ma] | Phase | Fault | Domain | Characteristics | Ref. | Remarks |
|---|---|---|---|---|---|---|
| **Estimated peaks of deformation** | | | | | | |
| ~ 65 | **D1** | | Penninic nappes | Accretion and subduction of Piemont-Liguria Ocean | E | |
| ~ 41 | **D2** | | Penninic and Subpenninic nappes | Subduction of Valais Ocean and parts of the distal European margin | E | |
| ~ 35 | **D3** | | Central TW | Exhumation of high-pressure units | E | Folding of D2 thrust, decompression |
| ~ 29 | **D4** | | Subpenninic nappes | European slab break off, Venediger Duplex formation and "Tauernkristallisation" | E | Contemporaneous intrusion of Periadriatic plutons and incipient NE-wards subduction of the Adriatic slab |
| ~ 23 - 21 | | | East of the Giudicarie Belt | Incipient indentation of the Southalpine Units in the Eastern Alps | D, E | |
| ~ 17 | **D5** | | TW | Indentation, doming and lateral extrusion | E | |
| **Faults motion** | | | | | | |
| 33 - 15 | | ASZ | Western TW | Sinistral ductile shear | F, G | Ductile continuation of the SEMP fault |
| 24 - 12 | | TSZ | Western TW | Sinistral ductile shear | B, F | |
| 20 - 7 | | GSZ | Western TW | Sinistral ductile shear | F | |
| 21 - 10 | | BNF | Western TW | Normal fault | C | |
| 22 - 13 | | KNF | Eastern TW | Normal fault | C | |

A: Bertrand et al., 2017, 2015; B: Blanckenburg et al., 1989; C: Favaro et al., 2017; D: Scharf et al. 2013; E: Schmid et al., 2013; F: Schneider et al., 2013; G: Rosenberg and Schneider, 2008. ASZ: Ahorn Shear Zone, BNF: Brenner Normal Fault, GSZ: Greiner Shear Zone, KNF: Katschberg Normal Fault, SEMP: Salzach-Ennstal-Mariazell-Puchberg fault, TSZ: Tuxer Shear Zones.

Table 3: Th-U-Pb analyses of monazite by ion microprobe (SwissSIMS and Nordsim). Analyses resulting in unreliable dates (e.g. presence of cracks, affected by Pbc causing high uncertainty) were not considered and are written in italic.

| | | | | | | | | | | | | 204-corr | | 204-corr spot ages | | 207-corr | | 207-corr spot ages | |
| --- | --- | --- | --- | --- | --- | --- | --- | --- | --- | --- | --- | --- | --- | --- | --- | --- | --- | --- | --- |
| Groups | Analysis ID | U (ppm) | Th (ppm) | Th/U | $^{208}Pb/^{204}Pb$ | 1σ (%) | $^{207}Pb/^{206}Pb$ | 1σ (%) | $^{208}Pb/^{232}Th$ | 1σ (%) | f208 from 207 (%) | $^{208}Pb/^{232}Th$ | 1σ (%) | $^{208}Pb/^{232}Th$ Age (Ma) | 1σ (abs.) | $^{208}Pb/^{232}Th$ | 1σ (%) | $^{208}Pb/^{232}Th$ Age (Ma) | 1σ (abs.) |
| *Western Tauern Window* | | | | | | | | | | | | | | | | | | | |
| | **INNB1@01** | 258 | 8810 | 34 | 180 | 11 | 0.359 | 3.0 | 0.000621 | 1.4 | 13 | 0.000514 | 2.7 | 10.39 | 0.29 | 0.000543 | 1.5 | **10.97** | **0.16** |
| | **INNB1@02** | 397 | 26461 | 67 | 497 | 10 | 0.253 | 2.8 | 0.000592 | 1.6 | 4 | 0.000560 | 1.7 | 11.32 | 0.19 | 0.000570 | 1.6 | **11.52** | **0.18** |
| | **INNB1@03** | 341 | 19205 | 56 | 421 | 11 | 0.280 | 3.0 | 0.000589 | 1.4 | 5 | 0.000547 | 1.6 | 11.05 | 0.18 | 0.000561 | 1.4 | **11.34** | **0.16** |
| | **INNB1@04** | 1098 | 8717 | 8 | 202 | 12 | 0.158 | 2.7 | 0.000607 | 1.6 | 12 | 0.000536 | 2.8 | 10.83 | 0.31 | 0.000534 | 1.6 | **10.80** | **0.18** |
| | **INNB1@05** | 924 | 23297 | 25 | 421 | 10 | 0.216 | 2.3 | 0.000552 | 1.8 | 7 | 0.000518 | 1.9 | 10.47 | 0.20 | 0.000516 | 1.8 | **10.43** | **0.18** |
| | **INNB1@06** | 883 | 24902 | 28 | 364 | 9 | 0.215 | 2.3 | 0.000557 | 1.4 | 6 | 0.000514 | 1.7 | 10.39 | 0.17 | 0.000521 | 1.4 | **10.54** | **0.15** |
| | **INNB1@07** | 923 | 10635 | 12 | 200 | 11 | 0.194 | 2.6 | 0.000594 | 1.6 | 12 | 0.000508 | 2.6 | 10.26 | 0.27 | 0.000521 | 1.6 | **10.53** | **0.17** |
| | *INNB1@08* | *1217* | *15622* | *13* | *283* | *13* | *0.228* | *2.7* | *0.000466* | *5.1* | *13* | *0.000433* | *5.1* | *8.74* | *0.45* | *0.000406* | *5.1* | *8.21* | *0.42* |
| | **ZEI1@01** | 84 | 2630 | 31 | 393 | 21 | 0.202 | 5.2 | 0.000515 | 2.6 | 6 | 0.000465 | 3.1 | 9.39 | 0.29 | 0.000485 | 2.7 | **9.80** | **0.26** |
| | **ZEI1@04** | 75 | 2041 | 27 | 408 | 25 | 0.196 | 5.8 | 0.000509 | 3.0 | 6 | 0.000475 | 3.8 | 9.60 | 0.36 | 0.000477 | 3.0 | **9.65** | **0.29** |
| | **ZEI1@06** | 147 | 2086 | 14 | 251 | 18 | 0.185 | 4.2 | 0.000558 | 2.6 | 9 | 0.000485 | 3.6 | 9.79 | 0.36 | 0.000505 | 2.7 | **10.20** | **0.28** |
| | **ZEI1@07** | 185 | 2785 | 15 | 547 | 24 | 0.159 | 4.4 | 0.000524 | 2.6 | 7 | 0.000503 | 3.2 | 10.16 | 0.32 | 0.000487 | 2.7 | **9.84** | **0.26** |
| | **ZEI1@08** | 189 | 2292 | 12 | 316 | 21 | 0.185 | 4.2 | 0.000519 | 2.7 | 10 | 0.000456 | 3.5 | 9.21 | 0.32 | 0.000465 | 2.7 | **9.39** | **0.26** |
| | **ZEI1@10** | 198 | 3078 | 16 | 421 | 20 | 0.143 | 4.0 | 0.000542 | 2.7 | 7 | 0.000493 | 3.0 | 9.96 | 0.30 | 0.000505 | 2.7 | **10.21** | **0.27** |
| | **ZEI1@11** | 107 | 1820 | 17 | 298 | 20 | 0.180 | 4.6 | 0.000546 | 2.6 | 9 | 0.000475 | 3.5 | 9.60 | 0.34 | 0.000496 | 2.7 | **10.03** | **0.27** |
| | **ZEI1@13** | 230 | 5870 | 26 | 559 | 15 | 0.163 | 3.2 | 0.000558 | 2.6 | 5 | 0.000519 | 2.6 | 10.49 | 0.28 | 0.000532 | 2.6 | **10.75** | **0.28** |
| | **ZEI1@14** | 226 | 4846 | 21 | 532 | 18 | 0.153 | 3.8 | 0.000542 | 2.6 | 5 | 0.000509 | 2.8 | 10.29 | 0.29 | 0.000514 | 2.6 | **10.38** | **0.27** |
| A | **ZEI1@15** | 284 | 4694 | 16 | 412 | 16 | 0.204 | 3.4 | 0.000523 | 2.7 | 8 | 0.000474 | 2.8 | 9.58 | 0.27 | 0.000482 | 2.7 | **9.75** | **0.26** |
| | **ZEI1@16** | 192 | 3057 | 16 | 422 | 18 | 0.158 | 3.7 | 0.000553 | 2.6 | 7 | 0.000506 | 2.9 | 10.23 | 0.30 | 0.000513 | 2.7 | **10.37** | **0.28** |
| | **ZEI1@17** | 200 | 3756 | 19 | 282 | 14 | 0.234 | 3.1 | 0.000561 | 2.7 | 11 | 0.000484 | 3.0 | 9.79 | 0.30 | 0.000500 | 2.7 | **10.11** | **0.28** |
| | **ZEI1@19** | 325 | 1152 | 4 | 188 | 22 | 0.114 | 3.8 | 0.000622 | 3.3 | 18 | 0.000495 | 5.2 | 10.00 | 0.52 | 0.000513 | 3.6 | **10.37** | **0.38** |
| | **ZEI1@20** | 391 | 2715 | 7 | 485 | 27 | 0.094 | 4.6 | 0.000533 | 2.8 | 6 | 0.000491 | 3.3 | 9.91 | 0.33 | 0.000499 | 2.8 | **10.09** | **0.28** |

| Group | Sample | | | | | | | | | | | | | | | | | | |
|---|---|---|---|---|---|---|---|---|---|---|---|---|---|---|---|---|---|---|---|
| | ZEI1@21 | 115 | 7335 | 64 | 555 | 15 | 0.219 | 4.3 | 0.000509 | 2.6 | 3 | 0.000476 | 2.7 | 9.62 | 0.26 | 0.000492 | 2.6 | **9.95** | **0.26** |
| | ZEI1@22 | 74 | 9153 | 124 | 780 | 15 | 0.281 | 4.0 | 0.000509 | 2.8 | 3 | 0.000484 | 2.7 | 9.78 | 0.27 | 0.000495 | 2.8 | **10.01** | **0.28** |
| | ZEI1@24 | 58 | 4809 | 82 | 617 | 19 | 0.235 | 4.7 | 0.000523 | 2.2 | 3 | 0.000491 | 2.4 | 9.91 | 0.24 | 0.000507 | 2.2 | **10.24** | **0.23** |
| | ZEI1@25 | 60 | 5152 | 86 | 670 | 21 | 0.230 | 4.8 | 0.000485 | 2.3 | 3 | 0.000459 | 2.5 | 9.28 | 0.23 | 0.000470 | 2.3 | **9.49** | **0.22** |
| | ZEI1@26 | 70 | 8182 | 117 | 609 | 18 | 0.255 | 4.8 | 0.000499 | 2.3 | 3 | 0.000468 | 2.4 | 9.45 | 0.23 | 0.000486 | 2.3 | **9.81** | **0.22** |
| | ZEI1@27 | 207 | 6179 | 30 | 565 | 21 | 0.109 | 4.4 | 0.000485 | 2.3 | 3 | 0.000455 | 2.6 | 9.19 | 0.24 | 0.000471 | 2.3 | **9.52** | **0.22** |
| | ZEI1@02 | 99 | 2499 | 25 | 352 | 23 | 0.194 | 6.0 | 0.000451 | 2.8 | 6 | 0.000402 | 3.6 | 8.12 | 0.29 | 0.000423 | 2.8 | **8.56** | **0.24** |
| | ZEI1@03 | 130 | 2246 | 17 | 663 | 41 | 0.177 | 7.7 | 0.000378 | 2.7 | 6 | 0.000356 | 3.5 | 7.19 | 0.25 | 0.000355 | 2.8 | **7.18** | **0.20** |
| | ZEI1@05 | 45 | 2301 | 51 | 447 | 26 | 0.286 | 5.9 | 0.000468 | 3.1 | 7 | 0.000434 | 3.7 | 8.78 | 0.33 | 0.000436 | 3.2 | **8.82** | **0.28** |
| B | ZEI1@09 | 325 | 2860 | 9 | 615 | 35 | 0.127 | 6.4 | 0.000429 | 2.7 | 6 | 0.000402 | 3.4 | 8.12 | 0.27 | 0.000403 | 2.8 | **8.15** | **0.22** |
| | ZEI1@12 | 117 | 1943 | 17 | 312 | 24 | 0.139 | 5.4 | 0.000394 | 2.7 | 8 | 0.000352 | 3.9 | 7.12 | 0.28 | 0.000361 | 2.7 | **7.30** | **0.20** |
| | ZEI1@18 | 258 | 3058 | 12 | 393 | 25 | 0.127 | 5.0 | 0.000398 | 2.6 | 8 | 0.000365 | 3.5 | 7.37 | 0.26 | 0.000365 | 2.7 | **7.38** | **0.20** |
| | ZEI1@23 | 78 | 6403 | 82 | 744 | 20 | 0.307 | 4.7 | 0.000464 | 2.6 | 4 | 0.000440 | 2.7 | 8.89 | 0.24 | 0.000447 | 2.6 | **9.03** | **0.24** |
| | SCHR1@01 | 423 | 37708 | 89 | 1766 | 6 | 0.126 | 2.4 | 0.001025 | 2.19 | 1 | 0.001011 | 2.2 | 20.42 | 0.44 | 0.001016 | 2.2 | **20.53** | **0.45** |
| | SCHR1@02 | 499 | 36495 | 73 | 2194 | 8 | 0.124 | 2.7 | 0.001012 | 2.51 | 1 | 0.001004 | 2.5 | 20.29 | 0.51 | 0.001004 | 2.5 | **20.28** | **0.51** |
| A | SCHR1@03 | 417 | 45163 | 108 | 1969 | 5 | 0.132 | 2.2 | 0.001090 | 2.18 | 1 | 0.001079 | 2.2 | 21.80 | 0.47 | 0.001082 | 2.2 | **21.85** | **0.48** |
| | SCHR1@04 | 438 | 43245 | 99 | 1960 | 6 | 0.142 | 2.3 | 0.001048 | 2.18 | 1 | 0.001038 | 2.2 | 20.97 | 0.45 | 0.001039 | 2.2 | **20.98** | **0.46** |
| | SCHR1@05 | 465 | 37059 | 80 | 2172 | 7 | 0.126 | 2.5 | 0.001016 | 2.18 | 1 | 0.001006 | 2.2 | 20.32 | 0.44 | 0.001007 | 2.2 | **20.35** | **0.44** |
| | SCHR1@06 | 443 | 46170 | 104 | 1859 | 5 | 0.137 | 2.3 | 0.001064 | 2.20 | 1 | 0.001051 | 2.2 | 21.24 | 0.46 | 0.001056 | 2.2 | **21.33** | **0.47** |
| | SCHR1@07 | 509 | 24001 | 47 | 1168 | 6 | 0.160 | 2.0 | 0.001037 | 2.18 | 2 | 0.001009 | 2.1 | 20.38 | 0.43 | 0.001013 | 2.2 | **20.47** | **0.45** |
| | SCHR1@08 | 482 | 25025 | 52 | 1167 | 6 | 0.158 | 2.0 | 0.001060 | 2.20 | 2 | 0.001033 | 2.2 | 20.88 | 0.45 | 0.001037 | 2.2 | **20.95** | **0.46** |
| | SCHR1@09 | 478 | 26320 | 55 | 1119 | 6 | 0.167 | 1.9 | 0.001040 | 2.19 | 2 | 0.001012 | 2.1 | 20.44 | 0.44 | 0.001016 | 2.2 | **20.54** | **0.45** |
| | SCHR1@10 | 510 | 28255 | 55 | 1211 | 6 | 0.168 | 2.0 | 0.001051 | 2.17 | 2 | 0.001026 | 2.1 | 20.72 | 0.44 | 0.001030 | 2.2 | **20.81** | **0.45** |
| | SCHR1@11 | 465 | 26453 | 57 | 1145 | 6 | 0.169 | 1.8 | 0.001076 | 2.18 | 2 | 0.001053 | 2.1 | 21.28 | 0.46 | 0.001052 | 2.2 | **21.25** | **0.46** |
| | SCHR1@12 | 331 | 27180 | 82 | 1099 | 6 | 0.212 | 2.0 | 0.001043 | 2.18 | 2 | 0.001018 | 2.1 | 20.57 | 0.44 | 0.001020 | 2.2 | **20.60** | **0.45** |
| | SCHR1@13 | 312 | 28957 | 93 | 1243 | 6 | 0.233 | 2.0 | 0.001052 | 2.19 | 2 | 0.001028 | 2.1 | 20.76 | 0.44 | 0.001027 | 2.2 | **20.75** | **0.45** |
| B | SCHR1@14 | 316 | 29632 | 94 | 1200 | 6 | 0.239 | 2.1 | 0.001031 | 2.22 | 2 | 0.001004 | 2.2 | 20.29 | 0.44 | 0.001008 | 2.2 | **20.37** | **0.45** |
| | SCHR1@15 | 331 | 31105 | 94 | 1151 | 5 | 0.247 | 2.1 | 0.001016 | 2.21 | 2 | 0.000993 | 2.2 | 20.05 | 0.43 | 0.000993 | 2.2 | **20.07** | **0.44** |
| | SCHR1@16 | 471 | 25641 | 54 | 1676 | 7 | 0.132 | 2.3 | 0.001020 | 2.19 | 1 | 0.001004 | 2.2 | 20.27 | 0.44 | 0.001005 | 2.2 | **20.31** | **0.44** |
| | SCHR1@17 | 477 | 28482 | 60 | 1358 | 6 | 0.151 | 2.3 | 0.001020 | 2.20 | 2 | 0.001001 | 2.2 | 20.23 | 0.44 | 0.001005 | 2.2 | **20.30** | **0.45** |

| | | | | | | | | | | | | | | | | | | |
|---|---|---|---|---|---|---|---|---|---|---|---|---|---|---|---|---|---|---|
| | **SCHR1@18** | 480 | 18234 | 38 | 1599 | 9 | 0.119 | 2.5 | 0.000998 | 2.21 | 2 | 0.000976 | 2.2 | 19.72 | 0.43 | 0.000981 | 2.2 | **19.83** | **0.44** |
| | **SCHR1@19** | 492 | 18252 | 37 | 1339 | 8 | 0.116 | 2.5 | 0.001004 | 2.21 | 2 | 0.000981 | 2.2 | 19.81 | 0.43 | 0.000988 | 2.2 | **19.95** | **0.44** |
| | **SCHR1@20** | 506 | 18432 | 36 | 1461 | 8 | 0.117 | 2.5 | 0.000992 | 2.25 | 2 | 0.000973 | 2.2 | 19.66 | 0.44 | 0.000976 | 2.3 | **19.72** | **0.44** |
| | **SCHR1@21** | 443 | 21802 | 49 | 1500 | 8 | 0.134 | 2.4 | 0.000997 | 2.19 | 2 | 0.000979 | 2.2 | 19.78 | 0.43 | 0.000980 | 2.2 | **19.81** | **0.43** |
| | **SCHR1@22** | 453 | 24816 | 55 | 1475 | 7 | 0.149 | 2.4 | 0.001007 | 2.18 | 2 | 0.000991 | 2.2 | 20.03 | 0.43 | 0.000991 | 2.2 | **20.01** | **0.44** |
| | **SCHR1@23** | 211 | 16187 | 77 | 921 | 7 | 0.323 | 2.4 | 0.000994 | 2.42 | 4 | 0.000957 | 2.3 | 19.34 | 0.45 | 0.000954 | 2.4 | **19.27** | **0.47** |
| | **SCHR1@24** | 306 | 17861 | 58 | 513 | 5 | 0.371 | 2.0 | 0.001025 | 2.39 | 6 | 0.000960 | 2.3 | 19.39 | 0.44 | 0.000963 | 2.4 | **19.45** | **0.47** |
| C | **SCHR1@25** | 363 | 17322 | 48 | 546 | 6 | 0.370 | 2.3 | 0.001036 | 2.65 | 6 | 0.000968 | 2.5 | 19.55 | 0.49 | 0.000975 | 2.7 | **19.70** | **0.52** |
| | **SCHR1@26** | 447 | 17105 | 38 | 1579 | 11 | 0.168 | 3.8 | 0.001047 | 3.28 | 2 | 0.001027 | 3.2 | 20.74 | 0.67 | 0.001029 | 3.3 | **20.78** | **0.68** |
| | **SCHR1@27** | 351 | 18354 | 52 | 1635 | 10 | 0.177 | 3.3 | 0.000973 | 2.75 | 2 | 0.000954 | 2.7 | 19.28 | 0.52 | 0.000955 | 2.8 | **19.30** | **0.53** |
| | **SCHR1@28** | 499 | 18145 | 36 | 1394 | 11 | 0.215 | 3.5 | 0.001031 | 3.21 | 3 | 0.001006 | 3.1 | 20.33 | 0.64 | 0.001004 | 3.2 | **20.28** | **0.65** |
| | | | | | | | | | | | | | | | | | | | |
| | **MAYR4@01** | 259 | 5113 | 20 | 514 | 18 | 0.127 | 3.3 | 0.000567 | 1.3 | 4 | 0.000527 | 1.8 | 10.65 | 0.19 | 0.000542 | 1.3 | **10.95** | **0.14** |
| | **MAYR4@02** | 340 | 3449 | 10 | 341 | 19 | 0.120 | 3.5 | 0.000527 | 1.5 | 6 | 0.000468 | 2.5 | 9.45 | 0.23 | 0.000493 | 1.5 | **9.96** | **0.15** |
| | **MAYR4@03** | 336 | 3763 | 11 | 305 | 17 | 0.109 | 3.6 | 0.000541 | 1.5 | 5 | 0.000476 | 2.5 | 9.62 | 0.24 | 0.000512 | 1.5 | **10.35** | **0.15** |
| | **MAYR4@04** | 368 | 3691 | 10 | 424 | 21 | 0.114 | 3.8 | 0.000496 | 1.6 | 6 | 0.000451 | 2.4 | 9.12 | 0.22 | 0.000467 | 1.7 | **9.44** | **0.16** |
| | **MAYR4@05** | 311 | 2301 | 7 | 246 | 17 | 0.135 | 3.0 | 0.000636 | 1.9 | 11 | 0.000536 | 3.1 | 10.84 | 0.34 | 0.000565 | 2.0 | **11.41** | **0.23** |
| | **MAYR4@06** | 385 | 1190 | 3 | 189 | 21 | 0.097 | 3.2 | 0.000618 | 2.3 | 15 | 0.000504 | 4.8 | 10.19 | 0.49 | 0.000524 | 2.6 | **10.58** | **0.28** |
| | **MAYR4@07** | 136 | 6949 | 51 | 411 | 13 | 0.294 | 2.8 | 0.000601 | 1.2 | 7 | 0.000544 | 1.6 | 10.99 | 0.18 | 0.000561 | 1.2 | **11.33** | **0.13** |
| | **MAYR4@08** | 192 | 5038 | 26 | 370 | 15 | 0.224 | 2.9 | 0.000599 | 1.2 | 8 | 0.000536 | 1.9 | 10.84 | 0.21 | 0.000553 | 1.3 | **11.17** | **0.14** |
| | **MAYR4@09** | 201 | 4829 | 24 | 293 | 13 | 0.233 | 2.7 | 0.000602 | 1.4 | 9 | 0.000522 | 2.1 | 10.55 | 0.22 | 0.000547 | 1.4 | **11.06** | **0.16** |
| | **MAYR4@10** | 251 | 4665 | 19 | 427 | 16 | 0.139 | 3.2 | 0.000600 | 1.2 | 5 | 0.000553 | 1.9 | 11.17 | 0.21 | 0.000568 | 1.3 | **11.48** | **0.15** |
| | **MAYR4@11** | 334 | 1748 | 5 | 300 | 24 | 0.103 | 3.4 | 0.000511 | 2.2 | 14 | 0.000446 | 3.7 | 9.00 | 0.33 | 0.000442 | 2.4 | **8.93** | **0.21** |
| | **MAYR4@12** | 295 | 2169 | 7 | 229 | 18 | 0.112 | 3.5 | 0.000582 | 2.0 | 9 | 0.000491 | 3.5 | 9.92 | 0.35 | 0.000531 | 2.1 | **10.72** | **0.22** |
| | **MAYR4@13** | 81 | 5010 | 62 | 404 | 16 | 0.335 | 3.4 | 0.000627 | 1.3 | 7 | 0.000574 | 1.9 | 11.60 | 0.23 | 0.000584 | 1.3 | **11.80** | **0.15** |
| | **MAYR4@14** | 103 | 5475 | 53 | 381 | 14 | 0.289 | 3.3 | 0.000602 | 1.2 | 6 | 0.000541 | 1.8 | 10.92 | 0.20 | 0.000564 | 1.2 | **11.39** | **0.13** |
| | **MAYR4@15** | 94 | 6085 | 65 | 457 | 15 | 0.274 | 3.6 | 0.000545 | 1.4 | 5 | 0.000499 | 1.8 | 10.09 | 0.18 | 0.000519 | 1.4 | **10.48** | **0.15** |
| | **MAYR4@16** | 122 | 6082 | 50 | 486 | 15 | 0.236 | 3.5 | 0.000546 | 1.0 | 4 | 0.000503 | 1.5 | 10.16 | 0.15 | 0.000522 | 1.0 | **10.56** | **0.11** |
| | **MAYR4@17** | 134 | 5776 | 43 | 381 | 14 | 0.270 | 3.1 | 0.000553 | 0.9 | 6 | 0.000497 | 1.6 | 10.04 | 0.16 | 0.000519 | 1.0 | **10.50** | **0.10** |
| | **MAYR4@18** | 180 | 4227 | 23 | 351 | 14 | 0.200 | 3.0 | 0.000603 | 1.1 | 7 | 0.000543 | 1.9 | 10.96 | 0.21 | 0.000561 | 1.1 | **11.34** | **0.12** |

| Group | ID | | | | | | | | | | | | | | | | | |
|---|---|---|---|---|---|---|---|---|---|---|---|---|---|---|---|---|---|---|
| A | PFIT1@12 | 103 | 11683 | 114 | 554 | 6 | 0.387 | 2.6 | 0.000852 | 2.19 | 5 | 0.000809 | 2.1 | 16.34 | 0.35 | 0.000813 | 2.2 | **16.42** | **0.36** |
| | PFIT1@13 | 112 | 18466 | 165 | 580 | 5 | 0.450 | 2.4 | 0.000891 | 2.29 | 4 | 0.000845 | 2.2 | 17.06 | 0.38 | 0.000852 | 2.3 | **17.21** | **0.39** |
| | PFIT1@14 | 117 | 13986 | 120 | 569 | 5 | 0.413 | 2.4 | 0.000918 | 2.31 | 5 | 0.000868 | 2.2 | 17.54 | 0.39 | 0.000874 | 2.3 | **17.66** | **0.41** |
| | PFIT1@15 | 104 | 13449 | 130 | 473 | 5 | 0.430 | 2.3 | 0.000895 | 2.41 | 5 | 0.000835 | 2.3 | 16.87 | 0.39 | 0.000849 | 2.4 | **17.15** | **0.41** |
| | PFIT1@16 | 65 | 15956 | 244 | 629 | 5 | 0.497 | 2.8 | 0.000913 | 2.26 | 4 | 0.000877 | 2.2 | 17.72 | 0.39 | 0.000881 | 2.3 | **17.80** | **0.40** |
| | PFIT1@17 | 78 | 13368 | 171 | 586 | 7 | 0.506 | 3.3 | 0.000877 | 2.25 | 4 | 0.000833 | 2.2 | 16.82 | 0.37 | 0.000843 | 2.3 | **17.03** | **0.38** |
| | PFIT1@18 | 79 | 12054 | 153 | 662 | 7 | 0.365 | 3.6 | 0.000892 | 2.40 | 3 | 0.000859 | 2.4 | 17.35 | 0.41 | 0.000866 | 2.4 | **17.49** | **0.42** |
| | PFIT1@19 | 121 | 9937 | 82 | 697 | 10 | 0.328 | 4.6 | 0.000896 | 2.19 | 3 | 0.000856 | 2.2 | 17.29 | 0.38 | 0.000869 | 2.2 | **17.57** | **0.39** |
| | PFIT1@20 | 119 | 8358 | 70 | 871 | 11 | 0.287 | 4.9 | 0.000885 | 2.19 | 3 | 0.000853 | 2.2 | 17.23 | 0.37 | 0.000862 | 2.2 | **17.42** | **0.38** |
| | *PFIT1@01* | *30* | *5962* | *199* | *808* | *12* | *0.360* | *5.4* | *0.000658* | *2.29* | *3* | *0.000631* | *2.3* | *12.74* | *0.29* | *0.000640* | *2.3* | *12.94* | *0.30* |
| | *PFIT1@02* | *32* | *6627* | *209* | *723* | *10* | *0.350* | *5.0* | *0.000754* | *2.25* | *2* | *0.000721* | *2.2* | *14.57* | *0.32* | *0.000736* | *2.3* | *14.86* | *0.34* |
| | *PFIT1@03* | *13* | *5940* | *466* | *1033* | *11* | *0.414* | *6.9* | *0.000824* | *2.24* | *1* | *0.000808* | *2.2* | *16.33* | *0.37* | *0.000814* | *2.2* | *16.45* | *0.37* |
| | *PFIT1@04* | *36* | *6027* | *166* | *859* | *13* | *0.317* | *5.7* | *0.000705* | *2.28* | *3* | *0.000681* | *2.3* | *13.77* | *0.31* | *0.000687* | *2.3* | *13.88* | *0.32* |
| | *PFIT1@05* | *84* | *8610* | *102* | *1011* | *19* | *0.389* | *7.1* | *0.000710* | *3.39* | *4* | *0.000690* | *3.4* | *13.94* | *0.47* | *0.000685* | *3.4* | *13.84* | *0.47* |
| | *PFIT1@11* | *110* | *13146* | *119* | *548* | *6* | *0.428* | *2.7* | *0.000800* | *2.18* | *4* | *0.000756* | *2.1* | *15.28* | *0.32* | *0.000764* | *2.2* | *15.43* | *0.34* |
| B | PFIT1@06 | 245 | 5464 | 22 | 525 | 10 | 0.155 | 3.0 | 0.000697 | 2.20 | 6 | 0.000664 | 2.2 | 13.42 | 0.30 | 0.000658 | 2.2 | **13.30** | **0.29** |
| | PFIT1@07 | 336 | 4523 | 13 | 381 | 11 | 0.161 | 3.6 | 0.000706 | 2.33 | 6 | 0.000642 | 2.4 | 12.98 | 0.31 | 0.000661 | 2.3 | **13.35** | **0.31** |
| | PFIT1@08 | 347 | 3300 | 10 | 314 | 12 | 0.201 | 3.5 | 0.000728 | 2.21 | 11 | 0.000647 | 2.4 | 13.07 | 0.32 | 0.000648 | 2.3 | **13.09** | **0.30** |
| | PFIT1@09 | 383 | 3220 | 8 | 343 | 13 | 0.170 | 3.9 | 0.000704 | 2.21 | 9 | 0.000636 | 2.5 | 12.86 | 0.32 | 0.000638 | 2.3 | **12.89** | **0.29** |
| | PFIT1@10 | 388 | 2889 | 7 | 238 | 11 | 0.190 | 3.7 | 0.000740 | 2.33 | 12 | 0.000641 | 2.8 | 12.95 | 0.36 | 0.000650 | 2.4 | **13.13** | **0.32** |
| | BURG2@07 | 30 | 769 | 26 | 193 | 20 | 0.165 | 4.7 | 0.000732 | 2.3 | 10 | 0.000585 | 4.4 | 11.83 | 0.53 | 0.000659 | 2.4 | **13.32** | **0.31** |
| | BURG2@08 | 18 | 1014 | 57 | 219 | 16 | 0.306 | 3.6 | 0.000959 | 2.2 | 12 | 0.000790 | 3.4 | 15.97 | 0.54 | 0.000844 | 2.3 | **17.06** | **0.39** |
| | BURG2@09 | 53 | 626 | 12 | 275 | 24 | 0.087 | 5.1 | 0.000797 | 2.3 | 6 | 0.000685 | 3.9 | 13.85 | 0.54 | 0.000751 | 2.4 | **15.17** | **0.36** |
| | BURG2@10 | 30 | 1067 | 36 | 324 | 19 | 0.179 | 4.1 | 0.000878 | 2.3 | 7 | 0.000774 | 3.0 | 15.63 | 0.47 | 0.000818 | 2.3 | **16.53** | **0.38** |
| | BURG2@13 | 164 | 529 | 3 | 403 | 55 | 0.112 | 7.5 | 0.000683 | 2.3 | 12 | 0.000660 | 6.5 | 13.35 | 0.86 | 0.000600 | 2.8 | **12.12** | **0.33** |
| | *BURG2@12* | *52* | *2273* | *44* | *69* | *18* | *0.415* | *3.7* | *0.001048* | *8.6* | *46* | *0.000462* | *10.7* | *9.34* | *1.00* | *0.000571* | *9.5* | *11.53* | *1.10* |
| | PLAN1@01 | 190 | 1647 | 9 | 56 | 9 | 0.577 | 2.2 | 0.001582 | 3.1 | 63 | 0.000706 | 6.9 | 14.27 | 0.98 | 0.000580 | 5.1 | **11.73** | **0.60** |
| | PLAN1@02 | 177 | 12498 | 71 | 284 | 11 | 0.361 | 2.9 | 0.000643 | 1.4 | 9 | 0.000585 | 2.1 | 11.81 | 0.25 | 0.000587 | 1.5 | **11.86** | **0.17** |
| | PLAN1@03 | 209 | 2322 | 11 | 56 | 8 | 0.620 | 1.9 | 0.001634 | 3.3 | 63 | 0.000578 | 5.5 | 11.67 | 0.64 | 0.000605 | 4.7 | **12.23** | **0.57** |

| | | | | | | | | | | | | | | | | | | |
|---|---|---|---|---|---|---|---|---|---|---|---|---|---|---|---|---|---|---|
| | PLAN1@04 | 410 | 3160 | 8 | 101 | 12 | 0.269 | 2.8 | 0.000788 | 2.2 | 28 | 0.000583 | 5.4 | 11.78 | 0.63 | 0.000565 | 2.6 | **11.41** | **0.29** |
| | PLAN1@05 | 438 | 1542 | 4 | 59 | 11 | 0.261 | 2.7 | 0.001043 | 3.3 | 45 | 0.000489 | 7.9 | 9.88 | 0.78 | 0.000574 | 4.5 | **11.61** | **0.52** |
| | PLAN1@06 | 458 | 1687 | 4 | 58 | 10 | 0.377 | 2.4 | 0.001231 | 6.5 | 55 | 0.000575 | 8.0 | 11.62 | 0.93 | 0.000550 | 7.5 | **11.11** | **0.84** |
| | PLAN1@07 | 462 | 2662 | 6 | 90 | 12 | 0.284 | 2.5 | 0.000864 | 4.0 | 40 | 0.000635 | 6.3 | 12.83 | 0.81 | 0.000520 | 4.5 | **10.50** | **0.47** |
| | PLAN1@08 | 509 | 3332 | 7 | 104 | 14 | 0.242 | 3.3 | 0.000751 | 2.3 | 24 | 0.000607 | 6.0 | 12.26 | 0.74 | 0.000569 | 2.7 | **11.50** | **0.31** |
| | PLAN1@09 | 594 | 4167 | 7 | 101 | 12 | 0.238 | 2.9 | 0.000690 | 2.4 | 24 | 0.000501 | 5.1 | 10.12 | 0.51 | 0.000522 | 2.6 | **10.55** | **0.28** |
| | PLAN1@10 | 774 | 5878 | 8 | 148 | 12 | 0.190 | 2.8 | 0.000661 | 2.1 | 17 | 0.000557 | 3.9 | 11.26 | 0.44 | 0.000550 | 2.2 | **11.11** | **0.25** |
| | PLAN1@13 | 1215 | 1759 | 1 | 68 | 12 | 0.136 | 2.5 | 0.000967 | 5.4 | 41 | 0.000576 | 8.2 | 11.65 | 0.96 | 0.000567 | 6.6 | **11.45** | **0.75** |
| | PLAN1@14 | 589 | 4750 | 8 | 176 | 14 | 0.167 | 3.2 | 0.000680 | 1.5 | 15 | 0.000581 | 3.7 | 11.73 | 0.43 | 0.000579 | 1.6 | **11.71** | **0.19** |
| | PLAN1@15 | 424 | 3653 | 9 | 98 | 11 | 0.233 | 2.9 | 0.000783 | 1.8 | 22 | 0.000624 | 5.1 | 12.61 | 0.65 | 0.000610 | 2.0 | **12.32** | **0.25** |
| | PLAN1@16 | 448 | 3567 | 8 | 86 | 10 | 0.288 | 2.4 | 0.000869 | 2.9 | 30 | 0.000596 | 5.2 | 12.04 | 0.63 | 0.000611 | 3.2 | **12.35** | **0.39** |
| | PLAN1@17 | 454 | 3815 | 8 | 108 | 11 | 0.198 | 3.0 | 0.000744 | 1.9 | 18 | 0.000566 | 4.7 | 11.44 | 0.54 | 0.000608 | 2.1 | **12.28** | **0.26** |
| | PLAN1@18 | 443 | 4422 | 10 | 161 | 13 | 0.250 | 2.7 | 0.000763 | 3.9 | 21 | 0.000670 | 4.9 | 13.53 | 0.67 | 0.000600 | 4.0 | **12.13** | **0.48** |
| | PLAN1@19 | 365 | 10351 | 28 | 214 | 10 | 0.305 | 2.7 | 0.000685 | 1.8 | 11 | 0.000596 | 2.5 | 12.04 | 0.31 | 0.000609 | 1.8 | **12.30** | **0.22** |
| | PLAN1@20 | 467 | 8041 | 17 | 153 | 9 | 0.359 | 2.1 | 0.000751 | 1.5 | 22 | 0.000606 | 2.8 | 12.25 | 0.34 | 0.000588 | 1.6 | **11.88** | **0.20** |
| A | PLAN1@21 | 627 | 6050 | 10 | 103 | 9 | 0.322 | 2.0 | 0.000836 | 2.0 | 28 | 0.000577 | 3.7 | 11.66 | 0.43 | 0.000600 | 2.2 | **12.12** | **0.27** |
| | PLAN1@23 | 373 | 4646 | 12 | 715 | 17 | 0.102 | 2.9 | 0.000645 | 2.6 | 4 | 0.000611 | 2.7 | 12.34 | 0.33 | 0.000617 | 2.7 | **12.46** | **0.33** |
| | PLAN1@24 | 390 | 4664 | 12 | 677 | 18 | 0.090 | 3.5 | 0.000610 | 2.6 | 4 | 0.000576 | 2.7 | 11.63 | 0.31 | 0.000588 | 2.7 | **11.89** | **0.32** |
| | PLAN1@25 | 359 | 5180 | 14 | 519 | 17 | 0.121 | 3.0 | 0.000639 | 2.7 | 6 | 0.000591 | 2.8 | 11.94 | 0.33 | 0.000598 | 2.7 | **12.09** | **0.32** |
| | PLAN1@26 | 365 | 4817 | 13 | 401 | 15 | 0.116 | 3.0 | 0.000627 | 2.6 | 6 | 0.000570 | 2.8 | 11.52 | 0.32 | 0.000588 | 2.7 | **11.88** | **0.32** |
| | PLAN1@27 | 476 | 3538 | 7 | 541 | 19 | 0.101 | 2.9 | 0.000650 | 2.7 | 8 | 0.000603 | 2.8 | 12.19 | 0.35 | 0.000601 | 2.7 | **12.14** | **0.33** |
| | PLAN1@28 | 564 | 8192 | 15 | 460 | 11 | 0.125 | 2.3 | 0.000660 | 2.6 | 6 | 0.000604 | 2.6 | 12.21 | 0.32 | 0.000622 | 2.6 | **12.56** | **0.33** |
| | PLAN1@29 | 620 | 14199 | 23 | 619 | 10 | 0.130 | 2.1 | 0.000637 | 2.6 | 4 | 0.000598 | 2.6 | 12.09 | 0.31 | 0.000609 | 2.6 | **12.32** | **0.33** |
| | PLAN1@30 | 528 | 8241 | 16 | 420 | 13 | 0.127 | 2.7 | 0.000632 | 2.7 | 6 | 0.000574 | 2.7 | 11.60 | 0.31 | 0.000595 | 2.7 | **12.02** | **0.32** |
| | PLAN1@31 | 1028 | 1527 | 1 | 124 | 12 | 0.090 | 2.0 | 0.000802 | 3.1 | 26 | 0.000552 | 4.4 | 11.16 | 0.49 | 0.000595 | 3.8 | **12.02** | **0.45** |
| | PLAN1@32 | 1152 | 1721 | 1 | 227 | 19 | 0.066 | 2.7 | 0.000661 | 2.7 | 12 | 0.000548 | 4.0 | 11.08 | 0.44 | 0.000579 | 3.1 | **11.70** | **0.36** |
| | PLAN1@33 | 926 | 4303 | 5 | 361 | 14 | 0.085 | 2.2 | 0.000642 | 2.7 | 8 | 0.000573 | 2.8 | 11.57 | 0.33 | 0.000588 | 2.8 | **11.87** | **0.33** |
| | PLAN1@34 | 623 | 7966 | 13 | 483 | 12 | 0.125 | 2.3 | 0.000669 | 2.6 | 7 | 0.000615 | 2.6 | 12.43 | 0.32 | 0.000624 | 2.6 | **12.61** | **0.33** |
| | PLAN1@35 | 408 | 6529 | 16 | 185 | 8 | 0.307 | 1.7 | 0.000746 | 3.1 | 19 | 0.000592 | 3.0 | 11.97 | 0.36 | 0.000605 | 3.2 | **12.22** | **0.39** |
| | PLAN1@36 | 304 | 7521 | 25 | 427 | 13 | 0.182 | 2.7 | 0.000646 | 2.6 | 7 | 0.000587 | 2.7 | 11.87 | 0.31 | 0.000603 | 2.6 | **12.19** | **0.32** |
| | PLAN1@38 | 580 | 9607 | 17 | 525 | 14 | 0.112 | 2.4 | 0.000626 | 2.2 | 5 | 0.000580 | 2.3 | 11.73 | 0.27 | 0.000596 | 2.2 | **12.05** | **0.27** |

| | | | | | | | | | | | | | | | | | | |
|---|---|---|---|---|---|---|---|---|---|---|---|---|---|---|---|---|---|---|
| | **PLAN1@39** | 331 | 10707 | 32 | 497 | 13 | 0.164 | 2.7 | 0.000628 | 2.2 | 4 | 0.000581 | 2.3 | 11.74 | 0.27 | 0.000600 | 2.2 | **12.12** | **0.27** |
| | **PLAN1@40** | 344 | 2342 | 7 | 311 | 21 | 0.143 | 2.9 | 0.000698 | 2.3 | 15 | 0.000612 | 3.3 | 12.36 | 0.41 | 0.000593 | 2.5 | **11.98** | **0.30** |
| | **PLAN1@41** | 487 | 10698 | 22 | 360 | 11 | 0.172 | 2.1 | 0.000654 | 2.2 | 7 | 0.000584 | 2.3 | 11.80 | 0.27 | 0.000608 | 2.3 | **12.28** | **0.28** |
| | **PLAN1@11** | 1410 | 6659 | 5 | 175 | 20 | 0.204 | 4.3 | 0.000568 | 2.5 | 18 | 0.000578 | 6.4 | 11.69 | 0.74 | 0.000464 | 2.8 | **9.38** | **0.26** |
| B | **PLAN1@12** | 1310 | 7371 | 6 | 160 | 23 | 0.206 | 5.3 | 0.000467 | 2.1 | 18 | 0.000509 | 7.9 | 10.28 | 0.81 | 0.000384 | 2.5 | **7.75** | **0.19** |
| | **PLAN1@37** | 619 | 3146 | 5 | 312 | 20 | 0.085 | 2.7 | 0.000555 | 2.6 | 11 | 0.000493 | 3.5 | 9.96 | 0.34 | 0.000492 | 2.7 | **9.94** | **0.27** |
| | *PLAN1@22* | *733* | *1514* | *2* | *52* | *9* | *0.291* | *2.1* | *0.001227* | *7.7* | *60* | *0.000544* | *8.3* | *10.99* | *0.92* | *0.000491* | *9.1* | *9.91* | *0.90* |

*Central Tauern Window*

| | | | | | | | | | | | | | | | | | | |
|---|---|---|---|---|---|---|---|---|---|---|---|---|---|---|---|---|---|---|
| | **SCHEI1@16** | 408 | 58860 | 144 | 1205 | 6 | 0.181 | 1.8 | 0.000876 | 2.6 | 1 | 0.000849 | 2.6 | 17.14 | 0.44 | 0.000864 | 2.6 | **17.46** | **0.46** |
| A | **SCHEI1@17** | 304 | 61361 | 202 | 1483 | 7 | 0.210 | 1.9 | 0.000939 | 2.6 | 1 | 0.000915 | 2.6 | 18.49 | 0.48 | 0.000928 | 2.6 | **18.75** | **0.50** |
| | **SCHEI1@18** | 265 | 60179 | 227 | 1432 | 6 | 0.235 | 1.9 | 0.000949 | 2.6 | 1 | 0.000924 | 2.6 | 18.66 | 0.48 | 0.000937 | 2.6 | **18.93** | **0.50** |
| | **SCHEI1@20** | 311 | 62554 | 201 | 1416 | 7 | 0.222 | 2.0 | 0.000906 | 2.7 | 1 | 0.000881 | 2.6 | 17.81 | 0.46 | 0.000893 | 2.7 | **18.05** | **0.48** |
| | **SCHEI1@11** | 394 | 53473 | 136 | 780 | 7 | 0.309 | 2.0 | 0.000868 | 1.4 | 3 | 0.000829 | 1.4 | 16.75 | 0.23 | 0.000842 | 1.4 | **17.02** | **0.23** |
| | **SCHEI1@12** | 434 | 47204 | 109 | 857 | 9 | 0.268 | 2.2 | 0.000886 | 1.4 | 3 | 0.000855 | 1.4 | 17.28 | 0.24 | 0.000862 | 1.4 | **17.41** | **0.24** |
| B | **SCHEI1@13** | 360 | 50556 | 140 | 639 | 8 | 0.349 | 2.1 | 0.000907 | 1.4 | 3 | 0.000861 | 1.4 | 17.39 | 0.24 | 0.000876 | 1.4 | **17.71** | **0.24** |
| | **SCHEI1@14** | 321 | 53530 | 167 | 655 | 7 | 0.402 | 1.9 | 0.000911 | 1.4 | 4 | 0.000863 | 1.4 | 17.45 | 0.24 | 0.000876 | 1.4 | **17.69** | **0.25** |
| | **SCHEI1@32** | 291 | 55126 | 189 | 1026 | 6 | 0.303 | 1.5 | 0.000868 | 2.8 | 2 | 0.000836 | 2.7 | 16.88 | 0.46 | 0.000848 | 2.8 | **17.14** | **0.48** |
| | **SCHEI1@01** | 776 | 17285 | 22 | 303 | 8 | 0.196 | 2.1 | 0.000888 | 1.4 | 7 | 0.000802 | 1.7 | 16.20 | 0.27 | 0.000825 | 1.4 | **16.67** | **0.23** |
| | **SCHEI1@02** | 485 | 32602 | 67 | 655 | 11 | 0.283 | 2.6 | 0.000843 | 1.4 | 4 | 0.000808 | 1.5 | 16.33 | 0.25 | 0.000808 | 1.4 | **16.33** | **0.24** |
| | **SCHEI1@03** | 496 | 23525 | 47 | 411 | 9 | 0.285 | 2.2 | 0.000899 | 1.4 | 6 | 0.000826 | 1.6 | 16.70 | 0.26 | 0.000844 | 1.4 | **17.05** | **0.25** |
| | **SCHEI1@04** | 619 | 20187 | 33 | 312 | 10 | 0.254 | 2.5 | 0.000862 | 1.5 | 7 | 0.000797 | 1.9 | 16.10 | 0.31 | 0.000802 | 1.5 | **16.21** | **0.25** |
| | **SCHEI1@05** | 702 | 14215 | 20 | 252 | 9 | 0.255 | 2.1 | 0.000929 | 1.5 | 11 | 0.000806 | 1.9 | 16.29 | 0.32 | 0.000825 | 1.5 | **16.66** | **0.25** |
| | **SCHEI1@06** | 831 | 11524 | 14 | 190 | 8 | 0.274 | 1.7 | 0.000974 | 1.8 | 18 | 0.000810 | 2.3 | 16.36 | 0.37 | 0.000800 | 1.8 | **16.16** | **0.30** |
| | **SCHEI1@07** | 545 | 19842 | 36 | 279 | 8 | 0.383 | 1.9 | 0.000949 | 1.6 | 12 | 0.000839 | 1.8 | 16.95 | 0.31 | 0.000840 | 1.6 | **16.97** | **0.27** |
| | **SCHEI1@08** | 902 | 10829 | 12 | 216 | 9 | 0.220 | 2.0 | 0.000979 | 1.4 | 14 | 0.000843 | 2.1 | 17.04 | 0.36 | 0.000843 | 1.5 | **17.03** | **0.25** |
| | **SCHEI1@09** | 888 | 10863 | 12 | 239 | 11 | 0.212 | 2.3 | 0.000942 | 1.8 | 13 | 0.000809 | 2.4 | 16.35 | 0.39 | 0.000819 | 1.8 | **16.55** | **0.31** |
| | **SCHEI1@10** | 1103 | 20671 | 19 | 314 | 11 | 0.206 | 2.6 | 0.000864 | 1.5 | 7 | 0.000791 | 2.0 | 15.98 | 0.31 | 0.000801 | 1.5 | **16.18** | **0.24** |
| | **SCHEI1@15** | 427 | 31409 | 74 | 280 | 6 | 0.426 | 1.7 | 0.000933 | 1.6 | 9 | 0.000818 | 1.7 | 16.53 | 0.28 | 0.000851 | 1.6 | **17.19** | **0.28** |
| C | **SCHEI1@19** | 650 | 22198 | 34 | 1132 | 11 | 0.093 | 2.6 | 0.000890 | 2.6 | 1 | 0.000860 | 2.6 | 17.37 | 0.45 | 0.000878 | 2.6 | **17.73** | **0.47** |

| Group | Sample | | | | | | | | | | | | | | | | | |
|---|---|---|---|---|---|---|---|---|---|---|---|---|---|---|---|---|---|---|
| | SCHEI1@21 | 704 | 20090 | 29 | 1036 | 10 | 0.086 | 2.2 | 0.000849 | 2.6 | 2 | 0.000818 | 2.6 | 16.52 | 0.42 | 0.000836 | 2.6 | **16.89** | **0.44** |
| | SCHEI1@22 | 815 | 10348 | 13 | 1078 | 15 | 0.078 | 2.3 | 0.000846 | 2.6 | 3 | 0.000818 | 2.6 | 16.52 | 0.43 | 0.000825 | 2.6 | **16.66** | **0.44** |
| | SCHEI1@23 | 417 | 36731 | 88 | 1703 | 12 | 0.133 | 2.8 | 0.000865 | 2.6 | 1 | 0.000847 | 2.6 | 17.11 | 0.44 | 0.000855 | 2.6 | **17.28** | **0.45** |
| | SCHEI1@24 | 656 | 16727 | 25 | 1183 | 11 | 0.083 | 2.3 | 0.000848 | 2.6 | 2 | 0.000821 | 2.6 | 16.59 | 0.43 | 0.000834 | 2.6 | **16.86** | **0.44** |
| | SCHEI1@27 | 720 | 19708 | 27 | 976 | 10 | 0.108 | 2.0 | 0.000831 | 2.6 | 3 | 0.000798 | 2.6 | 16.13 | 0.41 | 0.000809 | 2.6 | **16.35** | **0.43** |
| | SCHEI1@28 | 420 | 36082 | 86 | 1062 | 9 | 0.169 | 2.4 | 0.000860 | 2.6 | 2 | 0.000829 | 2.6 | 16.76 | 0.43 | 0.000846 | 2.6 | **17.08** | **0.45** |
| | SCHEI1@29 | 513 | 37241 | 73 | 665 | 9 | 0.231 | 2.2 | 0.000854 | 2.6 | 4 | 0.000804 | 2.5 | 16.25 | 0.41 | 0.000823 | 2.6 | **16.64** | **0.44** |
| | SCHEI1@30 | 588 | 23098 | 39 | 738 | 10 | 0.122 | 2.3 | 0.000808 | 2.7 | 3 | 0.000765 | 2.6 | 15.46 | 0.40 | 0.000787 | 2.7 | **15.89** | **0.43** |
| | SCHEI1@31 | 435 | 37275 | 86 | 945 | 8 | 0.176 | 2.0 | 0.000817 | 2.6 | 2 | 0.000784 | 2.5 | 15.84 | 0.40 | 0.000800 | 2.6 | **16.16** | **0.43** |
| | SCHEI1@33 | 678 | 12021 | 18 | 1406 | 18 | 0.060 | 2.6 | 0.000800 | 2.2 | 1 | 0.000778 | 2.2 | 15.73 | 0.35 | 0.000792 | 2.2 | **16.00** | **0.36** |
| | SCHEI1@34 | 996 | 8782 | 9 | 2159 | 31 | 0.056 | 3.1 | 0.000807 | 2.3 | 1 | 0.000793 | 2.3 | 16.02 | 0.37 | 0.000796 | 2.3 | **16.08** | **0.37** |
| A | HOPF2@15 | 82 | 61530 | 749 | 720 | 4 | 0.611 | 2.2 | 0.000560 | 2.37 | 2 | 0.000542 | 2.3 | 10.94 | 0.25 | 0.000546 | 2.4 | **11.03** | **0.26** |
| | HOPF2@16 | 69 | 33784 | 490 | 619 | 4 | 0.641 | 2.6 | 0.000647 | 2.36 | 3 | 0.000620 | 2.3 | 12.54 | 0.29 | 0.000625 | 2.4 | **12.63** | **0.30** |
| | HOPF2@17 | 97 | 33439 | 343 | 972 | 7 | 0.571 | 3.6 | 0.000663 | 2.91 | 2 | 0.000643 | 2.8 | 13.00 | 0.37 | 0.000647 | 2.9 | **13.08** | **0.38** |
| | HOPF2@18 | 77 | 40713 | 530 | 274 | 3 | 0.761 | 1.5 | 0.000762 | 2.67 | 11 | 0.000667 | 2.4 | 13.48 | 0.32 | 0.000677 | 2.7 | **13.67** | **0.37** |
| | HOPF2@19 | 120 | 62586 | 520 | 840 | 5 | 0.572 | 2.8 | 0.000623 | 3.49 | 2 | 0.000605 | 3.4 | 12.23 | 0.42 | 0.000609 | 3.5 | **12.30** | **0.43** |
| | HOPF2@20 | 120 | 49522 | 413 | 853 | 7 | 0.590 | 3.7 | 0.000594 | 2.39 | 3 | 0.000571 | 2.3 | 11.54 | 0.27 | 0.000577 | 2.4 | **11.67** | **0.28** |
| | HOPF2@21 | 173 | 37157 | 215 | 1309 | 13 | 0.585 | 6.1 | 0.000683 | 2.78 | 2 | 0.000667 | 2.7 | 13.47 | 0.37 | 0.000667 | 2.8 | **13.47** | **0.38** |
| B | HOPF2@07 | 49 | 28763 | 591 | 480 | 5 | 0.735 | 2.5 | 0.000600 | 2.26 | 6 | 0.000559 | 2.1 | 11.29 | 0.24 | 0.000561 | 2.3 | **11.35** | **0.26** |
| | HOPF2@08 | 46 | 28779 | 627 | 495 | 5 | 0.740 | 2.3 | 0.000630 | 2.20 | 6 | 0.000593 | 2.1 | 11.98 | 0.25 | 0.000592 | 2.2 | **11.97** | **0.26** |
| | HOPF2@09 | 55 | 32338 | 593 | 429 | 4 | 0.721 | 1.9 | 0.000649 | 2.17 | 7 | 0.000603 | 2.1 | 12.18 | 0.25 | 0.000604 | 2.2 | **12.20** | **0.27** |
| | HOPF2@10 | 76 | 22645 | 296 | 595 | 9 | 0.714 | 4.1 | 0.000637 | 3.67 | 6 | 0.000597 | 3.5 | 12.06 | 0.42 | 0.000596 | 3.7 | **12.05** | **0.44** |
| | HOPF2@11 | 40 | 28273 | 701 | 140 | 3 | 0.810 | 1.2 | 0.000782 | 3.59 | 24 | 0.000580 | 2.8 | 11.72 | 0.32 | 0.000597 | 3.6 | **12.07** | **0.43** |
| | HOPF2@12 | 38 | 26357 | 689 | 346 | 4 | 0.780 | 2.1 | 0.000694 | 2.20 | 9 | 0.000628 | 2.1 | 12.68 | 0.26 | 0.000628 | 2.2 | **12.69** | **0.28** |
| | HOPF2@13 | 52 | 40578 | 775 | 305 | 3 | 0.777 | 1.7 | 0.000688 | 2.54 | 10 | 0.000615 | 2.3 | 12.43 | 0.29 | 0.000616 | 2.5 | **12.45** | **0.32** |
| | HOPF2@14 | 55 | 27790 | 505 | 261 | 3 | 0.773 | 1.6 | 0.000708 | 2.22 | 12 | 0.000614 | 2.0 | 12.41 | 0.25 | 0.000626 | 2.2 | **12.65** | **0.28** |
| C | HOPF2@01 | 40 | 23096 | 582 | 683 | 6 | 0.646 | 3.0 | 0.000591 | 2.20 | 4 | 0.000565 | 2.1 | 11.43 | 0.24 | 0.000567 | 2.2 | **11.45** | **0.25** |
| | HOPF2@02 | 42 | 25870 | 612 | 584 | 6 | 0.682 | 3.3 | 0.000619 | 2.23 | 4 | 0.000590 | 2.2 | 11.92 | 0.26 | 0.000594 | 2.2 | **11.99** | **0.27** |
| | HOPF2@03 | 45 | 29023 | 642 | 538 | 5 | 0.659 | 2.6 | 0.000647 | 2.27 | 4 | 0.000617 | 2.2 | 12.46 | 0.27 | 0.000619 | 2.3 | **12.50** | **0.28** |
| | HOPF2@04 | 50 | 30246 | 610 | 510 | 4 | 0.677 | 2.3 | 0.000656 | 2.35 | 5 | 0.000621 | 2.2 | 12.54 | 0.28 | 0.000622 | 2.3 | **12.58** | **0.30** |

| Group | Sample | | | | | | | | | | | | | | | | | | |
|---|---|---|---|---|---|---|---|---|---|---|---|---|---|---|---|---|---|---|
| | HOPF2@05 | 57 | 32908 | 577 | 451 | 4 | 0.690 | 2.1 | 0.000670 | 2.48 | 6 | 0.000627 | 2.4 | 12.67 | 0.30 | 0.000629 | 2.5 | **12.70** | **0.32** |
| | HOPF2@06 | 58 | 28865 | 500 | 481 | 4 | 0.680 | 2.1 | 0.000642 | 2.18 | 6 | 0.000602 | 2.1 | 12.16 | 0.25 | 0.000602 | 2.2 | **12.17** | **0.27** |
| A | GART1@07 | 232 | 13668 | 59 | 1022 | 10 | 0.175 | 2.0 | 0.000830 | 2.3 | 2 | 0.000799 | 2.3 | 16.14 | 0.36 | 0.000811 | 2.3 | **16.38** | **0.38** |
| A | GART1@08 | 302 | 9312 | 31 | 523 | 8 | 0.208 | 1.6 | 0.000831 | 2.3 | 6 | 0.000769 | 2.3 | 15.54 | 0.35 | 0.000785 | 2.3 | **15.85** | **0.37** |
| A | GART1@09 | 206 | 14195 | 69 | 1054 | 9 | 0.173 | 2.0 | 0.000815 | 2.0 | 2 | 0.000785 | 2.0 | 15.86 | 0.31 | 0.000799 | 2.0 | **16.14** | **0.32** |
| A | GART1@10 | 199 | 17962 | 90 | 716 | 7 | 0.192 | 2.1 | 0.000838 | 2.1 | 2 | 0.000793 | 2.1 | 16.01 | 0.33 | 0.000823 | 2.1 | **16.63** | **0.36** |
| A | GART1@11 | 221 | 9789 | 44 | 881 | 12 | 0.162 | 2.5 | 0.000813 | 2.3 | 3 | 0.000778 | 2.3 | 15.71 | 0.36 | 0.000792 | 2.3 | **16.00** | **0.38** |
| A | GART1@22 | 86 | 8047 | 93 | 412 | 8 | 0.321 | 1.8 | 0.000889 | 2.0 | 6 | 0.000806 | 2.0 | 16.29 | 0.32 | 0.000835 | 2.0 | **16.86** | **0.34** |
| A | GART1@23 | 142 | 9137 | 65 | 454 | 8 | 0.317 | 1.8 | 0.000858 | 1.9 | 6 | 0.000786 | 1.9 | 15.88 | 0.29 | 0.000810 | 1.9 | **16.36** | **0.31** |
| A | GART1@24 | 133 | 7292 | 55 | 371 | 8 | 0.322 | 1.8 | 0.000864 | 2.3 | 7 | 0.000774 | 2.2 | 15.64 | 0.35 | 0.000808 | 2.3 | **16.32** | **0.38** |
| A | GART1@25 | 132 | 8048 | 61 | 435 | 9 | 0.318 | 2.0 | 0.000855 | 2.4 | 6 | 0.000779 | 2.3 | 15.74 | 0.37 | 0.000804 | 2.5 | **16.24** | **0.40** |
| A | GART1@26 | 139 | 9550 | 69 | 512 | 8 | 0.328 | 1.8 | 0.000866 | 2.1 | 6 | 0.000800 | 2.1 | 16.17 | 0.33 | 0.000816 | 2.1 | **16.49** | **0.35** |
| B | GART1@01 | 14 | 7467 | 551 | 446 | 9 | 0.626 | 2.3 | 0.000807 | 2.3 | 6 | 0.000737 | 2.3 | 14.89 | 0.34 | 0.000758 | 2.4 | **15.32** | **0.37** |
| B | GART1@02 | 16 | 11180 | 718 | 285 | 14 | 0.707 | 3.8 | 0.000774 | 2.3 | 7 | 0.000669 | 2.7 | 13.51 | 0.37 | 0.000716 | 2.4 | **14.47** | **0.35** |
| B | GART1@04 | 28 | 11198 | 401 | 967 | 11 | 0.299 | 3.0 | 0.000783 | 2.2 | 1 | 0.000752 | 2.2 | 15.19 | 0.33 | 0.000771 | 2.2 | **15.59** | **0.35** |
| B | GART1@05 | 31 | 13066 | 424 | 1149 | 11 | 0.232 | 3.4 | 0.000731 | 3.0 | 1 | 0.000707 | 2.9 | 14.28 | 0.42 | 0.000724 | 3.0 | **14.64** | **0.44** |
| B | GART1@17 | 86 | 22525 | 263 | 679 | 7 | 0.397 | 1.5 | 0.000854 | 1.8 | 4 | 0.000806 | 1.8 | 16.28 | 0.29 | 0.000820 | 1.8 | **16.56** | **0.30** |
| B | GART1@18 | 110 | 23948 | 218 | 894 | 7 | 0.307 | 1.7 | 0.000820 | 1.8 | 2 | 0.000784 | 1.8 | 15.84 | 0.28 | 0.000799 | 1.8 | **16.15** | **0.30** |
| B | GART1@19 | 14 | 7182 | 501 | 832 | 14 | 0.437 | 3.9 | 0.000778 | 2.2 | 2 | 0.000742 | 2.2 | 14.98 | 0.32 | 0.000763 | 2.2 | **15.42** | **0.34** |
| B | GART1@20 | 108 | 19460 | 180 | 740 | 7 | 0.321 | 1.8 | 0.000828 | 2.0 | 3 | 0.000785 | 2.0 | 15.86 | 0.31 | 0.000805 | 2.0 | **16.26** | **0.33** |
| B | GART1@21 | 106 | 23334 | 220 | 775 | 7 | 0.330 | 1.6 | 0.000839 | 2.0 | 3 | 0.000797 | 2.0 | 16.10 | 0.32 | 0.000816 | 2.0 | **16.48** | **0.34** |
| | *GART1@03* | *16* | *15175* | *927* | *53* | *3* | *0.802* | *0.8* | *0.001916* | *11.9* | *62* | *0.000516* | *4.1* | *10.43* | *0.43* | *0.000729* | *17.7* | *14.72* | *2.61* |
| | *GART1@04* | *19* | *13099* | *696* | *69* | *4* | *0.810* | *1.0* | *0.001303* | *7.8* | *48* | *0.000571* | *4.2* | *11.53* | *0.48* | *0.000679* | *12.9* | *13.72* | *1.76* |
| | *GART1@06* | *93* | *12315* | *133* | *2322* | *26* | *0.182* | *6.4* | *0.000453* | *4.0* | *1* | *0.000446* | *3.9* | *9.01* | *0.35* | *0.000450* | *4.0* | *9.09* | *0.36* |
| | NOWA3@01 | 99 | 7183 | 73 | 397 | 10 | 0.433 | 2.1 | 0.000894 | 4.3 | 9 | 0.000807 | 4.0 | 16.31 | 0.65 | 0.000809 | 4.3 | **16.35** | **0.71** |
| | NOWA3@02 | 111 | 7973 | 72 | 205 | 7 | 0.569 | 1.8 | 0.000864 | 3.8 | 15 | 0.000704 | 3.4 | 14.23 | 0.48 | 0.000733 | 3.8 | **14.80** | **0.57** |
| | NOWA3@03 | 95 | 10612 | 112 | 506 | 9 | 0.474 | 2.1 | 0.000852 | 3.8 | 7 | 0.000787 | 3.6 | 15.90 | 0.57 | 0.000793 | 3.8 | **16.03** | **0.61** |
| | NOWA3@04 | 93 | 10175 | 109 | 103 | 5 | 0.724 | 1.2 | 0.001132 | 5.8 | 37 | 0.000707 | 4.1 | 14.29 | 0.58 | 0.000713 | 5.9 | **14.40** | **0.84** |
| | NOWA3@05 | 107 | 6658 | 62 | 94 | 5 | 0.695 | 1.4 | 0.001145 | 5.8 | 38 | 0.000674 | 4.1 | 13.62 | 0.56 | 0.000711 | 5.9 | **14.37** | **0.84** |

| Group | Sample | | | | | | | | | | | | | | | | | |
|---|---|---|---|---|---|---|---|---|---|---|---|---|---|---|---|---|---|---|
| A | **NOWA3@06** | 146 | 7715 | 53 | 176 | 9 | 0.606 | 2.0 | 0.000875 | 5.6 | 22 | 0.000683 | 4.8 | 13.80 | 0.66 | 0.000681 | 5.6 | **13.75** | **0.77** |
| | **NOWA3@07** | 162 | 4216 | 26 | 137 | 7 | 0.461 | 1.6 | 0.001023 | 4.8 | 27 | 0.000734 | 4.0 | 14.84 | 0.59 | 0.000743 | 4.8 | **15.01** | **0.72** |
| | **NOWA3@09** | 187 | 2571 | 14 | 81 | 7 | 0.445 | 1.7 | 0.001077 | 5.9 | 36 | 0.000565 | 4.4 | 11.42 | 0.50 | 0.000686 | 5.9 | **13.86** | **0.82** |
| | **NOWA3@10** | 178 | 3925 | 22 | 124 | 7 | 0.478 | 1.5 | 0.001053 | 6.0 | 30 | 0.000726 | 4.6 | 14.68 | 0.68 | 0.000738 | 6.0 | **14.91** | **0.89** |
| | **NOWA3@11** | 106 | 5924 | 56 | 188 | 7 | 0.477 | 1.8 | 0.000922 | 4.7 | 17 | 0.000732 | 4.0 | 14.80 | 0.59 | 0.000761 | 4.7 | **15.38** | **0.72** |
| | **NOWA3@22** | 106 | 6470 | 61 | 1230 | 19 | 0.133 | 3.9 | 0.000837 | 1.2 | 2 | 0.000811 | 1.3 | 16.38 | 0.22 | 0.000824 | 1.2 | **16.65** | **0.20** |
| | **NOWA3@23** | 94 | 9038 | 96 | 1036 | 14 | 0.300 | 2.7 | 0.000864 | 0.9 | 4 | 0.000832 | 1.0 | 16.81 | 0.18 | 0.000834 | 0.9 | **16.85** | **0.16** |
| | **NOWA3@24** | 100 | 9053 | 90 | 896 | 14 | 0.265 | 3.1 | 0.000786 | 1.0 | 3 | 0.000752 | 1.2 | 15.19 | 0.18 | 0.000765 | 1.1 | **15.45** | **0.16** |
| | **NOWA3@25** | 91 | 7689 | 84 | 1010 | 15 | 0.176 | 3.6 | 0.000857 | 0.9 | 2 | 0.000824 | 1.0 | 16.65 | 0.17 | 0.000842 | 0.9 | **17.01** | **0.15** |
| B | **NOWA3@17** | 192 | 9482 | 50 | 555 | 10 | 0.290 | 1.7 | 0.000859 | 3.7 | 7 | 0.000799 | 3.5 | 16.15 | 0.57 | 0.000799 | 3.7 | **16.15** | **0.61** |
| | **NOWA3@18** | 193 | 8656 | 45 | 388 | 8 | 0.288 | 1.7 | 0.000838 | 3.8 | 8 | 0.000755 | 3.5 | 15.24 | 0.53 | 0.000774 | 3.8 | **15.64** | **0.59** |
| | **NOWA3@19** | 179 | 11023 | 62 | 531 | 9 | 0.286 | 1.7 | 0.000832 | 3.9 | 6 | 0.000771 | 3.7 | 15.58 | 0.57 | 0.000786 | 3.9 | **15.87** | **0.62** |
| | **NOWA3@20** | 189 | 10600 | 56 | 451 | 8 | 0.278 | 1.7 | 0.000836 | 3.9 | 6 | 0.000765 | 3.6 | 15.45 | 0.56 | 0.000786 | 3.9 | **15.87** | **0.62** |
| | **NOWA3@21** | 212 | 10750 | 51 | 225 | 13 | 0.368 | 2.5 | 0.000907 | 4.0 | 17 | 0.000752 | 4.0 | 15.18 | 0.60 | 0.000757 | 4.0 | **15.30** | **0.61** |
| C | **NOWA3@12** | 50 | 12537 | 253 | 347 | 7 | 0.547 | 1.8 | 0.000868 | 3.9 | 8 | 0.000771 | 3.6 | 15.58 | 0.56 | 0.000796 | 3.9 | **16.08** | **0.63** |
| | **NOWA3@13** | 34 | 9676 | 287 | 371 | 8 | 0.556 | 1.9 | 0.000839 | 3.8 | 8 | 0.000751 | 3.5 | 15.18 | 0.54 | 0.000773 | 3.8 | **15.62** | **0.60** |
| | **NOWA3@14** | 39 | 10246 | 260 | 323 | 8 | 0.534 | 2.0 | 0.000783 | 3.9 | 8 | 0.000689 | 3.6 | 13.93 | 0.50 | 0.000723 | 3.9 | **14.61** | **0.58** |
| | **NOWA3@15** | 48 | 11123 | 231 | 498 | 10 | 0.504 | 2.1 | 0.000777 | 3.8 | 7 | 0.000717 | 3.6 | 14.48 | 0.52 | 0.000726 | 3.8 | **14.66** | **0.56** |
| | **NOWA3@16** | 48 | 10718 | 225 | 349 | 9 | 0.590 | 2.0 | 0.000745 | 4.4 | 9 | 0.000663 | 4.0 | 13.39 | 0.53 | 0.000677 | 4.4 | **13.68** | **0.60** |
| | *NOWA3@08* | *212* | *3097* | *15* | *123* | *8* | *0.375* | *1.7* | *0.001111* | *6.8* | *29* | *0.000763* | *5.3* | *15.41* | *0.81* | *0.000788* | *6.9* | ***15.93*** | ***1.09*** |
| A | **GART3@13** | 85 | 8547 | 101 | 226 | 9 | 0.537 | 2.1 | 0.000901 | 2.6 | 14 | 0.000747 | 2.7 | 15.09 | 0.41 | 0.000775 | 2.6 | **15.65** | **0.41** |
| | **GART3@14** | 112 | 9936 | 89 | 207 | 10 | 0.541 | 2.4 | 0.000879 | 2.8 | 14 | 0.000715 | 3.0 | 14.46 | 0.43 | 0.000760 | 2.9 | **15.36** | **0.44** |
| | **GART3@15** | 95 | 9197 | 97 | 184 | 10 | 0.595 | 2.1 | 0.000969 | 3.9 | 19 | 0.000769 | 3.7 | 15.54 | 0.58 | 0.000781 | 3.9 | **15.78** | **0.62** |
| | **GART3@16** | 94 | 9295 | 99 | 183 | 9 | 0.585 | 1.9 | 0.000917 | 3.2 | 20 | 0.000723 | 3.2 | 14.61 | 0.47 | 0.000733 | 3.3 | **14.82** | **0.48** |
| | **GART3@17** | 159 | 9327 | 59 | 179 | 13 | 0.621 | 2.6 | 0.000881 | 2.9 | 25 | 0.000691 | 3.7 | 13.95 | 0.51 | 0.000662 | 3.0 | **13.38** | **0.40** |
| | **GART3@18** | 161 | 8380 | 52 | 128 | 11 | 0.623 | 2.6 | 0.000904 | 2.9 | 25 | 0.000638 | 4.0 | 12.90 | 0.51 | 0.000676 | 3.0 | **13.65** | **0.41** |
| | **GART3@07** | 251 | 11739 | 47 | 318 | 10 | 0.312 | 2.1 | 0.000865 | 2.8 | 8 | 0.000760 | 2.7 | 15.36 | 0.41 | 0.000794 | 2.8 | **16.04** | **0.44** |
| | **GART3@08** | 318 | 9593 | 30 | 219 | 11 | 0.374 | 2.3 | 0.000836 | 2.7 | 15 | 0.000689 | 3.0 | 13.92 | 0.41 | 0.000711 | 2.7 | **14.37** | **0.39** |
| | **GART3@09** | 335 | 8995 | 27 | 220 | 11 | 0.379 | 2.2 | 0.000858 | 2.7 | 17 | 0.000711 | 3.0 | 14.37 | 0.44 | 0.000714 | 2.8 | **14.43** | **0.40** |
| | **GART3@10** | 318 | 9819 | 31 | 221 | 11 | 0.377 | 2.2 | 0.000875 | 2.8 | 15 | 0.000722 | 3.0 | 14.58 | 0.44 | 0.000745 | 2.9 | **15.05** | **0.43** |

| | | | | | | | | | | | | | | | | | | |
|---|---|---|---|---|---|---|---|---|---|---|---|---|---|---|---|---|---|---|
| **B** | **GART3@11** | 387 | 10683 | 28 | 244 | 13 | 0.326 | 2.8 | 0.000829 | 2.7 | 11 | 0.000703 | 3.1 | 14.20 | 0.44 | 0.000738 | 2.7 | **14.92** | **0.41** |
| | **GART3@12** | 484 | 11366 | 24 | 210 | 13 | 0.388 | 3.1 | 0.000864 | 2.6 | 12 | 0.000712 | 3.3 | 14.38 | 0.47 | 0.000760 | 2.6 | **15.35** | **0.40** |
| | **GART3@19** | 129 | 5307 | 41 | 100 | 7 | 0.543 | 1.7 | 0.001103 | 4.2 | 31 | 0.000681 | 3.9 | 13.75 | 0.53 | 0.000756 | 4.3 | **15.27** | **0.65** |
| | **GART3@20** | 251 | 4358 | 17 | 122 | 10 | 0.408 | 2.1 | 0.000990 | 3.6 | 29 | 0.000676 | 4.1 | 13.65 | 0.56 | 0.000707 | 3.7 | **14.29** | **0.53** |
| | **GART3@21** | 247 | 13787 | 56 | 218 | 8 | 0.411 | 1.9 | 0.000901 | 2.7 | 12 | 0.000742 | 2.7 | 14.99 | 0.40 | 0.000797 | 2.7 | **16.09** | **0.44** |
| | **GART3@22** | 431 | 10631 | 25 | 215 | 13 | 0.466 | 2.5 | 0.000880 | 2.6 | 19 | 0.000734 | 3.3 | 14.83 | 0.49 | 0.000712 | 2.7 | **14.38** | **0.39** |
| | **GART3@23** | 240 | 12846 | 54 | 247 | 10 | 0.414 | 2.2 | 0.000879 | 2.9 | 12 | 0.000745 | 2.9 | 15.05 | 0.44 | 0.000773 | 2.9 | **15.61** | **0.45** |
| **C** | **GART3@01** | 114 | 7936 | 69 | 285 | 12 | 0.352 | 2.7 | 0.000762 | 2.6 | 10 | 0.000659 | 2.8 | 13.31 | 0.37 | 0.000689 | 2.6 | **13.92** | **0.36** |
| | **GART3@03** | 183 | 3983 | 22 | 100 | 13 | 0.392 | 3.3 | 0.000778 | 2.9 | 23 | 0.000477 | 5.4 | 9.65 | 0.52 | 0.000602 | 3.1 | **12.17** | **0.37** |
| | **GART3@04** | 176 | 3297 | 19 | 138 | 17 | 0.392 | 3.3 | 0.000802 | 2.9 | 26 | 0.000578 | 5.1 | 11.67 | 0.60 | 0.000595 | 3.1 | **12.02** | **0.37** |
| | **GART3@05** | 123 | 4784 | 39 | 191 | 14 | 0.393 | 2.9 | 0.000770 | 2.8 | 17 | 0.000614 | 3.6 | 12.41 | 0.44 | 0.000639 | 2.8 | **12.92** | **0.36** |
| | **GART3@06** | 166 | 4521 | 27 | 197 | 17 | 0.374 | 3.5 | 0.000763 | 2.6 | 17 | 0.000623 | 4.0 | 12.58 | 0.51 | 0.000630 | 2.7 | **12.72** | **0.34** |
| | *GART3@02* | *161* | *3479* | *22* | *109* | *12* | *0.449* | *2.6* | *0.000824* | *8.0* | *32* | *0.000531* | *6.8* | *10.73* | *0.73* | *0.000558* | *8.1* | *11.28* | *0.91* |
| | **STEI2@01** | 395 | 15567 | 39 | 1346 | 11 | 0.106 | 2.0 | 0.000873 | 2.12 | 2 | 0.000848 | 2.1 | 17.14 | 0.36 | 0.000859 | 2.1 | **17.36** | **0.37** |
| | **STEI2@02** | 403 | 16229 | 40 | 1096 | 10 | 0.113 | 2.0 | 0.000862 | 2.11 | 2 | 0.000833 | 2.1 | 16.82 | 0.35 | 0.000847 | 2.1 | **17.11** | **0.36** |
| | **STEI2@03** | 417 | 16643 | 40 | 1022 | 9 | 0.115 | 1.9 | 0.000869 | 2.04 | 2 | 0.000836 | 2.0 | 16.89 | 0.34 | 0.000854 | 2.0 | **17.25** | **0.35** |
| | **STEI2@04** | 376 | 17989 | 48 | 1436 | 11 | 0.097 | 2.2 | 0.000854 | 2.15 | 1 | 0.000831 | 2.1 | 16.79 | 0.35 | 0.000844 | 2.1 | **17.05** | **0.37** |
| | **STEI2@05** | 348 | 7706 | 22 | 844 | 13 | 0.101 | 2.2 | 0.000860 | 2.50 | 3 | 0.000820 | 2.5 | 16.57 | 0.41 | 0.000837 | 2.5 | **16.92** | **0.42** |
| | **STEI2@06** | 288 | 6287 | 22 | 700 | 13 | 0.125 | 2.3 | 0.000892 | 1.90 | 4 | 0.000843 | 1.9 | 17.02 | 0.33 | 0.000857 | 1.9 | **17.32** | **0.33** |
| | **STEI2@07** | 257 | 5814 | 23 | 556 | 12 | 0.176 | 2.0 | 0.000915 | 2.06 | 6 | 0.000851 | 2.1 | 17.20 | 0.36 | 0.000858 | 2.1 | **17.34** | **0.36** |
| | **STEI2@08** | 387 | 5851 | 15 | 730 | 13 | 0.090 | 2.3 | 0.000874 | 2.49 | 3 | 0.000830 | 2.5 | 16.76 | 0.41 | 0.000848 | 2.5 | **17.13** | **0.43** |
| | **STEI2@09** | 443 | 5143 | 12 | 649 | 14 | 0.080 | 2.3 | 0.000859 | 2.51 | 3 | 0.000810 | 2.5 | 16.37 | 0.41 | 0.000832 | 2.5 | **16.82** | **0.43** |
| | **STEI2@10** | 295 | 6041 | 20 | 689 | 14 | 0.118 | 2.4 | 0.000875 | 2.18 | 3 | 0.000828 | 2.2 | 16.73 | 0.37 | 0.000844 | 2.2 | **17.06** | **0.37** |
| | **STEI2@11** | 370 | 7478 | 20 | 1151 | 15 | 0.088 | 2.4 | 0.000858 | 2.11 | 2 | 0.000829 | 2.1 | 16.76 | 0.35 | 0.000841 | 2.1 | **16.99** | **0.36** |
| | **STEI2@12** | 548 | 11690 | 21 | 755 | 10 | 0.103 | 1.8 | 0.000884 | 1.96 | 3 | 0.000838 | 1.9 | 16.94 | 0.33 | 0.000859 | 2.0 | **17.36** | **0.34** |
| | **STEI2@13** | 579 | 10095 | 17 | 798 | 11 | 0.101 | 1.8 | 0.000871 | 2.26 | 3 | 0.000829 | 2.2 | 16.75 | 0.37 | 0.000844 | 2.3 | **17.06** | **0.39** |
| | **STEI2@14** | 601 | 10283 | 17 | 745 | 10 | 0.099 | 1.8 | 0.000869 | 2.15 | 3 | 0.000824 | 2.1 | 16.65 | 0.35 | 0.000843 | 2.2 | **17.03** | **0.37** |
| | **STEI2@15** | 352 | 19439 | 55 | 962 | 9 | 0.138 | 2.0 | 0.000864 | 1.83 | 2 | 0.000829 | 1.8 | 16.75 | 0.30 | 0.000848 | 1.8 | **17.14** | **0.31** |
| | **STEI2@16** | 455 | 10595 | 23 | 676 | 10 | 0.127 | 1.8 | 0.000891 | 2.06 | 3 | 0.000840 | 2.0 | 16.96 | 0.34 | 0.000861 | 2.1 | **17.39** | **0.36** |
| | **STEI2@17** | 387 | 8068 | 21 | 902 | 13 | 0.091 | 2.2 | 0.000864 | 2.03 | 2 | 0.000827 | 2.0 | 16.71 | 0.34 | 0.000846 | 2.0 | **17.10** | **0.35** |

| | | | | | | | | | | | | | | | | | | | |
|---|---|---|---|---|---|---|---|---|---|---|---|---|---|---|---|---|---|---|---|
| | STEI2@18 | 348 | 11391 | 33 | 1180 | 13 | 0.101 | 2.5 | 0.000846 | 2.09 | 1 | 0.000819 | 2.1 | 16.55 | 0.34 | 0.000835 | 2.1 | **16.87** | **0.35** |
| | STEI2@19 | 339 | 12077 | 36 | 1048 | 11 | 0.095 | 2.3 | 0.000876 | 2.20 | 1 | 0.000844 | 2.2 | 17.05 | 0.37 | 0.000864 | 2.2 | **17.45** | **0.39** |
| | STEI2@20 | 323 | 7488 | 23 | 943 | 13 | 0.084 | 2.6 | 0.000871 | 2.04 | 1 | 0.000835 | 2.0 | 16.87 | 0.34 | 0.000858 | 2.0 | **17.33** | **0.35** |
| | KNOR1@19 | 149 | 12269 | 82 | 466 | 7 | 0.392 | 3.5 | 0.000558 | 2.34 | 4 | 0.000527 | 2.3 | 10.64 | 0.24 | 0.000535 | 2.3 | **10.81** | **0.25** |
| | KNOR1@20 | 181 | 9307 | 52 | 285 | 7 | 0.488 | 3.2 | 0.000587 | 3.28 | 11 | 0.000515 | 3.1 | 10.41 | 0.32 | 0.000522 | 3.3 | **10.55** | **0.35** |
| A | KNOR1@21 | 239 | 9337 | 39 | 642 | 13 | 0.400 | 5.1 | 0.000604 | 3.67 | 5 | 0.000574 | 3.6 | 11.60 | 0.42 | 0.000572 | 3.7 | **11.56** | **0.42** |
| | KNOR1@22 | 172 | 8289 | 48 | 512 | 9 | 0.321 | 4.1 | 0.000570 | 2.76 | 4 | 0.000537 | 2.7 | 10.86 | 0.29 | 0.000545 | 2.8 | **11.01** | **0.30** |
| | KNOR1@24 | 194 | 7199 | 37 | 605 | 13 | 0.268 | 5.3 | 0.000549 | 3.18 | 4 | 0.000517 | 3.1 | 10.44 | 0.32 | 0.000529 | 3.2 | **10.68** | **0.34** |
| | KNOR1@01 | 32 | 7767 | 246 | 189 | 5 | 0.737 | 2.5 | 0.000615 | 2.18 | 18 | 0.000498 | 2.1 | 10.06 | 0.21 | 0.000505 | 2.2 | **10.21** | **0.23** |
| | KNOR1@02 | 35 | 6107 | 175 | 197 | 6 | 0.731 | 2.9 | 0.000637 | 2.60 | 18 | 0.000521 | 2.5 | 10.53 | 0.26 | 0.000524 | 2.6 | **10.59** | **0.28** |
| | KNOR1@03 | 36 | 9885 | 273 | 115 | 4 | 0.784 | 1.8 | 0.000767 | 2.66 | 32 | 0.000518 | 2.3 | 10.48 | 0.24 | 0.000518 | 2.7 | **10.47** | **0.29** |
| B | KNOR1@05 | 27 | 9535 | 356 | 102 | 3 | 0.786 | 1.4 | 0.000869 | 2.35 | 36 | 0.000560 | 1.9 | 11.32 | 0.22 | 0.000558 | 2.4 | **11.28** | **0.27** |
| | KNOR1@06 | 33 | 9203 | 277 | 167 | 4 | 0.734 | 2.1 | 0.000661 | 2.18 | 19 | 0.000522 | 2.0 | 10.54 | 0.21 | 0.000533 | 2.2 | **10.77** | **0.24** |
| | KNOR1@07 | 33 | 4983 | 152 | 112 | 5 | 0.726 | 2.1 | 0.000768 | 2.20 | 31 | 0.000522 | 2.2 | 10.56 | 0.23 | 0.000527 | 2.3 | **10.64** | **0.25** |
| | KNOR1@08 | 36 | 3783 | 105 | 136 | 6 | 0.652 | 2.8 | 0.000701 | 2.17 | 24 | 0.000516 | 2.4 | 10.42 | 0.25 | 0.000533 | 2.3 | **10.76** | **0.24** |
| | KNOR1@09 | 39 | 5846 | 149 | 281 | 7 | 0.573 | 3.4 | 0.000577 | 2.30 | 11 | 0.000513 | 2.3 | 10.37 | 0.24 | 0.000514 | 2.3 | **10.38** | **0.24** |
| | KNOR1@04 | 30 | 6902 | 234 | 95 | 3 | 0.759 | 1.6 | 0.000920 | 2.21 | 37 | 0.000555 | 1.9 | 11.21 | 0.22 | 0.000576 | 2.3 | **11.63** | **0.27** |
| | KNOR1@10 | 48 | 5633 | 118 | 428 | 11 | 0.495 | 5.2 | 0.000542 | 2.37 | 6 | 0.000499 | 2.4 | 10.07 | 0.24 | 0.000508 | 2.4 | **10.27** | **0.24** |
| | KNOR1@11 | 48 | 4611 | 96 | 444 | 12 | 0.478 | 5.6 | 0.000560 | 2.48 | 6 | 0.000525 | 2.6 | 10.61 | 0.27 | 0.000528 | 2.5 | **10.67** | **0.27** |
| | KNOR1@12 | 57 | 4583 | 81 | 399 | 11 | 0.355 | 5.3 | 0.000540 | 2.18 | 5 | 0.000498 | 2.3 | 10.06 | 0.23 | 0.000513 | 2.2 | **10.36** | **0.23** |
| C | KNOR1@13 | 54 | 4104 | 77 | 445 | 11 | 0.331 | 5.0 | 0.000541 | 2.29 | 5 | 0.000500 | 2.4 | 10.10 | 0.24 | 0.000513 | 2.3 | **10.37** | **0.24** |
| | KNOR1@14 | 56 | 6093 | 108 | 729 | 13 | 0.285 | 5.6 | 0.000516 | 2.19 | 3 | 0.000495 | 2.2 | 10.00 | 0.22 | 0.000501 | 2.2 | **10.12** | **0.22** |
| | KNOR1@15 | 76 | 6847 | 91 | 639 | 11 | 0.237 | 5.4 | 0.000529 | 2.19 | 2 | 0.000508 | 2.2 | 10.27 | 0.23 | 0.000515 | 2.2 | **10.41** | **0.23** |
| | KNOR1@16 | 65 | 5872 | 91 | 524 | 11 | 0.224 | 5.8 | 0.000529 | 2.17 | 2 | 0.000503 | 2.2 | 10.17 | 0.23 | 0.000517 | 2.2 | **10.45** | **0.23** |
| | KNOR1@18 | 77 | 8867 | 114 | 628 | 10 | 0.141 | 5.7 | 0.000559 | 2.52 | 1 | 0.000544 | 2.6 | 11.00 | 0.28 | 0.000552 | 2.5 | **11.15** | **0.28** |
| | KNOR1@17 | 73 | 6281 | 86 | 585 | 10 | 0.191 | 5.4 | 0.000590 | 2.24 | 2 | 0.000564 | 2.3 | 11.41 | 0.26 | 0.000578 | 2.2 | **11.68** | **0.26** |

*Eastern Tauern Window*

| | KAIS6@29 | 12 | 32285 | 2726 | 324 | 2 | 0.816 | 1.1 | 0.001135 | 2.68 | 11 | 0.001014 | 2.4 | 20.48 | 0.49 | 0.001015 | 2.7 | **20.51** | **0.55** |

| Group | ID | | | | | | | | | | | | | | | | | |
|---|---|---|---|---|---|---|---|---|---|---|---|---|---|---|---|---|---|---|
| A | KAIS6@37 | 10 | 35152 | 3587 | 322 | 2 | 0.817 | 1.1 | 0.001192 | 2.71 | 10 | 0.001065 | 2.4 | 21.51 | 0.52 | 0.001072 | 2.7 | **21.66** | **0.59** |
| | KAIS6@38 | 10 | 35481 | 3480 | 428 | 3 | 0.797 | 1.4 | 0.001137 | 2.74 | 8 | 0.001046 | 2.5 | 21.13 | 0.54 | 0.001045 | 2.8 | **21.12** | **0.58** |
| | KAIS6@39 | 10 | 31600 | 3326 | 316 | 3 | 0.820 | 1.3 | 0.001154 | 2.73 | 11 | 0.001022 | 2.4 | 20.65 | 0.50 | 0.001030 | 2.7 | **20.81** | **0.57** |
| | KAIS6@41 | 8 | 32642 | 4280 | 351 | 2 | 0.824 | 1.2 | 0.001173 | 2.74 | 9 | 0.001057 | 2.5 | 21.36 | 0.53 | 0.001063 | 2.8 | **21.48** | **0.59** |
| | KAIS6@42 | 6 | 29873 | 4677 | 292 | 2 | 0.836 | 1.2 | 0.001200 | 2.69 | 11 | 0.001054 | 2.4 | 21.28 | 0.51 | 0.001066 | 2.7 | **21.54** | **0.58** |
| B | KAIS6@15 | 5 | 21644 | 4568 | 142 | 2 | 0.833 | 0.9 | 0.001399 | 2.69 | 25 | 0.001031 | 2.0 | 20.83 | 0.43 | 0.001045 | 2.8 | **21.12** | **0.59** |
| | KAIS6@16 | 5 | 22052 | 4771 | 161 | 2 | 0.831 | 0.9 | 0.001329 | 2.69 | 23 | 0.001020 | 2.1 | 20.61 | 0.44 | 0.001022 | 2.7 | **20.65** | **0.57** |
| | KAIS6@17 | 5 | 25100 | 4752 | 189 | 2 | 0.825 | 1.0 | 0.001299 | 2.73 | 19 | 0.001046 | 2.2 | 21.12 | 0.47 | 0.001049 | 2.8 | **21.20** | **0.59** |
| | KAIS6@18 | 5 | 25260 | 4905 | 148 | 2 | 0.827 | 0.9 | 0.001392 | 2.81 | 26 | 0.001040 | 2.2 | 21.01 | 0.45 | 0.001025 | 2.9 | **20.71** | **0.61** |
| | KAIS6@19 | 5 | 23495 | 5173 | 148 | 2 | 0.825 | 0.9 | 0.001390 | 2.75 | 24 | 0.001038 | 2.1 | 20.96 | 0.44 | 0.001053 | 2.8 | **21.28** | **0.60** |
| | KAIS6@20 | 5 | 24260 | 5377 | 157 | 2 | 0.829 | 0.9 | 0.001351 | 2.83 | 24 | 0.001032 | 2.2 | 20.85 | 0.46 | 0.001033 | 2.9 | **20.86** | **0.61** |
| | KAIS6@21 | 5 | 25950 | 5040 | 155 | 2 | 0.837 | 0.9 | 0.001320 | 2.73 | 24 | 0.001003 | 2.1 | 20.27 | 0.43 | 0.001007 | 2.8 | **20.35** | **0.57** |
| | KAIS6@22 | 4 | 20175 | 4625 | 134 | 2 | 0.813 | 0.9 | 0.001424 | 2.76 | 27 | 0.001034 | 2.1 | 20.88 | 0.43 | 0.001043 | 2.8 | **21.07** | **0.60** |
| | KAIS6@23 | 3 | 18427 | 5349 | 153 | 2 | 0.828 | 1.0 | 0.001353 | 2.82 | 24 | 0.001028 | 2.2 | 20.76 | 0.46 | 0.001027 | 2.9 | **20.74** | **0.60** |
| | KAIS6@24 | 4 | 17328 | 4822 | 178 | 2 | 0.830 | 1.1 | 0.001313 | 2.69 | 21 | 0.001045 | 2.2 | 21.11 | 0.46 | 0.001041 | 2.7 | **21.02** | **0.58** |
| | KAIS6@25 | 5 | 17901 | 3588 | 154 | 2 | 0.833 | 0.9 | 0.001366 | 2.77 | 24 | 0.001042 | 2.2 | 21.05 | 0.46 | 0.001039 | 2.8 | **20.98** | **0.59** |
| | KAIS6@26 | 4 | 19477 | 5281 | 167 | 2 | 0.820 | 1.0 | 0.001304 | 2.75 | 22 | 0.001020 | 2.2 | 20.60 | 0.46 | 0.001017 | 2.8 | **20.56** | **0.58** |
| | KAIS6@27 | 4 | 19595 | 5290 | 165 | 2 | 0.821 | 1.1 | 0.001325 | 2.72 | 22 | 0.001030 | 2.2 | 20.80 | 0.45 | 0.001036 | 2.8 | **20.93** | **0.58** |
| | KAIS6@28 | 3 | 17683 | 5368 | 162 | 2 | 0.817 | 1.0 | 0.001354 | 2.76 | 23 | 0.001043 | 2.2 | 21.06 | 0.46 | 0.001041 | 2.8 | **21.02** | **0.59** |
| | KAIS6@31 | 3 | 19464 | 6314 | 207 | 2 | 0.826 | 1.2 | 0.001275 | 2.67 | 17 | 0.001049 | 2.2 | 21.19 | 0.48 | 0.001059 | 2.7 | **21.39** | **0.58** |
| | KAIS6@32 | 3 | 17623 | 5729 | 145 | 2 | 0.823 | 1.0 | 0.001352 | 2.75 | 26 | 0.001007 | 2.1 | 20.35 | 0.43 | 0.001003 | 2.8 | **20.26** | **0.57** |
| | KAIS6@33 | 4 | 23433 | 5395 | 191 | 2 | 0.819 | 1.0 | 0.001257 | 2.79 | 18 | 0.001018 | 2.3 | 20.56 | 0.47 | 0.001026 | 2.8 | **20.73** | **0.59** |
| | KAIS6@34 | 3 | 22517 | 6923 | 175 | 2 | 0.836 | 0.9 | 0.001289 | 2.73 | 21 | 0.001016 | 2.2 | 20.52 | 0.45 | 0.001015 | 2.8 | **20.51** | **0.57** |
| | KAIS6@35 | 4 | 23426 | 6129 | 175 | 2 | 0.827 | 0.9 | 0.001299 | 2.72 | 21 | 0.001025 | 2.2 | 20.70 | 0.45 | 0.001031 | 2.8 | **20.82** | **0.58** |
| | KAIS6@36 | 4 | 24762 | 6924 | 169 | 2 | 0.821 | 0.9 | 0.001316 | 2.76 | 21 | 0.001031 | 2.2 | 20.83 | 0.46 | 0.001034 | 2.8 | **20.89** | **0.59** |
| | KAIS6@43 | 4 | 19511 | 4948 | 229 | 2 | 0.828 | 1.2 | 0.001210 | 2.76 | 15 | 0.001024 | 2.4 | 20.68 | 0.49 | 0.001023 | 2.8 | **20.66** | **0.58** |
| | KAIS6@44 | 5 | 20543 | 3995 | 252 | 3 | 0.828 | 1.5 | 0.001177 | 2.72 | 14 | 0.001007 | 2.4 | 20.35 | 0.48 | 0.001009 | 2.8 | **20.39** | **0.56** |
| | KAIS6@45 | 4 | 17366 | 4418 | 225 | 3 | 0.834 | 1.4 | 0.001269 | 2.67 | 15 | 0.001067 | 2.3 | 21.55 | 0.49 | 0.001076 | 2.7 | **21.73** | **0.59** |
| | KAIS6@46 | 4 | 14595 | 3614 | 231 | 3 | 0.813 | 1.5 | 0.001284 | 2.70 | 15 | 0.001083 | 2.3 | 21.88 | 0.51 | 0.001094 | 2.7 | **22.11** | **0.61** |
| | KAIS6@01 | 2 | 13078 | 6002 | 83 | 2 | 0.844 | 0.8 | 0.001865 | 2.70 | 46 | 0.001011 | 1.7 | 20.43 | 0.34 | 0.001001 | 3.1 | **20.21** | **0.63** |
| | KAIS6@02 | 3 | 14140 | 4835 | 97 | 2 | 0.823 | 0.9 | 0.001682 | 2.76 | 40 | 0.001031 | 1.9 | 20.83 | 0.39 | 0.001012 | 3.0 | **20.44** | **0.62** |

| | | | | | | | | | | | | | | | | | | |
|---|---|---|---|---|---|---|---|---|---|---|---|---|---|---|---|---|---|---|
| C | **KAIS6@05** | 3 | 10157 | 2957 | 96 | 2 | 0.837 | 1.0 | 0.001672 | 2.69 | 38 | 0.001010 | 1.9 | 20.41 | 0.38 | 0.001035 | 3.0 | **20.91** | **0.63** |
| | **KAIS6@06** | 4 | 15227 | 3566 | 104 | 2 | 0.828 | 1.0 | 0.001564 | 2.84 | 36 | 0.000995 | 2.0 | 20.11 | 0.40 | 0.000994 | 3.1 | **20.09** | **0.63** |
| | **KAIS6@09** | 3 | 10348 | 3946 | 74 | 1 | 0.828 | 0.7 | 0.002146 | 2.71 | 52 | 0.001030 | 1.5 | 20.81 | 0.32 | 0.001037 | 3.7 | **20.94** | **0.77** |
| | **KAIS6@10** | 3 | 12624 | 4224 | 90 | 2 | 0.828 | 0.9 | 0.001795 | 2.74 | 42 | 0.001037 | 1.8 | 20.96 | 0.37 | 0.001033 | 3.1 | **20.87** | **0.64** |
| | **KAIS6@12** | 3 | 10457 | 3171 | 91 | 2 | 0.822 | 1.0 | 0.001752 | 2.68 | 41 | 0.001020 | 1.8 | 20.61 | 0.37 | 0.001039 | 3.0 | **20.98** | **0.63** |
| | **KAIS6@03** | 4 | 15947 | 3922 | 100 | 2 | 0.844 | 1.0 | 0.001524 | 2.91 | 39 | 0.000944 | 2.0 | 19.07 | 0.38 | 0.000924 | 3.2 | **18.67** | **0.59** |
| | **KAIS6@04** | 4 | 16156 | 3905 | 91 | 2 | 0.829 | 1.0 | 0.001531 | 3.25 | 40 | 0.000890 | 2.1 | 17.98 | 0.38 | 0.000923 | 3.5 | **18.65** | **0.66** |
| | **KAIS6@07** | 6 | 11981 | 2002 | 89 | 4 | 0.843 | 1.6 | 0.001562 | 2.89 | 44 | 0.000887 | 2.3 | 17.93 | 0.40 | 0.000877 | 3.7 | **17.71** | **0.65** |
| | **KAIS6@08** | 3 | 10466 | 3145 | 87 | 2 | 0.824 | 0.8 | 0.001711 | 2.68 | 44 | 0.000959 | 1.7 | 19.37 | 0.33 | 0.000956 | 3.1 | **19.31** | **0.59** |
| D | **KAIS6@11** | 4 | 13094 | 3210 | 89 | 2 | 0.830 | 1.1 | 0.001493 | 2.92 | 42 | 0.000846 | 2.0 | 17.09 | 0.33 | 0.000870 | 3.3 | **17.58** | **0.58** |
| | **KAIS6@13** | 4 | 13271 | 3273 | 88 | 2 | 0.842 | 1.0 | 0.001621 | 2.84 | 41 | 0.000917 | 1.9 | 18.52 | 0.35 | 0.000950 | 3.3 | **19.19** | **0.63** |
| | **KAIS6@14** | 9 | 13459 | 1568 | 105 | 4 | 0.827 | 1.7 | 0.001323 | 3.57 | 33 | 0.000836 | 2.6 | 16.89 | 0.44 | 0.000881 | 3.8 | **17.79** | **0.68** |
| | **KAIS6@30** | 14 | 25376 | 1840 | 225 | 2 | 0.837 | 1.1 | 0.001130 | 2.68 | 16 | 0.000950 | 2.3 | 19.20 | 0.44 | 0.000951 | 2.7 | **19.22** | **0.52** |
| | **KAIS6@40** | 10 | 32326 | 3141 | 301 | 3 | 0.800 | 1.4 | 0.001094 | 2.93 | 12 | 0.000962 | 2.6 | 19.44 | 0.51 | 0.000967 | 3.0 | **19.53** | **0.58** |
| | **KAIS6@47** | 9 | 31411 | 3692 | 311 | 3 | 0.818 | 1.3 | 0.001069 | 2.68 | 11 | 0.000949 | 2.4 | 19.16 | 0.46 | 0.000955 | 2.7 | **19.28** | **0.52** |
| | *KAIS6@48* | *13* | *18728* | *1403* | *217* | *3* | *0.836* | *1.5* | *0.001218* | *2.67* | *17* | *0.001014* | *2.3* | *20.49* | *0.47* | *0.001014* | *2.7* | *20.48* | *0.56* |
| | *KAIS6@49* | *13* | *18241* | *1449* | *248* | *3* | *0.813* | *1.3* | *0.001252* | *2.69* | *14* | *0.001072* | *2.3* | *21.66* | *0.51* | *0.001083* | *2.7* | *21.87* | *0.59* |
| | **SALZ18@01** | 11 | 443 | 40 | 428 | 6 | 0.642 | 2.9 | 0.001001 | 2.30 | 8 | 0.000928 | 2.2 | 18.75 | 0.41 | 0.000921 | 2.31 | **18.61** | **0.43** |
| | **SALZ18@02** | 19 | 11907 | 639 | 515 | 6 | 0.546 | 3.0 | 0.000970 | 2.16 | 4 | 0.000914 | 2.1 | 18.46 | 0.39 | 0.000928 | 2.17 | **18.75** | **0.41** |
| | **SALZ18@03** | 23 | 12452 | 552 | 1291 | 11 | 0.191 | 7.3 | 0.000859 | 2.25 | 1 | 0.000853 | 2.3 | 17.23 | 0.39 | 0.000854 | 2.25 | **17.25** | **0.39** |
| | **SALZ18@04** | 20 | 8334 | 407 | 800 | 10 | 0.246 | 6.5 | 0.000943 | 2.52 | 1 | 0.000909 | 2.5 | 18.36 | 0.46 | 0.000933 | 2.52 | **18.84** | **0.47** |
| | **SALZ18@05** | 28 | 6912 | 245 | 1239 | 12 | 0.235 | 6.6 | 0.000911 | 2.20 | 1 | 0.000903 | 2.2 | 18.24 | 0.41 | 0.000902 | 2.20 | **18.22** | **0.40** |
| | **SALZ18@06** | 11 | 2256 | 200 | 1200 | 13 | 0.314 | 7.5 | 0.000947 | 2.26 | 1 | 0.000941 | 2.3 | 19.00 | 0.44 | 0.000935 | 2.26 | **18.90** | **0.43** |
| | **SALZ18@07** | 23 | 9797 | 429 | 1862 | 19 | 0.208 | 9.6 | 0.000863 | 2.34 | 1 | 0.000858 | 2.4 | 17.33 | 0.41 | 0.000857 | 2.34 | **17.31** | **0.40** |
| | **SALZ18@08** | 20 | 4607 | 232 | 980 | 14 | 0.186 | 10.0 | 0.000880 | 2.39 | 1 | 0.000856 | 2.4 | 17.29 | 0.42 | 0.000874 | 2.39 | **17.66** | **0.42** |
| | **SALZ18@09** | 18 | 3952 | 224 | 1353 | 18 | 0.180 | 9.3 | 0.000942 | 2.42 | 1 | 0.000934 | 2.5 | 18.87 | 0.47 | 0.000934 | 2.42 | **18.88** | **0.46** |
| | **SALZ18@10** | 19 | 3681 | 189 | 894 | 18 | 0.178 | 10.2 | 0.000917 | 2.57 | 1 | 0.000889 | 2.6 | 17.96 | 0.47 | 0.000907 | 2.57 | **18.33** | **0.47** |
| | **SALZ18@11** | 13 | 1751 | 134 | 1123 | 15 | 0.312 | 7.3 | 0.000979 | 2.38 | 2 | 0.000962 | 2.4 | 19.44 | 0.47 | 0.000963 | 2.38 | **19.46** | **0.46** |
| | **SALZ18@12** | 17 | 4555 | 267 | 731 | 19 | 0.169 | 10.8 | 0.000922 | 2.74 | 1 | 0.000881 | 2.8 | 17.80 | 0.50 | 0.000910 | 2.74 | **18.39** | **0.50** |
| | **SALZ18@13** | 17 | 1752 | 106 | 617 | 27 | 0.207 | 11.1 | 0.000887 | 3.43 | 3 | 0.000850 | 3.8 | 17.18 | 0.65 | 0.000860 | 3.43 | **17.37** | **0.60** |

| | | | | | | | | | | | | | | | | | | |
|---|---|---|---|---|---|---|---|---|---|---|---|---|---|---|---|---|---|---|
| **SALZ18@14** | 22 | 1037 | 48 | 975 | 11 | 0.270 | 6.1 | 0.000953 | 2.24 | 1 | 0.000938 | 2.3 | 18.95 | 0.43 | 0.000941 | 2.24 | **19.01** | **0.43** |
| **SALZ18@15** | 37 | 9684 | 262 | 1073 | 14 | 0.224 | 7.5 | 0.000792 | 2.37 | 1 | 0.000781 | 2.4 | 15.79 | 0.38 | 0.000783 | 2.37 | **15.81** | **0.37** |
| | | | | | | | | | | | | | | | | | | |
| **LOHN4@01** | 14 | 14552 | 1076 | 79 | 1 | 0.826 | 0.7 | 0.001981 | 2.96 | 47 | 0.001031 | 1.7 | 20.84 | 0.35 | 0.001059 | 3.57 | **21.39** | **0.76** |
| **LOHN4@02** | 39 | 25804 | 662 | 272 | 3 | 0.779 | 1.4 | 0.001112 | 3.21 | 13 | 0.000964 | 2.8 | 19.47 | 0.55 | 0.000969 | 3.24 | **19.58** | **0.63** |
| **LOHN4@03** | 11 | 12883 | 1175 | 151 | 2 | 0.805 | 1.2 | 0.001356 | 2.67 | 24 | 0.001026 | 2.1 | 20.73 | 0.44 | 0.001033 | 2.76 | **20.86** | **0.58** |
| **LOHN4@04** | 16 | 16561 | 1017 | 173 | 2 | 0.820 | 1.2 | 0.001332 | 2.87 | 22 | 0.001051 | 2.3 | 21.23 | 0.49 | 0.001045 | 2.96 | **21.12** | **0.63** |
| **LOHN4@07** | 34 | 35456 | 1048 | 338 | 2 | 0.779 | 1.1 | 0.001218 | 2.68 | 10 | 0.001094 | 2.4 | 22.11 | 0.53 | 0.001097 | 2.69 | **22.16** | **0.60** |
| **LOHN4@08** | 31 | 32345 | 1044 | 312 | 2 | 0.782 | 1.0 | 0.001208 | 2.75 | 11 | 0.001074 | 2.5 | 21.70 | 0.53 | 0.001070 | 2.77 | **21.62** | **0.60** |
| **LOHN4@09** | 33 | 34180 | 1040 | 323 | 2 | 0.790 | 1.1 | 0.001224 | 2.70 | 11 | 0.001094 | 2.4 | 22.09 | 0.54 | 0.001092 | 2.71 | **22.07** | **0.60** |
| **LOHN4@10** | 33 | 35690 | 1072 | 349 | 2 | 0.788 | 1.1 | 0.001171 | 2.68 | 10 | 0.001058 | 2.4 | 21.38 | 0.52 | 0.001060 | 2.69 | **21.41** | **0.57** |
| **LOHN4@11** | 34 | 29955 | 888 | 355 | 3 | 0.796 | 1.3 | 0.001085 | 2.67 | 10 | 0.000977 | 2.4 | 19.75 | 0.48 | 0.000977 | 2.68 | **19.74** | **0.53** |
| **LOHN4@12** | 27 | 29155 | 1065 | 366 | 3 | 0.776 | 1.3 | 0.001185 | 2.68 | 9 | 0.001078 | 2.5 | 21.77 | 0.53 | 0.001082 | 2.69 | **21.86** | **0.59** |
| **LOHN4@13** | 26 | 23964 | 935 | 97 | 2 | 0.823 | 0.7 | 0.001750 | 2.99 | 39 | 0.001071 | 1.9 | 21.64 | 0.42 | 0.001066 | 3.32 | **21.54** | **0.72** |
| **LOHN4@14** | 28 | 22365 | 792 | 156 | 2 | 0.806 | 1.0 | 0.001418 | 2.83 | 23 | 0.001087 | 2.2 | 21.97 | 0.49 | 0.001089 | 2.96 | **22.01** | **0.65** |
| **LOHN4@15** | 24 | 20864 | 866 | 523 | 4 | 0.742 | 1.9 | 0.001109 | 2.69 | 6 | 0.001040 | 2.5 | 21.00 | 0.53 | 0.001044 | 2.70 | **21.09** | **0.57** |
| **LOHN4@16** | 30 | 23937 | 797 | 575 | 4 | 0.728 | 2.2 | 0.001062 | 2.75 | 5 | 0.001003 | 2.6 | 20.25 | 0.53 | 0.001006 | 2.76 | **20.33** | **0.56** |
| **LOHN4@19** | 27 | 21599 | 807 | 299 | 3 | 0.788 | 1.3 | 0.001221 | 2.71 | 12 | 0.001075 | 2.4 | 21.71 | 0.52 | 0.001080 | 2.72 | **21.82** | **0.59** |
| **LOHN4@20** | 29 | 23422 | 801 | 350 | 3 | 0.785 | 1.4 | 0.001156 | 2.67 | 10 | 0.001042 | 2.4 | 21.06 | 0.51 | 0.001041 | 2.68 | **21.02** | **0.56** |
| **LOHN4@21** | 16 | 17205 | 1046 | 171 | 2 | 0.798 | 1.0 | 0.001354 | 2.70 | 22 | 0.001064 | 2.2 | 21.49 | 0.47 | 0.001060 | 2.76 | **21.41** | **0.59** |
| **LOHN4@22** | 17 | 16983 | 991 | 165 | 2 | 0.810 | 1.1 | 0.001359 | 2.76 | 22 | 0.001055 | 2.2 | 21.31 | 0.47 | 0.001057 | 2.82 | **21.35** | **0.60** |
| **LOHN4@23** | 16 | 17269 | 1091 | 162 | 2 | 0.819 | 1.0 | 0.001388 | 2.79 | 23 | 0.001074 | 2.2 | 21.70 | 0.48 | 0.001065 | 2.85 | **21.52** | **0.61** |
| **LOHN4@24** | 20 | 21768 | 1108 | 176 | 2 | 0.810 | 1.1 | 0.001258 | 2.87 | 22 | 0.000990 | 2.3 | 20.00 | 0.46 | 0.000985 | 2.93 | **19.90** | **0.58** |
| **LOHN4@25** | 18 | 15361 | 875 | 174 | 3 | 0.823 | 1.2 | 0.001232 | 2.68 | 21 | 0.000969 | 2.2 | 19.58 | 0.43 | 0.000979 | 2.74 | **19.77** | **0.54** |
| **LOHN4@26** | 15 | 17857 | 1172 | 172 | 2 | 0.806 | 1.1 | 0.001276 | 2.68 | 21 | 0.001001 | 2.2 | 20.21 | 0.44 | 0.001012 | 2.73 | **20.44** | **0.56** |
| **LOHN4@27** | 27 | 26405 | 974 | 184 | 2 | 0.819 | 0.9 | 0.001317 | 2.72 | 19 | 0.001056 | 2.2 | 21.34 | 0.47 | 0.001064 | 2.76 | **21.50** | **0.59** |
| **LOHN4@28** | 28 | 27181 | 955 | 190 | 2 | 0.816 | 0.9 | 0.001324 | 2.80 | 20 | 0.001068 | 2.3 | 21.58 | 0.49 | 0.001066 | 2.84 | **21.53** | **0.61** |
| **LOHN4@29** | 27 | 25484 | 955 | 191 | 2 | 0.809 | 1.0 | 0.001304 | 2.85 | 20 | 0.001051 | 2.3 | 21.22 | 0.50 | 0.001048 | 2.90 | **21.18** | **0.61** |
| **LOHN4@30** | 27 | 25099 | 915 | 190 | 2 | 0.798 | 0.9 | 0.001322 | 2.71 | 19 | 0.001067 | 2.2 | 21.55 | 0.48 | 0.001068 | 2.75 | **21.58** | **0.59** |
| **LOHN4@31** | 30 | 25226 | 855 | 433 | 3 | 0.777 | 1.6 | 0.001235 | 2.67 | 8 | 0.001139 | 2.5 | 23.00 | 0.57 | 0.001134 | 2.68 | **22.92** | **0.61** |
| **LOHN4@32** | 36 | 21812 | 606 | 287 | 4 | 0.797 | 1.7 | 0.001212 | 2.70 | 13 | 0.001061 | 2.4 | 21.43 | 0.52 | 0.001053 | 2.73 | **21.27** | **0.58** |

A

| | | | | | | | | | | | | | | | | | | |
|---|---|---|---|---|---|---|---|---|---|---|---|---|---|---|---|---|---|---|
| | LOHN4@33 | 27 | 23182 | 856 | 199 | 2 | 0.805 | 1.0 | 0.001233 | 2.67 | 18 | 0.001002 | 2.2 | 20.24 | 0.45 | 0.001011 | 2.71 | **20.42** | **0.55** |
| | LOHN4@34 | 26 | 24807 | 948 | 188 | 2 | 0.812 | 1.0 | 0.001293 | 2.71 | 19 | 0.001043 | 2.2 | 21.06 | 0.47 | 0.001044 | 2.75 | **21.09** | **0.58** |
| | LOHN4@35 | 25 | 23076 | 925 | 174 | 2 | 0.812 | 0.9 | 0.001327 | 3.07 | 21 | 0.001045 | 2.5 | 21.11 | 0.52 | 0.001052 | 3.14 | **21.26** | **0.67** |
| | LOHN4@36 | 26 | 25326 | 972 | 190 | 2 | 0.786 | 0.9 | 0.001308 | 2.73 | 19 | 0.001052 | 2.2 | 21.25 | 0.47 | 0.001061 | 2.77 | **21.43** | **0.59** |
| | LOHN4@37 | 27 | 26268 | 980 | 210 | 2 | 0.802 | 1.0 | 0.001282 | 2.78 | 17 | 0.001062 | 2.3 | 21.45 | 0.50 | 0.001061 | 2.81 | **21.43** | **0.60** |
| | LOHN4@38 | 27 | 27571 | 1016 | 218 | 2 | 0.812 | 1.0 | 0.001239 | 2.72 | 16 | 0.001033 | 2.3 | 20.87 | 0.48 | 0.001040 | 2.75 | **21.02** | **0.58** |
| | LOHN4@39 | 26 | 25481 | 988 | 207 | 2 | 0.806 | 0.9 | 0.001272 | 2.69 | 18 | 0.001047 | 2.2 | 21.16 | 0.48 | 0.001044 | 2.72 | **21.08** | **0.57** |
| | LOHN4@40 | 26 | 26144 | 1007 | 211 | 2 | 0.814 | 1.0 | 0.001245 | 2.78 | 17 | 0.001029 | 2.3 | 20.78 | 0.48 | 0.001032 | 2.81 | **20.85** | **0.59** |
| | LOHN4@41 | 19 | 24124 | 1248 | 221 | 2 | 0.800 | 1.0 | 0.001271 | 2.72 | 17 | 0.001061 | 2.3 | 21.44 | 0.49 | 0.001057 | 2.75 | **21.35** | **0.59** |
| | LOHN4@42 | 19 | 23602 | 1235 | 226 | 2 | 0.816 | 1.1 | 0.001279 | 2.74 | 16 | 0.001078 | 2.3 | 21.78 | 0.51 | 0.001076 | 2.77 | **21.73** | **0.60** |
| | LOHN4@43 | 21 | 23764 | 1157 | 236 | 2 | 0.802 | 1.1 | 0.001247 | 2.70 | 15 | 0.001057 | 2.3 | 21.36 | 0.49 | 0.001059 | 2.72 | **21.40** | **0.58** |
| | LOHN4@44 | 20 | 23194 | 1138 | 231 | 2 | 0.803 | 1.1 | 0.001254 | 2.72 | 15 | 0.001055 | 2.3 | 21.32 | 0.49 | 0.001067 | 2.74 | **21.56** | **0.59** |
| | LOHN4@45 | 20 | 23831 | 1188 | 232 | 2 | 0.787 | 1.0 | 0.001255 | 2.70 | 15 | 0.001058 | 2.3 | 21.37 | 0.49 | 0.001065 | 2.72 | **21.51** | **0.58** |
| | LOHN4@46 | 20 | 20913 | 1027 | 204 | 2 | 0.813 | 1.0 | 0.001277 | 2.67 | 17 | 0.001047 | 2.2 | 21.16 | 0.47 | 0.001058 | 2.70 | **21.38** | **0.58** |
| | LOHN4@48 | 22 | 23727 | 1093 | 262 | 2 | 0.808 | 1.1 | 0.001210 | 2.78 | 14 | 0.001045 | 2.4 | 21.12 | 0.51 | 0.001046 | 2.80 | **21.13** | **0.59** |
| | LOHN4@49 | 22 | 23546 | 1065 | 259 | 2 | 0.789 | 1.1 | 0.001179 | 2.73 | 13 | 0.001018 | 2.4 | 20.56 | 0.49 | 0.001022 | 2.75 | **20.65** | **0.57** |
| | LOHN4@50 | 21 | 22193 | 1049 | 260 | 2 | 0.806 | 1.2 | 0.001208 | 2.70 | 13 | 0.001045 | 2.4 | 21.12 | 0.50 | 0.001047 | 2.72 | **21.15** | **0.57** |
| | LOHN4@51 | 20 | 22103 | 1121 | 259 | 2 | 0.800 | 1.2 | 0.001179 | 2.67 | 14 | 0.001014 | 2.3 | 20.48 | 0.48 | 0.001011 | 2.69 | **20.43** | **0.55** |
| | LOHN4@52 | 20 | 20091 | 1023 | 242 | 2 | 0.785 | 1.2 | 0.001146 | 2.68 | 14 | 0.000973 | 2.3 | 19.66 | 0.45 | 0.000987 | 2.70 | **19.93** | **0.54** |
| | LOHN4@53 | 10 | 15092 | 1539 | 136 | 2 | 0.818 | 1.0 | 0.001422 | 2.69 | 28 | 0.001029 | 2.0 | 20.80 | 0.42 | 0.001030 | 2.79 | **20.80** | **0.58** |
| | LOHN4@54 | 10 | 13028 | 1330 | 137 | 2 | 0.812 | 1.1 | 0.001350 | 2.66 | 26 | 0.000979 | 2.0 | 19.78 | 0.40 | 0.000993 | 2.78 | **20.07** | **0.56** |
| | LOHN4@55 | 10 | 14821 | 1515 | 137 | 2 | 0.806 | 1.0 | 0.001355 | 2.67 | 27 | 0.000982 | 2.0 | 19.83 | 0.40 | 0.000983 | 2.78 | **19.85** | **0.55** |
| | LOHN4@05 | 19 | 14990 | 787 | 171 | 4 | 0.821 | 1.6 | 0.001129 | 3.06 | 22 | 0.000882 | 2.5 | 17.83 | 0.45 | 0.000875 | 3.15 | **17.68** | **0.56** |
| | LOHN4@06 | 31 | 9191 | 298 | 146 | 5 | 0.794 | 2.3 | 0.001237 | 3.80 | 26 | 0.000913 | 3.1 | 18.45 | 0.58 | 0.000912 | 3.95 | **18.42** | **0.73** |
| | LOHN4@17 | 32 | 23011 | 717 | 714 | 6 | 0.732 | 2.8 | 0.000980 | 3.29 | 5 | 0.000936 | 3.2 | 18.91 | 0.60 | 0.000933 | 3.29 | **18.85** | **0.62** |
| B | LOHN4@18 | 39 | 23334 | 591 | 461 | 5 | 0.760 | 2.3 | 0.001003 | 3.06 | 8 | 0.000927 | 2.9 | 18.73 | 0.54 | 0.000923 | 3.07 | **18.66** | **0.57** |
| | LOHN4@47 | 23 | 19054 | 816 | 222 | 3 | 0.808 | 1.2 | 0.001120 | 2.66 | 15 | 0.000933 | 2.3 | 18.85 | 0.43 | 0.000946 | 2.70 | **19.12** | **0.52** |
| | LOHN4@56 | 9 | 11815 | 1254 | 129 | 2 | 0.833 | 1.2 | 0.001272 | 2.78 | 28 | 0.000895 | 2.1 | 18.07 | 0.38 | 0.000913 | 2.90 | **18.45** | **0.53** |
| | LOHN4@57 | 13 | 11550 | 862 | 133 | 3 | 0.826 | 1.5 | 0.001194 | 3.44 | 28 | 0.000847 | 2.6 | 17.12 | 0.45 | 0.000855 | 3.58 | **17.28** | **0.62** |
| | ORT1@01 | 367 | 5825 | 16 | 1263 | 17 | 0.084 | 2.3 | 0.000945 | 2.48 | 2 | 0.000912 | 2.5 | 18.43 | 0.46 | 0.000922 | 2.5 | **18.63** | **0.47** |

| | | | | | | | | | | | | | | | | | | |
|---|---|---|---|---|---|---|---|---|---|---|---|---|---|---|---|---|---|---|
| **ORT1@02** | 316 | 4437 | 14 | 711 | 23 | 0.087 | 2.4 | 0.000956 | 2.43 | 3 | 0.000905 | 2.4 | 18.27 | 0.44 | 0.000930 | 2.5 | **18.80** | **0.46** |
| **ORT1@03** | 277 | 4950 | 18 | 1166 | 16 | 0.097 | 2.5 | 0.000936 | 2.26 | 3 | 0.000896 | 2.3 | 18.10 | 0.41 | 0.000911 | 2.3 | **18.41** | **0.42** |
| **ORT1@04** | 356 | 4831 | 14 | 926 | 16 | 0.088 | 2.4 | 0.000905 | 2.92 | 3 | 0.000861 | 2.9 | 17.40 | 0.50 | 0.000877 | 3.0 | **17.73** | **0.53** |
| **ORT1@05** | 161 | 5336 | 33 | 930 | 18 | 0.139 | 2.6 | 0.000970 | 2.17 | 3 | 0.000924 | 2.2 | 18.67 | 0.41 | 0.000941 | 2.2 | **19.01** | **0.42** |
| **ORT1@06** | 222 | 3688 | 17 | 382 | 16 | 0.166 | 2.1 | 0.001012 | 2.99 | 7 | 0.000897 | 3.0 | 18.12 | 0.54 | 0.000938 | 3.2 | **18.95** | **0.61** |
| **ORT1@07** | 219 | 5005 | 23 | 277 | 13 | 0.272 | 1.6 | 0.001039 | 2.23 | 13 | 0.000893 | 2.3 | 18.05 | 0.41 | 0.000902 | 2.4 | **18.21** | **0.43** |
| **ORT1@08** | 189 | 5091 | 27 | 422 | 18 | 0.220 | 2.0 | 0.000979 | 2.96 | 8 | 0.000896 | 2.9 | 18.09 | 0.52 | 0.000901 | 3.1 | **18.21** | **0.56** |
| **ORT1@09** | 262 | 6320 | 24 | 591 | 17 | 0.139 | 2.3 | 0.000880 | 3.78 | 4 | 0.000824 | 3.6 | 16.65 | 0.60 | 0.000841 | 3.8 | **16.98** | **0.65** |
| **ORT1@10** | 199 | 7415 | 37 | 214 | 11 | 0.411 | 1.2 | 0.001086 | 2.24 | 16 | 0.000892 | 2.1 | 18.01 | 0.38 | 0.000912 | 2.4 | **18.43** | **0.44** |
| **ORT1@11** | 101 | 3205 | 32 | 655 | 17 | 0.172 | 3.0 | 0.000981 | 2.61 | 4 | 0.000926 | 2.7 | 18.71 | 0.50 | 0.000941 | 2.7 | **19.01** | **0.51** |
| **ORT1@12** | 106 | 3602 | 34 | 661 | 28 | 0.191 | 2.9 | 0.000933 | 3.58 | 4 | 0.000863 | 3.5 | 17.44 | 0.60 | 0.000895 | 3.6 | **18.08** | **0.66** |
| **ORT1@13** | 95 | 4969 | 52 | 607 | 18 | 0.195 | 2.9 | 0.000910 | 2.82 | 3 | 0.000855 | 2.8 | 17.28 | 0.48 | 0.000883 | 2.9 | **17.83** | **0.51** |
| **ORT1@14** | 96 | 3780 | 40 | 628 | 13 | 0.183 | 2.9 | 0.000960 | 2.17 | 4 | 0.000884 | 2.3 | 17.86 | 0.40 | 0.000923 | 2.2 | **18.65** | **0.42** |
| *ORT1@15* | *105* | *4304* | *41* | *445* | *21* | *0.370* | *1.9* | *0.001022* | *4.60* | *12* | *0.000894* | *4.2* | *18.06* | *0.76* | *0.000903* | *5.7* | *18.24* | *1.04* |
| | | | | | | | | | | | | | | | | | | |
| **EUKL2@01** | 9 | 296 | 32 | 160 | 3 | 0.803 | 1.2 | 0.001359 | 2.12 | 22 | 0.001043 | 1.7 | 21.07 | 0.37 | 0.001063 | 2.21 | **21.47** | **0.47** |
| **EUKL2@02** | 14 | 301 | 22 | 209 | 3 | 0.803 | 1.2 | 0.001317 | 2.51 | 18 | 0.001089 | 2.1 | 22.00 | 0.47 | 0.001080 | 2.56 | **21.81** | **0.56** |
| **EUKL2@03** | 13 | 21097 | 1580 | 205 | 2 | 0.824 | 1.2 | 0.001303 | 2.25 | 17 | 0.001072 | 1.9 | 21.65 | 0.41 | 0.001078 | 2.30 | **21.77** | **0.50** |
| **EUKL2@04** | 14 | 18352 | 1313 | 231 | 3 | 0.812 | 1.5 | 0.001253 | 2.08 | 16 | 0.001059 | 1.8 | 21.39 | 0.39 | 0.001048 | 2.13 | **21.16** | **0.45** |
| **EUKL2@05** | 33 | 52673 | 1620 | 345 | 3 | 0.782 | 1.3 | 0.001194 | 2.17 | 10 | 0.001076 | 2.0 | 21.74 | 0.43 | 0.001079 | 2.18 | **21.79** | **0.48** |
| **EUKL2@06** | 19 | 25562 | 1351 | 218 | 2 | 0.795 | 1.2 | 0.001315 | 2.10 | 16 | 0.001097 | 1.8 | 22.17 | 0.40 | 0.001102 | 2.14 | **22.26** | **0.48** |
| **EUKL2@07** | 19 | 16767 | 868 | 234 | 3 | 0.800 | 1.2 | 0.001282 | 2.10 | 15 | 0.001085 | 1.8 | 21.91 | 0.40 | 0.001085 | 2.14 | **21.92** | **0.47** |
| | | | | | | | | | | | | | | | | | | |
| **HOAR1@27** | 148 | 1591 | 11 | 235 | 13 | 0.195 | 2.2 | 0.001164 | 1.5 | 14 | 0.000973 | 2.4 | 19.65 | 0.48 | 0.001003 | 1.6 | **20.26** | **0.32** |
| **HOAR1@28** | 100 | 1383 | 14 | 149 | 11 | 0.248 | 2.2 | 0.001269 | 1.2 | 19 | 0.000940 | 2.9 | 19.00 | 0.55 | 0.001031 | 1.3 | **20.84** | **0.28** |
| **HOAR1@29** | 95 | 19676 | 208 | 1211 | 8 | 0.201 | 2.5 | 0.001023 | 0.8 | 1 | 0.000990 | 0.9 | 20.00 | 0.17 | 0.001013 | 0.8 | **20.47** | **0.17** |
| **HOAR1@30** | 80 | 650 | 8 | 214 | 19 | 0.170 | 3.3 | 0.001153 | 3.6 | 15 | 0.000946 | 4.6 | 19.10 | 0.88 | 0.000978 | 3.7 | **19.76** | **0.74** |
| **HOAR1@31** | 124 | 2725 | 22 | 497 | 14 | 0.169 | 2.6 | 0.001070 | 0.9 | 6 | 0.000987 | 1.4 | 19.94 | 0.28 | 0.001007 | 0.9 | **20.35** | **0.19** |
| **HOAR1@32** | 316 | 1076 | 3 | 160 | 12 | 0.132 | 2.0 | 0.001240 | 3.4 | 21 | 0.000950 | 4.0 | 19.20 | 0.76 | 0.000985 | 3.5 | **19.91** | **0.70** |
| **HOAR1@01** | 50 | 926 | 18 | 322 | 20 | 0.244 | 3.7 | 0.001121 | 4.3 | 9 | 0.000999 | 4.6 | 20.18 | 0.92 | 0.001017 | 4.4 | **20.55** | **0.91** |
| **HOAR1@02** | 51 | 782 | 15 | 200 | 18 | 0.296 | 3.5 | 0.001128 | 3.5 | 15 | 0.000911 | 4.5 | 18.40 | 0.83 | 0.000954 | 4.5 | **19.27** | **0.87** |

A

| | | | | | | | | | | | | | | | | | | |
|---|---|---|---|---|---|---|---|---|---|---|---|---|---|---|---|---|---|---|
| HOAR1@03 | 48 | 969 | 20 | 250 | 17 | 0.205 | 3.5 | 0.001088 | 4.7 | 8 | 0.000920 | 4.8 | 18.59 | 0.89 | 0.000997 | 5.0 | **20.13** | **1.00** |
| HOAR1@04 | 71 | 1075 | 15 | 259 | 18 | 0.182 | 3.5 | 0.001075 | 4.3 | 9 | 0.000926 | 4.6 | 18.70 | 0.85 | 0.000982 | 4.5 | **19.84** | **0.89** |
| HOAR1@05 | 79 | 643 | 8 | 233 | 21 | 0.197 | 3.4 | 0.001118 | 4.2 | 16 | 0.000932 | 4.9 | 18.84 | 0.92 | 0.000940 | 5.0 | **18.98** | **0.94** |
| HOAR1@06 | 118 | 1676 | 14 | 330 | 15 | 0.148 | 2.9 | 0.001046 | 2.8 | 7 | 0.000924 | 3.1 | 18.66 | 0.57 | 0.000977 | 3.0 | **19.73** | **0.59** |
| HOAR1@07 | 106 | 1764 | 17 | 448 | 17 | 0.162 | 2.9 | 0.001065 | 2.6 | 7 | 0.000973 | 2.8 | 19.66 | 0.55 | 0.000993 | 2.8 | **20.07** | **0.56** |
| HOAR1@08 | 67 | 1358 | 20 | 339 | 18 | 0.237 | 3.2 | 0.001049 | 2.8 | 11 | 0.000938 | 3.3 | 18.94 | 0.62 | 0.000938 | 3.0 | **18.95** | **0.57** |
| HOAR1@09 | 62 | 1579 | 25 | 364 | 16 | 0.191 | 3.3 | 0.001110 | 3.0 | 5 | 0.000992 | 3.1 | 20.04 | 0.63 | 0.001050 | 3.2 | **21.20** | **0.67** |
| HOAR1@10 | 266 | 1995 | 8 | 433 | 15 | 0.097 | 2.4 | 0.001076 | 2.4 | 5 | 0.000980 | 2.6 | 19.79 | 0.52 | 0.001019 | 2.6 | **20.59** | **0.54** |
| HOAR1@11 | 56 | 6609 | 117 | 714 | 11 | 0.242 | 2.7 | 0.001032 | 2.2 | 3 | 0.000976 | 2.2 | 19.71 | 0.43 | 0.001005 | 2.2 | **20.31** | **0.45** |
| HOAR1@12 | 104 | 2748 | 26 | 263 | 10 | 0.359 | 1.8 | 0.001163 | 2.3 | 15 | 0.000992 | 2.4 | 20.04 | 0.49 | 0.000988 | 2.4 | **19.96** | **0.48** |
| HOAR1@13 | 103 | 3259 | 32 | 679 | 15 | 0.171 | 2.7 | 0.000998 | 3.0 | 4 | 0.000941 | 3.0 | 19.01 | 0.56 | 0.000959 | 3.0 | **19.37** | **0.59** |
| HOAR1@14 | 107 | 1940 | 18 | 272 | 12 | 0.213 | 2.3 | 0.001115 | 3.9 | 10 | 0.000957 | 3.7 | 19.33 | 0.72 | 0.001005 | 4.4 | **20.30** | **0.88** |
| HOAR1@16 | 145 | 2711 | 19 | 632 | 16 | 0.139 | 2.6 | 0.001076 | 2.5 | 4 | 0.001010 | 2.5 | 20.41 | 0.51 | 0.001029 | 2.5 | **20.78** | **0.53** |
| HOAR1@17 | 72 | 1305 | 18 | 263 | 15 | 0.211 | 3.1 | 0.001059 | 3.7 | 8 | 0.000904 | 3.9 | 18.26 | 0.71 | 0.000969 | 3.9 | **19.58** | **0.77** |
| HOAR1@18 | 74 | 1381 | 19 | 292 | 15 | 0.196 | 3.1 | 0.001088 | 4.2 | 8 | 0.000944 | 4.2 | 19.07 | 0.79 | 0.000998 | 4.4 | **20.16** | **0.89** |
| HOAR1@19 | 60 | 1129 | 19 | 389 | 19 | 0.240 | 3.0 | 0.001129 | 3.7 | 10 | 0.001026 | 3.9 | 20.73 | 0.81 | 0.001019 | 4.0 | **20.58** | **0.82** |
| HOAR1@20 | 60 | 1818 | 30 | 499 | 17 | 0.206 | 3.2 | 0.001035 | 2.6 | 5 | 0.000955 | 2.7 | 19.29 | 0.53 | 0.000978 | 2.8 | **19.76** | **0.55** |
| HOAR1@21 | 45 | 1000 | 22 | 183 | 14 | 0.309 | 3.0 | 0.001130 | 3.8 | 15 | 0.000891 | 4.2 | 18.01 | 0.76 | 0.000964 | 4.2 | **19.47** | **0.82** |
| HOAR1@22 | 49 | 1540 | 31 | 194 | 12 | 0.305 | 2.8 | 0.001127 | 2.5 | 9 | 0.000909 | 3.1 | 18.37 | 0.57 | 0.001025 | 2.9 | **20.70** | **0.59** |
| HOAR1@23 | 126 | 1838 | 15 | 824 | 24 | 0.119 | 3.4 | 0.001015 | 3.2 | 5 | 0.000968 | 3.2 | 19.55 | 0.63 | 0.000966 | 3.3 | **19.52** | **0.64** |
| HOAR1@24 | 85 | 1927 | 23 | 511 | 18 | 0.147 | 3.3 | 0.001026 | 2.6 | 5 | 0.000954 | 2.8 | 19.27 | 0.53 | 0.000974 | 2.8 | **19.67** | **0.54** |
| HOAR1@25 | 53 | 1330 | 25 | 384 | 19 | 0.230 | 3.5 | 0.001022 | 2.9 | 8 | 0.000919 | 3.2 | 18.57 | 0.60 | 0.000944 | 3.0 | **19.08** | **0.58** |
| HOAR1@26 | 60 | 1206 | 20 | 343 | 18 | 0.197 | 3.5 | 0.001023 | 3.1 | 8 | 0.000908 | 3.5 | 18.34 | 0.63 | 0.000940 | 3.3 | **18.99** | **0.63** |
| *HOAR1@15* | *111* | *2877* | *26* | *112* | *6* | *0.516* | *1.2* | *0.001353* | *5.1* | *29* | *0.000885* | *3.9* | *17.88* | *0.70* | *0.000964* | *7.9* | *19.47* | *1.53* |
| MOKR1@07 | 307 | 3840 | 13 | 172 | 11 | 0.228 | 2.1 | 0.001121 | 3.2 | 18 | 0.000876 | 3.6 | 17.69 | 0.64 | 0.000915 | 3.3 | **18.48** | **0.61** |
| MOKR1@08 | 393 | 5762 | 15 | 259 | 15 | 0.228 | 2.8 | 0.000997 | 2.7 | 13 | 0.000848 | 3.2 | 17.13 | 0.55 | 0.000869 | 2.8 | **17.55** | **0.48** |
| MOKR1@09 | 916 | 5949 | 6 | 260 | 14 | 0.128 | 2.3 | 0.000993 | 2.6 | 12 | 0.000845 | 3.1 | 17.08 | 0.52 | 0.000879 | 2.7 | **17.75** | **0.48** |
| MOKR1@10 | 814 | 2999 | 4 | 166 | 13 | 0.134 | 2.0 | 0.001143 | 3.7 | 22 | 0.000878 | 4.1 | 17.74 | 0.73 | 0.000890 | 3.8 | **17.99** | **0.68** |
| MOKR1@11 | 1852 | 2971 | 2 | 148 | 21 | 0.138 | 3.2 | 0.001168 | 2.9 | 26 | 0.000976 | 6.4 | 19.71 | 1.25 | 0.000859 | 3.3 | **17.36** | **0.57** |
| MOKR1@12 | 702 | 7328 | 10 | 245 | 16 | 0.224 | 3.1 | 0.001029 | 2.6 | 13 | 0.000867 | 3.4 | 17.51 | 0.59 | 0.000899 | 2.6 | **18.15** | **0.47** |

B

| Group | Sample | | | | | | | | | | | | | | | | | | |
|---|---|---|---|---|---|---|---|---|---|---|---|---|---|---|---|---|---|---|---|
| A | MOKR1@20 | 442 | 2417 | 5 | 255 | 47 | 0.062 | 10.0 | 0.000754 | 1.0 | 5 | 0.000639 | 7.2 | 12.92 | 0.94 | 0.000714 | 1.4 | **14.42** | **0.21** |
| | MOKR1@21 | 473 | 5279 | 11 | 1029 | 55 | 0.063 | 10.2 | 0.001134 | 1.0 | 1 | 0.001091 | 2.3 | 22.05 | 0.50 | 0.001124 | 1.1 | **22.70** | **0.24** |
| | MOKR1@22 | 410 | 5545 | 14 | 2865 | 95 | 0.070 | 11.4 | 0.001130 | 1.6 | 1 | 0.001115 | 2.1 | 22.52 | 0.46 | 0.001119 | 1.7 | **22.61** | **0.37** |
| | MOKR1@23 | 341 | 2318 | 7 | 483 | 47 | 0.104 | 6.6 | 0.001093 | 1.3 | 8 | 0.001049 | 4.4 | 21.18 | 0.94 | 0.001011 | 1.6 | **20.43** | **0.32** |
| | MOKR1@24 | 507 | 7589 | 15 | 3668 | 67 | 0.057 | 6.1 | 0.001049 | 1.5 | 1 | 0.001038 | 1.7 | 20.97 | 0.35 | 0.001042 | 1.5 | **21.06** | **0.32** |
| | MOKR1@25 | 336 | 4029 | 12 | 1546 | 67 | 0.087 | 7.7 | 0.001041 | 1.5 | 3 | 0.001015 | 2.2 | 20.51 | 0.46 | 0.001011 | 1.6 | **20.43** | **0.32** |
| | MOKR1@26 | 220 | 4084 | 19 | 646 | 39 | 0.101 | 6.7 | 0.001072 | 1.3 | 4 | 0.001008 | 2.6 | 20.37 | 0.53 | 0.001034 | 1.3 | **20.89** | **0.28** |
| | MOKR1@27 | 295 | 3072 | 10 | 626 | 55 | 0.108 | 8.3 | 0.001021 | 1.3 | 5 | 0.000958 | 3.6 | 19.35 | 0.69 | 0.000969 | 1.4 | **19.58** | **0.28** |
| | MOKR1@28 | 395 | 5188 | 13 | 1611 | 67 | 0.067 | 9.2 | 0.001011 | 1.1 | 1 | 0.000987 | 1.9 | 19.93 | 0.39 | 0.000997 | 1.2 | **20.15** | **0.23** |
| | MOKR1@29 | 469 | 3095 | 7 | 846 | 55 | 0.066 | 6.5 | 0.001059 | 1.0 | 3 | 0.001011 | 2.7 | 20.42 | 0.55 | 0.001025 | 1.2 | **20.72** | **0.25** |
| | MOKR1@30 | 688 | 4284 | 6 | 802 | 55 | 0.062 | 7.1 | 0.000922 | 1.2 | 3 | 0.000878 | 2.9 | 17.74 | 0.51 | 0.000893 | 1.4 | **18.03** | **0.25** |
| | MOKR1@31 | 683 | 4946 | 7 | 2982 | 95 | 0.056 | 7.6 | 0.001030 | 1.0 | 1 | 0.001017 | 1.6 | 20.54 | 0.33 | 0.001015 | 1.1 | **20.51** | **0.23** |
| B | MOKR1@01 | 132 | 23439 | 178 | 573 | 9 | 0.324 | 2.3 | 0.000991 | 2.6 | 3 | 0.000924 | 2.5 | 18.68 | 0.46 | 0.000959 | 2.6 | **19.37** | **0.50** |
| | MOKR1@02 | 167 | 21412 | 128 | 789 | 12 | 0.283 | 2.7 | 0.001001 | 2.6 | 3 | 0.000952 | 2.5 | 19.23 | 0.49 | 0.000970 | 2.6 | **19.60** | **0.51** |
| | MOKR1@03 | 149 | 20039 | 134 | 694 | 12 | 0.307 | 2.7 | 0.001011 | 2.6 | 3 | 0.000955 | 2.6 | 19.29 | 0.49 | 0.000977 | 2.6 | **19.73** | **0.52** |
| | MOKR1@04 | 162 | 15315 | 95 | 570 | 12 | 0.296 | 2.6 | 0.001007 | 2.7 | 5 | 0.000943 | 2.6 | 19.05 | 0.50 | 0.000960 | 2.7 | **19.40** | **0.51** |
| | MOKR1@05 | 182 | 11017 | 61 | 391 | 12 | 0.299 | 2.6 | 0.000978 | 2.6 | 7 | 0.000882 | 2.6 | 17.81 | 0.47 | 0.000913 | 2.6 | **18.45** | **0.48** |
| | MOKR1@06 | 239 | 10746 | 45 | 355 | 12 | 0.242 | 2.6 | 0.000993 | 2.6 | 6 | 0.000885 | 2.6 | 17.88 | 0.47 | 0.000932 | 2.6 | **18.83** | **0.48** |
| | MOKR1@13 | 127 | 16222 | 128 | 541 | 10 | 0.343 | 2.3 | 0.001014 | 2.6 | 5 | 0.000942 | 2.5 | 19.03 | 0.47 | 0.000967 | 2.6 | **19.53** | **0.50** |
| | MOKR1@14 | 363 | 17133 | 47 | 608 | 12 | 0.200 | 2.4 | 0.000949 | 2.6 | 4 | 0.000889 | 2.5 | 17.95 | 0.45 | 0.000908 | 2.6 | **18.35** | **0.47** |
| | MOKR1@15 | 446 | 16751 | 38 | 468 | 11 | 0.191 | 2.5 | 0.000948 | 2.6 | 4 | 0.000869 | 2.5 | 17.57 | 0.45 | 0.000906 | 2.6 | **18.30** | **0.47** |
| | MOKR1@16 | 517 | 16572 | 32 | 462 | 12 | 0.190 | 2.6 | 0.000943 | 2.6 | 5 | 0.000864 | 2.6 | 17.46 | 0.45 | 0.000899 | 2.6 | **18.16** | **0.47** |
| | MOKR1@17 | 141 | 16023 | 113 | 717 | 13 | 0.323 | 2.6 | 0.001031 | 2.6 | 4 | 0.000975 | 2.6 | 19.70 | 0.50 | 0.000986 | 2.6 | **19.92** | **0.52** |
| | MOKR1@18 | 303 | 8214 | 27 | 380 | 18 | 0.303 | 3.2 | 0.000949 | 2.7 | 10 | 0.000859 | 3.0 | 17.35 | 0.53 | 0.000851 | 2.7 | **17.19** | **0.46** |
| | *MOKR1@19* | *263* | *4051* | *15* | */* | *0* | *0.072* | *7.2* | *0.001023* | *1.1* | *2* | *0.001023* | */* | *20.67* | */* | *0.001002* | *1.1* | *20.24* | *0.23* |
| | SAND1@01 | 259 | 3167 | 12 | 125 | 17 | 0.337 | 3.8 | 0.001114 | 4.0 | 21 | 0.000770 | 5.9 | 15.56 | 0.92 | 0.000877 | 4.1 | **17.72** | **0.73** |
| | SAND1@02 | 252 | 3570 | 14 | 175 | 21 | 0.430 | 3.9 | 0.001104 | 3.1 | 24 | 0.000860 | 5.2 | 17.38 | 0.90 | 0.000843 | 3.3 | **17.03** | **0.56** |
| | SAND1@04 | 323 | 5540 | 17 | 184 | 19 | 0.441 | 4.0 | 0.001150 | 2.6 | 17 | 0.000908 | 4.4 | 18.35 | 0.80 | 0.000958 | 2.7 | **19.34** | **0.51** |
| | SAND1@05 | 580 | 5757 | 10 | 203 | 19 | 0.331 | 3.8 | 0.001106 | 2.8 | 15 | 0.000896 | 4.2 | 18.10 | 0.76 | 0.000938 | 2.9 | **18.96** | **0.54** |
| | SAND1@06 | 601 | 5846 | 10 | 216 | 20 | 0.328 | 4.0 | 0.001123 | 2.7 | 15 | 0.000923 | 4.2 | 18.64 | 0.77 | 0.000959 | 2.8 | **19.38** | **0.54** |

| | Sample | | | | | | | | | | | | | | | | | |
|---|---|---|---|---|---|---|---|---|---|---|---|---|---|---|---|---|---|---|---|
| A | **SAND1@07** | 272 | 5768 | 21 | 212 | 12 | 0.274 | 2.6 | 0.001006 | 2.6 | 12 | 0.000829 | 3.1 | 16.74 | 0.52 | 0.000883 | 2.6 | **17.85** | **0.47** |
| | **SAND1@08** | 251 | 6655 | 27 | 332 | 15 | 0.277 | 2.9 | 0.000972 | 2.6 | 10 | 0.000859 | 2.9 | 17.35 | 0.50 | 0.000879 | 2.6 | **17.76** | **0.46** |
| | **SAND1@21** | 181 | 7714 | 43 | 2354 | 55 | 0.087 | 8.6 | 0.001049 | 1.3 | 1 | 0.001032 | 1.6 | 20.85 | 0.33 | 0.001039 | 1.3 | **20.99** | **0.28** |
| | **SAND1@22** | 285 | 6564 | 23 | 1186 | 55 | 0.096 | 10.8 | 0.001045 | 1.4 | 1 | 0.001011 | 2.3 | 20.43 | 0.46 | 0.001030 | 1.4 | **20.80** | **0.30** |
| | **SAND1@24** | 289 | 6472 | 22 | 1758 | 67 | 0.098 | 10.4 | 0.001054 | 1.2 | 2 | 0.001031 | 1.9 | 20.82 | 0.39 | 0.001036 | 1.2 | **20.93** | **0.26** |
| | **SAND1@25** | 302 | 7743 | 26 | 2074 | 67 | 0.107 | 10.3 | 0.001107 | 1.1 | 1 | 0.001086 | 1.6 | 21.94 | 0.36 | 0.001090 | 1.1 | **22.03** | **0.25** |
| | **SAND1@26** | 302 | 7711 | 26 | 1385 | 55 | 0.140 | 8.6 | 0.001117 | 1.4 | 3 | 0.001086 | 2.1 | 21.94 | 0.45 | 0.001089 | 1.4 | **22.01** | **0.32** |
| | **SAND1@28** | 468 | 728 | 2 | 513 | 95 | 0.066 | 8.6 | 0.001116 | 1.4 | 10 | 0.001032 | 7.3 | 20.85 | 1.51 | 0.001004 | 3.4 | **20.28** | **0.68** |
| B | **SAND1@09** | 483 | 2027 | 4 | 137 | 14 | 0.176 | 2.2 | 0.001145 | 4.2 | 28 | 0.000822 | 4.9 | 16.61 | 0.81 | 0.000820 | 4.4 | **16.56** | **0.72** |
| | **SAND1@10** | 646 | 2420 | 4 | 109 | 14 | 0.176 | 2.7 | 0.001136 | 3.3 | 26 | 0.000732 | 5.5 | 14.80 | 0.81 | 0.000842 | 3.5 | **17.01** | **0.60** |
| | **SAND1@12** | 729 | 3802 | 5 | 203 | 16 | 0.150 | 2.8 | 0.001027 | 2.6 | 16 | 0.000831 | 3.8 | 16.79 | 0.63 | 0.000858 | 2.7 | **17.34** | **0.47** |
| | **SAND1@13** | 628 | 2842 | 5 | 170 | 14 | 0.136 | 2.3 | 0.001072 | 3.0 | 19 | 0.000839 | 3.9 | 16.95 | 0.67 | 0.000873 | 3.0 | **17.63** | **0.54** |
| | **SAND1@14** | 966 | 3518 | 4 | 124 | 16 | 0.194 | 3.1 | 0.001118 | 2.6 | 23 | 0.000771 | 5.2 | 15.57 | 0.81 | 0.000862 | 2.8 | **17.41** | **0.48** |
| | **SAND1@18** | 278 | 1247 | 4 | 67 | 14 | 0.309 | 2.9 | 0.001372 | 5.0 | 43 | 0.000578 | 8.4 | 11.67 | 0.97 | 0.000783 | 5.5 | **15.81** | **0.88** |
| | **SAND1@19** | 277 | 811 | 3 | 70 | 16 | 0.321 | 3.0 | 0.001620 | 5.1 | 55 | 0.000731 | 9.2 | 14.77 | 1.36 | 0.000725 | 6.5 | **14.66** | **0.96** |
| | *SAND1@11* | *659* | *2325* | *4* | *92* | *16* | *0.217* | *3.1* | *0.001572* | *8.4* | *35* | *0.000909* | *8.5* | *18.36* | *1.56* | *0.001021* | *8.6* | *20.62* | *1.78* |
| | *SAND1@15* | *189* | *964* | *5* | *49* | *11* | *0.307* | *2.4* | *0.001476* | *6.1* | *49* | *0.000322* | *8.3* | *6.51* | *0.54* | *0.000751* | *6.7* | *15.17* | *1.01* |
| | *SAND1@16* | *228* | *601* | *3* | *59* | *15* | *0.307* | *2.8* | *0.001752* | *6.7* | *62* | *0.000612* | *10.1* | *12.36* | *1.25* | *0.000664* | *8.7* | *13.41* | *1.17* |
| | *SAND1@17* | *202* | *401* | *2* | *46* | *15* | *0.312* | *2.9* | *0.002061* | *8.4* | *73* | *0.000324* | *12.4* | *6.56* | *0.81* | *0.000559* | *13.0* | *11.30* | *1.47* |
| | *SAND1@20* | *233* | *376* | *2* | *54* | *17* | *0.305* | *3.0* | *0.002238* | *7.9* | *74* | *0.000741* | *12.4* | *14.98* | *1.86* | *0.000578* | *13.8* | *11.68* | *1.61* |
| | *SAND1@29* | *493* | *448* | *1* | *112* | *67* | *0.056* | *11.1* | *0.001114* | *2.4* | *9* | *0.000729* | *23.3* | *14.72* | *3.43* | *0.001014* | *6.8* | *20.49* | *1.39* |
| | *SAND1@30* | *640* | *398* | *1* | *186* | *95* | *0.060* | *11.3* | *0.001180* | *3.8* | *14* | *0.000935* | *20.0* | *18.88* | *3.78* | *0.001019* | *9.3* | *20.59* | *1.91* |
| | *SAND1@31* | *636* | *404* | *1* | *95* | *67* | *0.061* | *10.6* | *0.001085* | *3.4* | *21* | *0.001505* | *38.9* | *30.39* | *11.80* | *0.000856* | *10.2* | *17.30* | *1.76* |
| | *SAND1@23* | *299* | *5895* | *20* | */* | *0* | *0.112* | *10.5* | *0.000878* | *1.2* | *3* | *0.000878* | */* | *17.73* | */* | *0.000855* | *1.2* | *17.27* | *0.21* |
| | *SAND1@27* | *284* | *679* | *2* | */* | *0* | *0.069* | *7.7* | *0.001130* | *1.0* | *9* | *0.001130* | */* | *22.84* | */* | *0.001027* | *2.8* | *20.74* | *0.58* |
| | *SAND1@32* | *573* | *501* | *1* | *33* | *96* | *0.017* | *17.8* | *0.000118* | *15.8* | *89* | */* | */* | *-0.43* | *-0.49* | *0.000013* | *251* | *0.26* | *0.66* |
| | *SAND1@33* | *919* | *304* | *0* | */* | *0* | *0.035* | *18.9* | *0.000222* | *13.7* | *184* | *0.000222* | */* | *4.49* | */* | */* | *-76* | *-3.76* | *2.86* |
| | *SAND1@35* | *1160* | *691* | *1* | */* | *0* | *0.057* | *12.0* | *0.001323* | *2.2* | *8* | *0.001323* | */* | *26.73* | */* | *0.001215* | *5.7* | *24.54* | *1.39* |
| | *SAND1@36* | *935* | *512* | *1* | */* | *0* | *0.067* | *11.3* | *0.001293* | *1.9* | *20* | *0.001293* | */* | *26.11* | */* | *0.001037* | *8.8* | *20.95* | *1.84* |
| | **REIS1@27** | 166 | 541 | 3 | 188 | 25 | 0.088 | 4.1 | 0.000926 | 2.32 | 13 | 0.000762 | 5.7 | 15.39 | 0.88 | 0.000806 | 2.7 | **16.28** | **0.44** |

| | | | | | | | | | | | | | | | | | | | |
|---|---|---|---|---|---|---|---|---|---|---|---|---|---|---|---|---|---|---|---|
| A | **REIS1@28** | 288 | 1516 | 5 | 601 | 30 | 0.077 | 3.8 | 0.000836 | 2.25 | 6 | 0.000783 | 2.8 | 15.81 | 0.45 | 0.000788 | 2.3 | **15.93** | **0.37** |
| | **REIS1@29** | 185 | 697 | 4 | 530 | 42 | 0.093 | 4.6 | 0.000853 | 2.24 | 11 | 0.000791 | 3.7 | 15.99 | 0.59 | 0.000760 | 2.5 | **15.36** | **0.39** |
| | **REIS1@31** | 141 | 763 | 5 | 226 | 42 | 0.121 | 3.8 | 0.000977 | 2.60 | 20 | 0.000810 | 7.6 | 16.37 | 1.24 | 0.000779 | 4.2 | **15.74** | **0.66** |
| | **REIS1@32** | 306 | 2964 | 10 | 260 | 26 | 0.059 | 3.6 | 0.000893 | 2.48 | 15 | 0.000760 | 4.4 | 15.36 | 0.68 | 0.000759 | 2.8 | **15.33** | **0.42** |
| | **REIS1@33** | 762 | 737 | 1 | 874 | 24 | 0.050 | 2.7 | 0.000877 | 2.56 | 2 | 0.000839 | 2.7 | 16.94 | 0.45 | 0.000863 | 2.6 | **17.44** | **0.45** |
| | **REIS1@35** | 195 | 618 | 3 | 290 | 32 | 0.083 | 4.3 | 0.000966 | 2.95 | 10 | 0.000865 | 5.2 | 17.48 | 0.90 | 0.000865 | 3.2 | **17.48** | **0.56** |
| | **REIS1@36** | 153 | 668 | 4 | 222 | 27 | 0.125 | 4.0 | 0.000963 | 2.87 | 18 | 0.000796 | 5.3 | 16.08 | 0.86 | 0.000794 | 3.2 | **16.05** | **0.52** |
| | **REIS1@23** | 236 | 324 | 1 | 326 | 47 | 0.053 | 5.0 | 0.000821 | 2.56 | 8 | 0.000771 | 6.7 | 15.57 | 1.04 | 0.000756 | 3.8 | **15.27** | **0.58** |
| | **REIS1@17** | 437 | 738 | 2 | 515 | 33 | 0.055 | 2.9 | 0.000903 | 2.26 | 7 | 0.000852 | 3.4 | 17.21 | 0.59 | 0.000842 | 2.7 | **17.01** | **0.45** |
| | **REIS1@19** | 384 | 454 | 1 | 318 | 33 | 0.062 | 3.2 | 0.000932 | 2.69 | 15 | 0.000819 | 4.7 | 16.54 | 0.78 | 0.000793 | 3.8 | **16.03** | **0.61** |
| | **REIS1@20** | 353 | 562 | 2 | 1500 | 67 | 0.062 | 3.3 | 0.000894 | 2.24 | 11 | 0.000870 | 2.8 | 17.59 | 0.49 | 0.000798 | 2.9 | **16.13** | **0.46** |
| | **REIS1@21** | 288 | 433 | 2 | 336 | 36 | 0.060 | 3.4 | 0.000981 | 2.54 | 10 | 0.000868 | 4.7 | 17.54 | 0.82 | 0.000881 | 3.6 | **17.81** | **0.63** |
| B | **REIS1@18** | 328 | 486 | 1 | 467 | 42 | 0.061 | 3.3 | 0.000802 | 2.96 | 18 | 0.000736 | 4.4 | 14.87 | 0.66 | 0.000657 | 4.1 | **13.27** | **0.55** |
| | **REIS1@24** | 218 | 398 | 2 | 689 | 67 | 0.064 | 4.9 | 0.000791 | 3.13 | 14 | 0.000747 | 4.8 | 15.08 | 0.72 | 0.000679 | 4.1 | **13.71** | **0.56** |
| | **REIS1@25** | 189 | 225 | 1 | 321 | 55 | 0.061 | 5.0 | 0.000851 | 3.74 | 17 | 0.000748 | 7.4 | 15.12 | 1.11 | 0.000705 | 6.1 | **14.25** | **0.87** |
| | **REIS1@30** | 297 | 441 | 1 | 778 | 47 | 0.083 | 5.2 | 0.000749 | 3.54 | 10 | 0.000712 | 4.1 | 14.38 | 0.59 | 0.000676 | 4.6 | **13.66** | **0.63** |
| | **REIS1@34** | 560 | 505 | 1 | 437 | 47 | 0.044 | 3.3 | 0.000773 | 3.93 | 14 | 0.000738 | 6.0 | 14.90 | 0.89 | 0.000666 | 6.2 | **13.45** | **0.84** |
| | *REIS1@22* | *881* | *386* | *0* | *229* | *32* | *0.051* | *2.3* | *0.000979* | *2.77* | *25* | *0.000814* | *5.8* | *16.45* | *0.95* | *0.000736* | *34.6* | *14.88* | *5.15* |
| | *REIS1@26* | *201* | *197* | *1* | *199* | *47* | *0.075* | *4.5* | *0.000867* | *3.41* | *41* | *0.000781* | *10.7* | *15.79* | *1.69* | *0.000508* | *11.6* | *10.26* | *1.19* |

Table 4. Summary of weighted mean ages of monazite growth domains and spot age ranges of each grain from the TW.

| Sample domain | ID N° | Figure | Table | Zoning of the grains | Weighted mean domain $^{208}$Pb/$^{232}$Th ages (Ma, ± 1σ) | MSWD | Number of analyses | Spot $^{208}$Pb/$^{232}$Th age range of entire grain (Ma, ± 1σ) | Reference |
|---|---|---|---|---|---|---|---|---|---|
| *Western Tauern Window* | | | | | | | | | |
| INNB1 | 1 | 3a | 3 | Regular | – | – | – | 11.5 ± 0.2 - 10.4 ± 0.2 | this study |
| ZEI1 - A | 2 | 3b | 3 | Regular | 10.0 ± 0.2 | 1.8 | 20 | 10.8 ± 0.3 - 7.2 ± 0.2 | this study |
| SCHR1 - A | | | | | 20.9 ± 0.6 | 1.7 | 6 | | |
| SCHR1 - B | 3 | 3c | 4 | Regular | 20.3 ± 0.2 | 0.98 | 16 | 21.9 ± 0.5 - 19.3 ± 0.5 | this study |
| SCHR1 - C | | | | | 19.7 ± 0.4 | 1.00 | 6 | | |
| MAYR4 | 4 | 3d | 3 | Regular | – | – | – | 11.8 ± 0.2 - 8.9 ± 0.2 | this study |
| PFIT1 - A | 5 | 3e | 4 | Patchy core | 17.3 ± 0.3 | 1.2 | 9 | 17.8 ± 0.4 - 12.9 ± 0.3 | this study |
| PFIT1 - B | | | | | 13.2 ± 0.3 | 0.38 | 5 | | |
| BURG2 | 6 | 3f | 3 | Regular | – | – | – | 17.1 ± 0.4 – 12.1 ± 0.3 | this study |
| PLAN1 - A | 7 | 3g | 3 | Patchy core | 11.9 ± 0.2 | 2.2 | 37 | 12.6 ± 0.3 - 7.8 ± 0.2 | this study |
| *Central Tauern Window* | | | | | | | | | |
| SCHEI1 - A | | | | | 18.3 ± 1.1 | 2.0 | 4 | | |
| SCHEI1 - B | 8 | 4a | 3 | Regular | 17.4 ± 0.4 | 1.5 | 5 | 18.9 ± 0.5 - 15.9 ± 0.4 | this study |
| SCHEI1 - C | | | | | 16.6 ± 0.2 | 1.9 | 23 | | |
| HOPF2 - B | 9 | 4b | 4 | Regular | 12.2 ± 0.4 | 2.6 | 8 | 13.7 ± 0.4 - 11.0 ± 0.3 | this study |
| HOPF2 - C | | | | | 12.2 ± 0.5 | 2.9 | 6 | | |
| GART1 - A | 10 | 4c | 3 | Regular | 16.3 ± 0.2 | 0.69 | 10 | 16.9 ± 0.3 - 14.5 ± 0.4 | this study |
| NOWA3 - B | 11 | 4d | 3 | Regular | 15.8 ± 0.5 | 0.27 | 5 | 17.0 ± 0.2 - 13.8 ± 0.8 | this study |
| NOWA3 - C | | | | | 14.9 ± 1.1 | 2.4 | 5 | | |
| GART3 - B | 12 | 4e | 3 | Regular | 15.0 ± 0.5 | 2.3 | 11 | 16.1 ± 0.4 - 12.0 ± 0.4 | this study |
| STEI2 | 13 | 4f | 3 | Regular / Patchy tail | 17.2 ± 0.2 | 0.24 | 20 | 17.5 ± 0.4 - 16.8 ± 0.4 | this study |
| KNOR1 - A | | | | | 10.8 ± 0.3 | 1.02 | 5 | | |
| KNOR1 - B | 14 | 4g | 4 | Regular | 10.6 ± 0.3 | 1.6 | 8 | 11.6 ± 0.4 - 10.8 ± 0.3 | this study |
| KNOR1 - C | | | | | 10.4 ± 0.2 | 1.4 | 8 | | |

*Eastern Tauern Window*

| | | | | | | | | | |
|---|---|---|---|---|---|---|---|---|---|
| **KAIS6 - A** | | | | | 21.2 ± 0.5 | 0.64 | 6 | | |
| **KAIS6 - B** | **15** | 5a | 4 | Patchy border | 20.9 ± 0.2 | 0.53 | 24 | 22.1 ± 0.6 - 17.6 ± 0.6 | this study |
| **KAIS6 - C** | | | | | 20.6 ± 0.5 | 0.34 | 7 | | |
| **KAIS6 - D** | | | | | 18.8 ± 0.5 | 1.5 | 10 | | |
| **SALZ18-A** | **16a** | 5b | 4 | Regular | 18.3 ± 0.4 | 2.6 | 14 | 19.5 ± 0.5 - 15.8 ± 0.4 | this study |
| | | | | | *18.1 ± 0.4* | *0.51* | *4* | | |
| *T3* | *16b* | | - | *Regular* | *17.2 ± 0.5* | *3.4* | *10* | *18.5 ± 0.4 - 14.8 ± 0.4* | *Gnos et al., 2015* |
| | | | | | *16.0 ± 0.3* | *0.51* | *8* | | |
| | | | | | *15.5 ± 0.2* | *0.74* | *24* | | |
| **LOHN4 - A** | **17** | 5c | 4 | Patchy core | 21.1 ± 0.2 | 1.4 | 50 | 22.9 ± 0.6 - 17.3 ± 0.6 | this study |
| **LOHN4 - B** | | | | | 18.4 ± 0.6 | 1.3 | 7 | | |
| **ORT1** | **18** | 5d | 3 | Regular | 18.4 ± 0.3 | 1.07 | 13 | 19.0 ± 0.6 - 17.0 ± 0.7 | this study |
| **EUKL2** | **19a** | 5e | 4 | Regular | 21.7 ± 0.4 | 0.56 | 7 | 22.3 ± 0.5 - 21.2 ± 0.5 | this study |
| *T2* | *19b* | | - | *Patchy* | *15.1 ± 0.5* | *0.26* | *4* | *15.4 ± 0.4 - 15.0 ± 0.7* | *Gnos et al., 2015* |
| **HOAR1-A** | **20** | 5f | 3 | Patchy | 20.4 ± 0.2 | 0.80 | 6 | 21.2 ± 0.7 - 19.0 ± 0.9 | this study |
| **HOAR1-B** | | | | | 19.9 ± 0.3 | 0.95 | 25 | | |
| | | | | | *19.0 ± 0.5* | *0.51* | *5* | | |
| *T1* | *21* | | - | *Regular* | *17.6 ± 0.6* | *1.6* | *8* | *19.2 ± 0.5 - 14.3 ± 0.5* | *Gnos et al., 2015* |
| | | | | | *16.3 ± 0.6* | *3.0* | *12* | | |
| | | | | | *15.0 ± 0.5* | *1.7* | *8* | | |
| *T4* | *22* | | - | *Patchy* | *15.6 ± 0.7* | *9.1* | *21* | *18.3 ± 1.1 - 13.1 ± 0.8* | *Gnos et al., 2015* |
| **MOKR1 - B** | **23** | 5g | 3 | Regular | 18.8 ± 0.5 | 2.9 | 12 | 22.6 ± 0.4 - 14.4 ± 0.2 | this study |
| **SAND1 - B** | **24** | 5h | 3 | Regular | 17.0 ± 0.8 | 1.8 | 7 | 22.0 ± 0.3 - 14.7 ± 1.0 | this study |
| **REIS1 - A** | **25** | 5i | 3 | Regular | 16.2 ± 0.5 | 2.9 | 13 | 17.8 ± 0.6 - 13.5 ± 0.8 | this study |
| **REIS1 - B** | | | | | 13.6 ± 0.6 | 0.25 | 5 | | |