# Peer review of "Cenozoic deformation in the Tauern Window (Eastern Alps) constrained by in-situ Th-Pb dating of fissure monazite"

_Solid Earth, 2019_

## Referee Comment (RC1) · Urs Klötzli (Referee) · 13 Nov 2019

General comments: The manuscript of Ricchi et al. provides a substantial number of new SIMS monazite spot ages from fault zone-relate hydrothermal fissure monazites from the Tauern Window in the Eastern Alps substantially complementing available thermochronological data sets from the same region.

Three major periods of monazite growth are recorded between 22 – 19 Ma, 19 – 15 Ma and 13 – 8 Ma. The ages are interpreted to be related to N-S shortening in association with E-W extension, the beginning of strike-slip movements, and reactivation of strike-slip faulting in the Tauern Window. These findings very nicely support the interpretation

of former 40Ar-39Ar mica and zircon, apatite fission track ages.

I find a slight weakness in the formulation of the interpretation of the monazite SIMS dates: The monazite dates do not date tectonic activity in the sense of tectonic movements, as stated in the manuscript. They merely date fluid activity which may be indirectly related to tectonic movements. This is a slight but important difference and should be correctly stressed in the text.

Specific comments: There are a number of issues concerning BSE imaging, SIMS data reduction and interpretation which have to be addressed. Most importantly the authors use two different common Pb correction schemes which are not necessarily comparable. Therefore I have some doubts whether or not the two data sets are directly comparable, inasmuch as very small age differences (1-2 Ma) are interpreted to be significant. I would very much like to see both raw data sets reduced with the same common Pb correction scheme.

The authors interpret the BSE signal intensity, aka zonation, as representing growth domains/zones. This is not quite correct. What one sees in the BSE images is the chemical zonation and/or chemically $\pm$ homogeneous domains. That such domains are characteristic for growth domains is an (over-)interpretation. I therefore suggest to be more objective in interpreting the monazite BSE images.

There are a number of rather vaguely formulated statements which I find should be stated more precise and stringent. For instance line 239: '... bring the MSWD within acceptable values...'. What such acceptable values should be remains for the reader to find out by himself. I suggest that all such formulations are avoided.

Technical comments: I find the English to be very fine and have only found a few typos. So from this point of view the manuscript is easily acceptable.

I suggest to move some statements concerning tectonism from the 'Results' section to the 'Geologic settings' section.

Some of the figures show too small labelling. Probably this ought to be changed.

The SIMS data tables are not complete. They do not provide all necessary data for the reader. This has to be corrected.

I have given a number of specific comments directly in the manuscript.

Please also note the supplement to this comment:
https://www.solid-earth-discuss.net/se-2019-162/se-2019-162-RC1-supplement.pdf

**Supplement:**

[revised manuscript text omitted]

Groups: A, B, A (left margin labels)

[revised manuscript text omitted]

Row group labels: B (PFIT1 section); A, B, C (HOPF2 section)

| | Sample | | | | | | | | | | |
|---|---|---|---|---|---|---|---|---|---|---|---|
| | HOPF2@06 | 58 | 28865 | 500 | 480 | 4.524 | 6 | 0.000602 | 2.08 | 12.16 | 0.25 |
| A | KNOR1_@19 | 149 | 12269 | 82 | 465 | 6.834 | 6 | 0.000526 | 2.30 | 10.64 | 0.24 |
| | KNOR1_@20 | 181 | 9307 | 52 | 277 | 7.353 | 13 | 0.000512 | 3.04 | 10.34 | 0.31 |
| | KNOR1_@21 | 239 | 9337 | 39 | 647 | 14.146 | 5 | 0.000574 | 3.59 | 11.60 | 0.42 |
| | KNOR1_@22 | 172 | 8289 | 48 | 540 | 9.262 | 6 | 0.000534 | 2.67 | 10.78 | 0.29 |
| | KNOR1_@24 | 194 | 7199 | 37 | 615 | 12.664 | 6 | 0.000515 | 3.08 | 10.40 | 0.32 |
| B | KNOR1_@01 | 32 | 7767 | 246 | 194 | 5.350 | 19 | 0.000499 | 2.07 | 10.08 | 0.21 |
| | KNOR1_@02 | 35 | 6107 | 175 | 202 | 6.264 | 18 | 0.000524 | 2.46 | 10.59 | 0.26 |
| | KNOR1_@03 | 36 | 9885 | 273 | 114 | 4.841 | 33 | 0.000517 | 2.44 | 10.44 | 0.25 |
| | KNOR1_@04 | 30 | 6902 | 234 | 93 | 3.382 | 41 | 0.000545 | 1.92 | 11.00 | 0.21 |
| | KNOR1_@06 | 33 | 9203 | 277 | 171 | 4.416 | 21 | 0.000525 | 2.02 | 10.61 | 0.21 |
| | KNOR1_@07 | 33 | 4983 | 152 | 112 | 4.810 | 33 | 0.000518 | 2.24 | 10.46 | 0.23 |
| | KNOR1_@08 | 36 | 3783 | 105 | 136 | 5.985 | 26 | 0.000516 | 2.37 | 10.42 | 0.25 |
| | KNOR1_@09 | 39 | 5846 | 149 | 302 | 7.354 | 11 | 0.000515 | 2.28 | 10.41 | 0.24 |
| C | KNOR1_@05 | 27 | 9535 | 356 | 104 | 3.093 | 35 | 0.000565 | 1.93 | 11.41 | 0.22 |
| | KNOR1_@10 | 48 | 5633 | 118 | 453 | 11.056 | 7 | 0.000501 | 2.40 | 10.13 | 0.24 |
| | KNOR1_@11 | 48 | 4611 | 96 | 445 | 11.786 | 6 | 0.000525 | 2.59 | 10.62 | 0.28 |
| | KNOR1_@12 | 57 | 4583 | 81 | 459 | 10.996 | 7 | 0.000502 | 2.24 | 10.15 | 0.23 |
| | KNOR1_@13 | 54 | 4104 | 77 | 489 | 11.330 | 7 | 0.000504 | 2.33 | 10.18 | 0.24 |
| | KNOR1_@14 | 56 | 6093 | 108 | 724 | 13.355 | 4 | 0.000493 | 2.22 | 9.96 | 0.22 |
| | KNOR1_@15 | 76 | 6847 | 91 | 705 | 13.068 | 3 | 0.000511 | 2.26 | 10.33 | 0.23 |
| | KNOR1_@16 | 65 | 5872 | 91 | 522 | 14.528 | 5 | 0.000503 | 2.36 | 10.16 | 0.24 |
| | KNOR1_@18 | 77 | 8867 | 114 | 630 | 10.939 | 3 | 0.000541 | 2.55 | 10.93 | 0.28 |
| | KNOR1_@17 | 73 | 6281 | 86 | 577 | 10.481 | 4 | 0.000564 | 2.28 | 11.40 | 0.26 |

*Eastern Tauern Window*

[revised manuscript text omitted]

---

## Referee Comment (RC2) · Jan Pleuger (Referee) · 15 Dec 2019

General comments

The manuscript "Cenozoic deformation in the Tauern Window (Eastern Alps, Austria) constrained by in-situ Th-Pb dating of fissure monazite" by E. Ricchi et al. contains a rather large number of new data that add to extensive thermochronological data sets available for the Tauern Window. The new data are of broad interest both from a regional geological and from a methodological viewpoint because by comparing their new data set with apatite and zircon fission track and 40Ar/39Ar data from the literature, the authors illustrate convincingly the potential and limitations of assessing the age of

tectonism by monazite dating. The manuscript meets the quality criteria for publication in Solid Earth, however, I have a number of suggestions for improvement below that altogether require moderate revisions of the manuscript. I should also say that I read the comments of the other reviewer and agree with his specific comments, so in the following I will not dwell on the analytical procedures (e.g. BSE imaging and SIMS reduction) but concentrate on the regional geological aspects.

Chapters 4.1 and 5.1: I have some difficulties to understand if or how the authors connect different structures (i.e. C1, C2, and C3 in Fig. 2) to the age data for which also a red-green-blue colour scheme is used (Fig. 7). Is the colour coding for C1, C2, and C3 in Fig. 2 the same as in Fig. 7? In other words, are C1 fissures supposed to have formed at 22-19 Ma, C2 at 19-15 Ma and C3 at 13-8 Ma? If so, these links should be explicitly mentioned in chapter 5.1. Please mention also in chapter 4.1 to which generation of fissures PFIT1 belongs.

Figures: In basically all of the figures the labelling is to small, especially if the figures are given as poor-quality raster (like in the file that this review is based on) and not vector graphics.

Specific comments

Lines 105-108, MöF and MVF: I do not understand in how far the distinction between the Möll valley fault (MVF) and the Mölltal fault (MöF) is justified. The MVF is not mapped in the references cited in the caption of Fig. 1, except for Schneider et al. (2013) where, however, the fault itself is not marked but only a half-arrow indicating dextral shear sense is given. If the authors insist on the existence of an MVF, I see no reason why it should not be the same structure as the MöF.

Lines 105-108, Katschberg fault: The northern, E-W trending stretch of the KSZS is dextral, not sinistral. Somewhat northeast of and parallel to the northwestern stretch of the MöF in Fig. 1, there is a sinistral shear zone that is also part of the Katschberg fault.

Lines 116-118: Folding alone cannot have caused significant exhumation; the idea of Schmid et al. (2013) is that exhumation was achieved by extrusion of the HP unit that went together with folding.

Lines 118-120: "... this exhumation episode ..." logically refers to D3. However, breakoff of the European slab was linked to D4 by Schmid et al. (2013).

Lines 127, 128: What is "isochron corrected Rb-Sr dating"?

Lines 186-193 (chapter 4 Results): No results but rather general informations about fissure and cleft formation are given here. With respect to line 191, more complex (sigmoidal) shapes may also result from non-coaxial progressive deformation. In that case, also the orientation with respect to the foliation and lineation changes progressively.

Lines 202-203: The sigmoidal shape of C2 fissures should indicate a shear sense. It would be good to give information about the shear sense of strike-slip faulting here.

Lines 210-211: It is safer to write here "This would indicate a similar direction of extension for the development ..."

Lines 212-218: Since C1 and C2 are perpendicular to stretching lineations L1 and L2 that are parallel to each other (Fig. 2c), I do not understand how it is possible to distinguish the two fissure generations based on their orientation.

Lines 220, 222: According to Fig. 2b&c, C3 is not perpendicular to C1 and C2 and L3 is not down-dip.

Lines 406-407: I find it a bit misleading to say the faults were active at 21, 17 and 12 Ma because the time ranges for the three phases of fissure formation given just before indicate almost continuous monazite growth between 22 and 8 Ma.

Lines 407-410: The fission track age distribution in Fig. 7c is strictly speaking not U-shaped, but forms a dome or inverse U. More importantly, the relatively young fission track ages are from the margins of the Tauern dome from where the authors have not

dated monazite samples. Therefore, I find also the conclusion that "monazite crystallization ages do not show the U-shaped distribution as cooling ages" misleading. A more valid conclusion would be in my opinion that the latest stages of monazite crystallisation happened at temperatures that are between apatite and fission track "closure" temperatures.

Fig. 1: There is apparently no interference between the ZWD and the IsF. This is probably a result of compiling different maps and looks very peculiar.

Technical corrections

Line 55: "... fissures form."

Line 64: "... fluid that fills ..." or "... fluids that fill ..."

Line 95: "... (1-25) ..."

Lines 104, 648: "... Ahrntal shear zone ..."

Line 120: I suggest to write "This was followed by an inversion of subduction polarity ..." because it is clearer than "... slab dynamics ..."

Line 192: "... studies ..."

Line 197: I suggest to add "5" behind "sample locality" because then the place is much easier to find in Fig. 1

Line 210: "... in figure 2c ..."

Line 215: "... fold axes ..."

Line 221: I suggest to replace "displays" with "is related to"

Lines 222, 223, 225: I suggest to replace "oriented" with "striking"

Line 229: "... it shows ..."

Line 240: "In a few cases the dates for specific monazite domains ..."

Line 247: "... which delimits the TW to the west ..."

Line 273: " ... starts to grow (phase 1, red symbols in Fig. 7) ..."

Line 290: "... the peak activity of which ..."

Line 307: Replace "sinistral" with "dextral"

Line 322: "... was active ..."

Line 344: I suggest to replace "occurred" with "started"

Line 366: I suggest to replace "Whereas" with "By contrast,"

Line 374: "... monazites ..."

Line 388: Delete "decreases"

Lines 395-396: "... during the Miocene."

Line 404: "... associated with ..."

Lines 444, 452, 457, 533, 585, 587, 634: Please add names of journal or book

Lines 492, 495: Please check for special characters

Legend of Fig 1.: "Periadriatic"

Lines 637-639: Below the legend of Fig. 1, but not in the figure caption, Scharf et al. (2013) is cited as a source for the map.

Lines 648-649: "... Defereggen ... Mur-Mürz ..." (these are spelled incorrectly also in Schneider et al. 2013)

Line 694: "... sample locations ..."

Line 689: "... are presented ..."

Line 695: "DD' NE-SW cross section across the BNF."

Line 704: "Zircon and apatite fission track ages ..."

Figure 7: Please integrate the labelling at the bottom of Fig. 7 ("(1) Early record ... oblique-slip") into the figure caption

Text at the bottom of Table 2: "Blanckenburg"

Supplementary information (SI), description of INNB1: "... between 7265-21,420 and ..." – likewise for descriptions of MAYR4, BURG2, SCHEI1, SALZ18, ORT1

SI, description of ZEI1: "ZEI1 ... can be subdivided in three domains (ZEI1-A, -B, and –C) ..."

SI, description of SCHEI1: "... Table 5), thus assuming ..."

SI, description of NOWA3: "... displays three ..."

SI, description of GART3: "...GART3-C composition is strongly clustered ..."

SI, description of STEI2: "... between 2676-12,353 and 131-407 ppm ..."

---

## Author Comment (AC1) · 14 Feb 2020

Answers to the comments in this letter are bracketed by two dashes.

Urs Klötzli urs.kloetzli@univie.ac.at

General comments The manuscript of Ricchi et al. provides a substantial number of new SIMS monazite spot ages from fault zone-relate hydrothermal fissure monazites from the Tauern Window in the Eastern Alps substantially complementing available thermochronological data sets from the same region. Three major periods of monazite

growth are recorded between 22 – 19 Ma, 19 – 15 Ma and 13 – 8 Ma. The ages are interpreted to be related to N-S shortening in association with E-W extension, the beginning of strike-slip movements, and reactivation of strike slip faulting in the Tauern Window. These findings very nicely support the interpretation of former 40Ar-39Ar mica and zircon, apatite fission track ages.

I find a slight weakness in the formulation of the interpretation of the monazite SIMS dates: The monazite dates do not date tectonic activity in the sense of tectonic movements, as stated in the manuscript. They merely date fluid activity which may be indirectly related to tectonic movements. This is a slight but important difference and should be correctly stressed in the text. - -We agree that the formulation of the interpretation of monazite ages needs to be stressed. The following sentence was added to the manuscript: "Fissure monazites date crystallization following chemical disequilibrium within a fissure. This causes a dissolution-precipitation cycle that may include dissolution or partial dissolution of existing fissure monazite. This has the consequence that late dissolution/precipitation steps may be well recorded, whereas earlier growth domains may be completely destroyed. Thus, monazite crystallization due to chemical disequilibrium is interpreted as being related to tectonic activity (e.g. volume change, fissure propagation, exposure of fresh host-rock, delamination of fissure wall, seismic activity, fluid loss or gain)."- -

Specific comments There are a number of issues concerning BSE imaging, SIMS data reduction and interpretation which have to be addressed. Most importantly the authors use two different common Pb correction schemes which are not necessarily comparable. Therefore I have some doubts whether or not the two data sets are directly comparable, inasmuch as very small age differences (1-2 Ma) are interpreted to be significant. I would very much like to see both raw data sets reduced with the same common Pb correction scheme. - -We agree and for sake of consistency we decided to reduce all the data using the CIPS software. Th-U-Pb analyses of all the grains are now presented in the new Table 3 where both 204- and 207-corrected ratios and ages are provided. In every case 204- and 207-corrected ages agree within uncertainty. The weighted mean ages summarized in the new Table 4 and described in the SI file and in the reviewed manuscript were calculated using the 207-corrected ages because these ages are more robust and consistent (better statistics and less scatter in the data).- -

The authors interpret the BSE signal intensity, aka zonation, as representing growth domains/zones. This is not quite correct. What one sees in the BSE images is the chemical zonation and/or chemically $\pm$ homogeneous domains. That such domains are characteristic for growth domains is an (over-)interpretation. I therefore suggest to be more objective in interpreting the monazite BSE images. - -Chemical domains (A, B, etc.; displayed in Figs.3 and 4 and listed in Tables 3-4) were defined combining Th and U concentration with textural information (chemical zoning visible on BSE images). We interpret chemical domains as representing growth domains caused by chemical disequilibrium in the fissure. Most of these domains have regular zoning typical of growth. Quartz domains are visible in cathodoluminescence images and fluid inclusions in the different domains indicate different temperatures of crystallization. For these reasons we believe that the same applies to monazite (in most cases too small for fluid inclusions studies). Our experience shows that in fissure monazite, the chemical domains generally are growth domains of distinct age, in some cases overlapping within uncertainty.- -

There are a number of rather vaguely formulated statements which I find should be stated more precise and stringent. For instance line 239: '... bring the MSWD within acceptable values...'. What such acceptable values should be remains for the reader to find out by himself. I suggest that all such formulations are avoided. - -We agree and this was corrected as follows: "[...] MSWD within acceptable values (MSWD < 3; Spencer et al., 2016)."- -

Technical comments I find the English to be very fine and have only found a few typos. So from this point of view the manuscript is easily acceptable.

I suggest to move some statements concerning tectonism from the 'Results' section to the 'Geologic settings' section. - -We prefer to keep the statements regarding the field observations in the "Results" section because we develop here a description of the fissures we observed of the field.- -

Some of the figures show too small labelling. Probably this ought to be changed. - -Small labels have been marked in bold and figures provided in vector graphics.- -

The SIMS data tables are not complete. They do not provide all necessary data for the reader. This has to be corrected. - -The missing information is now provided.- -

I have given a number of specific comments directly in the manuscript. Please also note the supplement to this comment: https://www.solid-earth-discuss.net/se-2019-162/se-2019-162-RC1-supplement.pdf - -Please see all points addressed in the annotated manuscript.- -

Please also note the supplement to this comment:
https://www.solid-earth-discuss.net/se-2019-162/se-2019-162-AC1-supplement.pdf

**Supplement:**

[revised manuscript text omitted]

*Eastern Tauern Window*

|  |  | | | | | | | | | |
|---|---|---|---|---|---|---|---|---|---|---|
| A | **KAIS6@29** | 12 | 32285 | 2726 | 324 | 2.307 | 11 | 0.001014 | 2.41 | 20.48 | 0.49 |
|  | **KAIS6@37** | 10 | 35152 | 3587 | 323 | 2.469 | 11 | 0.001065 | 2.44 | 21.51 | 0.52 |
|  | **KAIS6@38** | 10 | 35481 | 3480 | 438 | 2.739 | 8 | 0.001048 | 2.54 | 21.17 | 0.54 |
|  | **KAIS6@39** | 10 | 31600 | 3326 | 311 | 2.609 | 12 | 0.001020 | 2.44 | 20.60 | 0.50 |
|  | **KAIS6@41** | 8 | 32642 | 4280 | 353 | 3.271 | 10 | 0.001058 | 2.50 | 21.36 | 0.53 |
|  | **KAIS6@42** | 6 | 29873 | 4677 | 291 | 2.296 | 12 | 0.001053 | 2.38 | 21.27 | 0.51 |
| B | **KAIS6@15** | 5 | 21644 | 4568 | 142 | 1.779 | 26 | 0.001030 | 2.04 | 20.82 | 0.42 |
|  | **KAIS6@16** | 5 | 22052 | 4771 | 161 | 1.922 | 23 | 0.001019 | 2.11 | 20.59 | 0.44 |
|  | **KAIS6@17** | 5 | 25100 | 4752 | 186 | 1.993 | 20 | 0.001042 | 2.23 | 21.06 | 0.47 |
|  | **KAIS6@18** | 5 | 25260 | 4905 | 148 | 1.871 | 25 | 0.001040 | 2.16 | 21.00 | 0.45 |
|  | **KAIS6@19** | 5 | 23495 | 5173 | 147 | 2.169 | 25 | 0.001037 | 2.13 | 20.94 | 0.45 |
|  | **KAIS6@20** | 5 | 24260 | 5377 | 155 | 1.894 | 24 | 0.001027 | 2.20 | 20.75 | 0.46 |
|  | **KAIS6@21** | 5 | 25950 | 5040 | 158 | 1.971 | 24 | 0.001010 | 2.15 | 20.40 | 0.44 |

| | | | | | | | | | | |
|---|---|---|---|---|---|---|---|---|---|---|
| **KAIS6@22** | 4 | 20175 | 4625 | 136 | 1.915 | 27 | 0.001037 | 2.08 | 20.95 | 0.44 |
| **KAIS6@23** | 3 | 18427 | 5349 | 153 | 2.062 | 24 | 0.001026 | 2.20 | 20.73 | 0.46 |
| **KAIS6@24** | 4 | 17328 | 4822 | 178 | 2.255 | 20 | 0.001045 | 2.20 | 21.11 | 0.46 |
| **KAIS6@25** | 5 | 17901 | 3588 | 154 | 2.266 | 24 | 0.001042 | 2.19 | 21.04 | 0.46 |
| **KAIS6@26** | 4 | 19477 | 5281 | 165 | 2.214 | 22 | 0.001014 | 2.20 | 20.49 | 0.45 |
| **KAIS6@27** | 4 | 19595 | 5290 | 165 | 2.205 | 22 | 0.001029 | 2.18 | 20.79 | 0.45 |
| **KAIS6@28** | 3 | 17683 | 5368 | 167 | 2.225 | 22 | 0.001052 | 2.21 | 21.25 | 0.47 |
| **KAIS6@31** | 3 | 19464 | 6314 | 208 | 2.462 | 18 | 0.001049 | 2.25 | 21.19 | 0.48 |
| **KAIS6@32** | 3 | 17623 | 5729 | 145 | 2.047 | 26 | 0.001007 | 2.12 | 20.34 | 0.43 |
| **KAIS6@33** | 4 | 23433 | 5395 | 190 | 2.027 | 19 | 0.001016 | 2.29 | 20.53 | 0.47 |
| **KAIS6@34** | 3 | 22517 | 6923 | 175 | 1.970 | 21 | 0.001015 | 2.19 | 20.51 | 0.45 |
| **KAIS6@35** | 4 | 23426 | 6129 | 178 | 1.965 | 21 | 0.001028 | 2.19 | 20.77 | 0.46 |
| **KAIS6@36** | 4 | 24762 | 6924 | 170 | 2.304 | 22 | 0.001032 | 2.23 | 20.86 | 0.46 |
| **KAIS6@43** | 4 | 19511 | 4948 | 229 | 2.717 | 15 | 0.001023 | 2.38 | 20.68 | 0.49 |
| **KAIS6@44** | 5 | 20543 | 3995 | 247 | 3.160 | 15 | 0.001003 | 2.38 | 20.27 | 0.48 |
| **KAIS6@45** | 4 | 17366 | 4418 | 219 | 2.782 | 16 | 0.001061 | 2.29 | 21.43 | 0.49 |
| **KAIS6@46** | 4 | 14595 | 3614 | 231 | 3.427 | 16 | 0.001080 | 2.34 | 21.81 | 0.51 |
| **KAIS6@01** | 2 | 13078 | 6002 | 83 | 2.339 | 46 | 0.001009 | 1.82 | 20.38 | 0.37 |
| **KAIS6@02** | 3 | 14140 | 4835 | 97 | 1.984 | 39 | 0.001031 | 1.87 | 20.82 | 0.39 |
| **KAIS6@05** | 3 | 10157 | 2957 | 98 | 2.228 | 39 | 0.001017 | 1.86 | 20.55 | 0.38 |
| **KAIS6@06** | 4 | 15227 | 3566 | 104 | 2.225 | 36 | 0.000994 | 1.99 | 20.08 | 0.40 |
| **KAIS6@09** | 3 | 10348 | 3946 | 74 | 1.649 | 52 | 0.001026 | 1.56 | 20.74 | 0.32 |
| **KAIS6@10** | 3 | 12624 | 4224 | 90 | 1.888 | 42 | 0.001036 | 1.78 | 20.92 | 0.37 |
| **KAIS6@12** | 3 | 10457 | 3171 | 91 | 2.238 | 42 | 0.001019 | 1.83 | 20.58 | 0.38 |
| **KAIS6@03** | 4 | 15947 | 3922 | 100 | 2.148 | 38 | 0.000941 | 1.98 | 19.02 | 0.38 |
| **KAIS6@04** | 4 | 16156 | 3905 | 91 | 2.181 | 42 | 0.000889 | 2.10 | 17.96 | 0.38 |
| **KAIS6@07** | 6 | 11981 | 2002 | 89 | 3.553 | 43 | 0.000886 | 2.26 | 17.91 | 0.40 |
| **KAIS6@08** | 3 | 10466 | 3145 | 86 | 1.846 | 45 | 0.000949 | 1.71 | 19.16 | 0.33 |
| **KAIS6@11** | 4 | 13094 | 3210 | 89 | 2.393 | 44 | 0.000842 | 1.95 | 17.01 | 0.33 |
| **KAIS6@13** | 4 | 13271 | 3273 | 88 | 2.913 | 44 | 0.000913 | 2.05 | 18.44 | 0.38 |
| **KAIS6@14** | 9 | 13459 | 1568 | 105 | 3.955 | 37 | 0.000834 | 2.69 | 16.85 | 0.45 |
| **KAIS6@30** | 14 | 25376 | 1840 | 229 | 2.365 | 16 | 0.000954 | 2.30 | 19.26 | 0.44 |
| **KAIS6@40** | 10 | 32326 | 3141 | 293 | 2.801 | 12 | 0.000958 | 2.59 | 19.36 | 0.50 |
| **KAIS6@47** | 9 | 31411 | 3692 | 312 | 3.664 | 11 | 0.000949 | 2.42 | 19.16 | 0.46 |
| *KAIS6@48* | *13* | *18728* | *1403* | *217* | *3.029* | *17* | *0.001013* | *2.29* | *20.47* | *0.47* |
| *KAIS6@49* | *13* | *18241* | *1449* | *248* | *2.679* | *14* | *0.001070* | *2.34* | *21.62* | *0.51* |
| | | | | | | | | | | |
| **SALZ18@01** | 11 | 443 | 40 | 426 | 9.205 | 7 | 0.000928 | 2.29 | 18.74 | 0.43 |
| **SALZ18@02** | 19 | 11907 | 639 | 555 | 11.284 | 5 | 0.000919 | 2.20 | 18.56 | 0.41 |
| **SALZ18@03** | 23 | 12452 | 552 | 1310 | 13.609 | 1 | 0.000853 | 2.29 | 17.24 | 0.39 |
| **SALZ18@04** | 20 | 8334 | 407 | 796 | 10.141 | 4 | 0.000908 | 2.48 | 18.35 | 0.46 |

The left margin contains group labels: **C** (aligned with the KAIS6@01–KAIS6@12 block) and **D** (aligned with the KAIS6@03–KAIS6@49 block).

[revised manuscript text omitted]

---

## Author Comment (AC2) · 14 Feb 2020

Answers to the comments in this letter are bracketed by two dashes.

Jan Pleuger (Referee)

jan.pleuger@fu-berlin.de

General comments The manuscript "Cenozoic deformation in the Tauern Window (Eastern Alps, Austria) constrained by in-situ Th-Pb dating of fissure monazite" by E. Ricchi et al. contains a rather large number of new data that add to extensive thermochronological data sets available for the Tauern Window. The new data are of broad

interest both from a regional geological and from a methodological viewpoint because by comparing their new data set with apatite and zircon fission track and 40Ar/39Ar data from the literature, the authors illustrate convincingly the potential and limitations of assessing the age of tectonism by monazite dating. The manuscript meets the quality criteria for publication in Solid Earth, however, I have a number of suggestions for improvement below that altogether require moderate revisions of the manuscript. I should also say that I read the comments of the other reviewer and agree with his specific comments, so in the following I will not dwell on the analytical procedures (e.g. BSE imaging and SIMS reduction) but concentrate on the regional geological aspects.

Chapters 4.1 and 5.1: I have some difficulties to understand if or how the authors connect different structures (i.e. C1, C2, and C3 in Fig. 2) to the age data for which also a red-green-blue colour scheme is used (Fig. 7). Is the colour coding for C1, C2, and C3 in Fig. 2 the same as in Fig. 7? In other words, are C1 fissures supposed to have formed at 22-19 Ma, C2 at 19-15 Ma and C3 at 13-8 Ma? If so, these links should be explicitly mentioned in chapter 5.1. Please mention also in chapter 4.1 to which generation of fissures PFIT1 belongs. - -The colour coding for C1, C2 and C3 in Fig. 2 and Fig. 7 is the same and this is related to the temporal intervals defined: 22-19, 19-15 and 13-8 Ma. However, these time intervals do not record the time of fissure formation, but record the duration of monazite crystallization in the fissures. In the Grimsel and Gotthard regions (e.g. Mullis, 1996) quartz fluid inclusion data have been used for estimating fissure formation in comparison with cooling path defined by thermochronometry. In these areas fissure formation predates monazite crystallization by a few million years (e.g. Ricchi et al., 2019).- -

Figures: In basically all of the figures the labelling is too small, especially if the figures are given as poor-quality raster (like in the file that this review is based on) and not vector graphics. - -Small labels are now written in bold and figures provided as vector graphics.- -

Specific comments Lines 105-108, MöF and MVF: I do not understand in how far the
distinction between the Möll valley fault (MVF) and the Mölltal fault (MöF) is justified. The MVF is not mapped in the references cited in the caption of Fig. 1, except for Schneider et al. (2013) where, however, the fault itself is not marked but only a half-arrow indicating dextral shear sense is given. If the authors insist on the existence of an MVF, I see no reason why it should not be the same structure as the MöF. - -We use now only MöF (see discussion below).- -

Lines 105-108, Katschberg fault: The northern, E-W trending stretch of the KSZS is dextral, not sinistral. Somewhat northeast of and parallel to the north-western stretch of the MöF in Fig. 1, there is a sinistral shear zone that is also part of the Katschberg fault. - -This was corrected: "The eastern sub-dome is bordered to the east by the Katschberg Normal Fault (KNF), continuing to the north into the dextral Katschberg Shear Zone System (KSZS), and to the south into an unnamed sinistral shear zone and oriented parallel to the Mölltal Fault (MöF)."- -

Lines 116-118: Folding alone cannot have caused significant exhumation; the idea of Schmid et al. (2013) is that exhumation was achieved by extrusion of the HP unit that went together with folding. - -This was modified as follows: "In the Late Eocene, exhumation was achieved by extrusion of the high-pressure units that went together with major folding of the D2 thrust formed between the subducted Glockner Nappe System and Modereck Nappe System (D3 deformation of Schmid et al. 2013; Table 2)."- -

Lines 118-120: "... this exhumation episode ..." logically refers to D3. However, breakoff of the European slab was linked to D4 by Schmid et al. (2013). - -This was modified as follows: "In the Early Oligocene, nearly contemporaneous break off of the subducting European slab and formation of the Venediger Duplex (crustal-scale duplex structure) occurred, followed by the "Tauernkristallisation" (reheating of the whole nappe stack to amphibolite-facies conditions) (D4 deformation of Schmid et al. 2013; Table 2)."- -

Lines 127, 128: What is "isochron corrected Rb-Sr dating"? - -This was corrected

as follows: "Previous shear zone age dating in the TW was achieved using different geochronometers: Rb–Sr whole-rock–phengite dating (20 Ma; Blanckenburg et al., 1989), Rb–Sr whole-rock–white mica dating (39 – 16 Ma; Glodny et al., 2008) [. . .]"- -

Lines 186-193 (chapter 4 Results): No results but rather general information about fissure and cleft formation are given here. With respect to line 191, more complex (sigmoidal) shapes may also result from non-coaxial progressive deformation. In that case, also the orientation with respect to the foliation and lineation changes progressively. - -This paragraph was moved to the Introduction chapter- -

Lines 202-203: The sigmoidal shape of C2 fissures should indicate a shear sense. It would be good to give information about the shear sense of strike-slip faulting here. - -The following sentence was added to the "Field observations" section: "At Pfitscherjoch, the shape of C2 fissures, indicating overprinting by sinistral sense of shear, is in agreement with the larger scale sinistral shearing of the GSZ shear zone."- -

Lines 210-211: It is safer to write here "This would indicate a similar direction of extension for the development ..." - -This sentence was modified as follows: "For C2, this would indicate a similar direction of extension for the development of this fissure type, which is in line with paleostress orientations provided by Bertrand et al. (2015)."- -

Lines 212-218: Since C1 and C2 are perpendicular to stretching lineations L1 and L2 that are parallel to each other (Fig. 2c), I do not understand how it is possible to distinguish the two fissure generations based on their orientation. - -C1 and C2 are distinguished by the host rock foliation.- -

Lines 220, 222: According to Fig. 2b&c, C3 is not perpendicular to C1 and C2 and L3 is not down-dip. - -A third generation of fissures (C3, Fig. 2c) is observed, for example, in the Pfitscherjoch locality (Fig. 2a and b) and is at high angle to the steeply-oriented C1 fissures. This third and sub-horizontal fissure orientation, associated with a subvertical E-W oriented foliation and steep lineation (L3) (Fig. 2) may be in continuation of heterogeneous rotation of C2 fissures in the same shear zones. Stretching lineation related

to the BNF activity is sub-parallel to C3 lineation, however its foliation is oriented N-S (Fig. 2c). We suggest that sub-horizontal C3 fissures are related to strike-slip faulting.--

Lines 406-407: I find it a bit misleading to say the faults were active at 21, 17 and 12 Ma because the time ranges for the three phases of fissure formation given just before indicate almost continuous monazite growth between 22 and 8 Ma. - -The sentence was modified as follows: "Overall, fissure monazite age recording indicates that in the TW Cenozoic faults show increased activity at ∼21, ∼17 and ∼12 Ma, probably due to reorganization of plate movements occurring at those times."- -

Lines 407-410: The fission track age distribution in Fig. 7c is strictly speaking not U-shaped, but forms a dome or inverse U. More importantly, the relatively young fission track ages are from the margins of the Tauern dome from where the authors have not dated monazite samples. Therefore, I find also the conclusion that "monazite crystallization ages do not show the U-shaped distribution as cooling ages" misleading. A more valid conclusion would be in my opinion that the latest stages of monazite crystallisation happened at temperatures that are between apatite and fission track "closure" temperatures. - -The sentence was modified as follows: "Comparison of Th-Pb fissure monazite crystallization ages with existing crystallization and cooling ages (e.g. AFT, ZFT, white mica from fault zones) show that the latest stages of monazite crystallisation occurred at temperatures between apatite and zircon fission track "closure" temperatures."- -

Fig. 1: There is apparently no interference between the ZWD and the IsF. This is probably a result of compiling different maps and looks very peculiar. - -This is correct. We just keep ZWD and remove DAV and IsF Oligocene-age ductile faults.- -

Technical corrections - -Corrections are marked in blue in the manuscript:- -

Line 55: "... fissures form." Line 64: "... fluid that fills ..." or "... fluids that fill ..." Line 95: "... (1-25) ..." Lines 104, 648: "... Ahrntal shear zone ..." Line 120: I suggest to write

"This was followed by an inversion of subduction polarity ..." because it is clearer than "... slab dynamics ..." Line 192: "... studies ..." Line 197: I suggest to add "5" behind "sample locality" because then the place is much easier to find in Fig. 1 Line 210: "... in figure 2c ..." Line 215: "... fold axes ..." Line 221: I suggest to replace "displays" with "is related to" Lines 222, 223, 225: I suggest to replace "oriented" with "striking" Line 229: "... it shows ..." Line 240: "In a few cases the dates for specific monazite domains ..." Line 247: "... which delimits the TW to the west ..." Line 273: "... starts to grow (phase 1, red symbols in Fig. 7) ..." Line 290: "... the peak activity of which ..." Line 307: Replace "sinistral" with "dextral" Line 322: "... was active ..." Line 344: I suggest to replace "occurred" with "started" Line 366: I suggest to replace "Whereas" with "By contrast," Line 374: "... monazites ..." Line 388: Delete "decreases" Lines 395-396: "... during the Miocene." Line 404: "... associated with ..." Lines 444, 452, 457, 533, 585, 587, 634: Please add names of journal or book Lines 492, 495: Please check for special characters Legend of Fig 1: "Periadriatic" Lines 637-639: Below the legend of Fig. 1, but not in the figure caption, Scharf et al. (2013) is cited as a source for the map. Lines 648-649: "... Defereggen ... Mur-Mürz ..." (these are spelled incorrectly also in Schneider et al. 2013) Line 694: "... sample locations ..." Line 689: "... are presented ..." Line 695: "DD' NE-SW cross section across the BNF." Line 704: "Zircon and apatite fission track ages ..." Figure 7: Please integrate the labelling at the bottom of Fig. 7 ("(1) Early record ... oblique-slip") into the figure caption

Text at the bottom of Table 2: "Blanckenburg"

Supplementary information (SI) - -Corrections are marked in blue in the SI file:- -

Description of INNB1: "... between 7265-21,420 and ..." – likewise for descriptions of MAYR4, BURG2, SCHEI1, SALZ18, ORT1 SI, description of ZEI1: "ZEI1 ... can be subdivided in three domains (ZEI1-A, -B, and –C) ..." SI, description of SCHEI1: "... Table 5), thus assuming ..." SI, description of NOWA3: "... displays three ..." SI, description of GART3: "...GART3-C composition is strongly clustered ..." SI, description of STEI2: "... between 2676-12,353 and 131-407 ppm ..."